# Larger Datasets Can Be Repeated More: A Theoretical Analysis of Multi-Epoch Scaling in Linear Regression

**Tingkai Yan**[*β]   **Haodong Wen**[*α]   **Binghui Li**[*γ]
**Kairong Luo**[ν]   **Wenguang Chen**[νλ]   **Kaifeng Lyu**[†α]

[α]Institute for Interdisciplinary Information Sciences, Tsinghua University
[β]School of Mathematical Sciences, Peking University
[γ]Center for Machine Learning Research, Peking University
[ν]Department of Computer Science and Technology, Tsinghua University
[λ]Peng Cheng Laboratory
yantingkai66@gmail.com,   whd25@mails.tsinghua.edu.cn
libinghui@pku.edu.cn,   luokr24@mails.tsinghua.edu.cn
{cwg,klyu}@mail.tsinghua.edu.cn

## ABSTRACT

While data scaling laws of large language models (LLMs) have been widely examined in the one-pass regime with massive corpora, their form under limited data and repeated epochs remains largely unexplored. This paper presents a theoretical analysis of how a common workaround, training for multiple epochs on the same dataset, reshapes the data scaling laws in linear regression. Concretely, we ask: to match the performance of training on a dataset of size $N$ for $K$ epochs, how much larger must a dataset be if the model is trained for only one pass? We quantify this using the *effective reuse rate* of the data, $E(K, N)$, which we define as the multiplicative factor by which the dataset must grow under one-pass training to achieve the same test loss as $K$-epoch training. Our analysis precisely characterizes the scaling behavior of $E(K, N)$ for SGD in linear regression under either strong convexity or Zipf-distributed data: (1) When $K$ is small, we prove that $E(K, N) \approx K$, indicating that every new epoch yields a linear gain; (2) As $K$ increases, $E(K, N)$ plateaus at a problem-dependent value that grows with $N$ ($\Theta(\log N)$ for the strongly-convex case), implying that larger datasets can be repeated more times before the marginal benefit vanishes. These theoretical findings point out a neglected factor in a recent empirical study by Muennighoff et al. (2023), which claimed that training LLMs for up to 4 epochs results in negligible loss differences compared to using fresh data at each step, *i.e.*, $E(K, N) \approx K$ for $K \leq 4$ in our notation. Supported by further empirical validation with LLMs, our results reveal that the maximum $K$ value for which $E(K, N) \approx K$ in fact depends on the data size and distribution, and underscore the need to explicitly model both factors in future studies of scaling laws with data reuse.

## 1 INTRODUCTION

Scaling laws (Hestness et al., 2017; Kaplan et al., 2020; Hoffmann et al., 2022) have emerged as a central framework for characterizing the behavior of large language model (LLM) pre-training. The Chinchilla scaling law (Hoffmann et al., 2022) established robust empirical trends in performance as a joint function of model size and dataset size under the one-pass training paradigm, in which each data point is used at most once. This assumption, however, is becoming increasingly untenable. The quest for more capable models has driven an unprecedented escalation in data requirements: from fewer than 10 billion tokens for GPT-2, to 300 billion for GPT-3 (Brown et al., 2020), 2 trillion

---

[*]Equal contribution.
[†]Corresponding author.

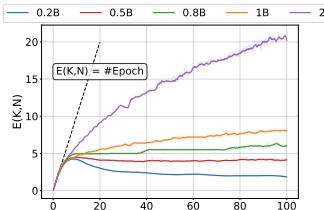 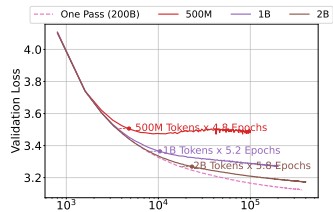 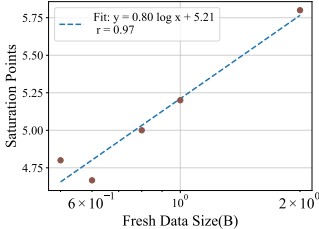

(a) $E(K, N)$ as a function of $K$.    (b) Validation loss of one-pass and multi-epoch training runs.    (c) The saturation points grows linearly with $\log N$.

Figure 1: Our LLM experiments show that larger datasets can be repeated more. **(a)** The effective reuse rate $E(K, N) \approx K$ when $K$ is small, but eventually saturates at certain number of epochs. **(b)** The saturation point shifts to larger $K$ as the dataset size increases. **(c)** Empirically, the saturation points scales linearly with $\log N$.

for Chinchilla and LLaMA 2 (Hoffmann et al., 2022; Touvron et al., 2023), and 36 trillion for Qwen3 (Yang et al., 2025). Projections further suggest that the pool of publicly available data may be exhausted as early as 2028 (Villalobos et al., 2024).

A common response to this emerging data scarcity is to train models for multiple epochs over the same dataset. Recent empirical studies have begun to examine the consequences of such repetition: for example, Muennighoff et al. (2023) and Xue et al. (2023) show that moderate reuse can still yield competitive pre-training performance. Yet the fundamental scaling behavior of multi-epoch training remains poorly understood—particularly from a theoretical standpoint.

In this paper, we study a fundamental question in understanding how multi-epoch training affects the data scaling laws: *To what extent can training for $K$ epochs on $N$ samples be effectively seen as one-pass training with an increased number of data samples?* Formally, let $\mathcal{L}(K, N)$ denote the expected loss of $K$-epoch training on $N$ samples. We define the *effective dataset size* $N'(K, N)$ as the minimal number of samples in one-pass training that achieves a comparable or lower loss $\mathcal{L}(1, N') \leq \mathcal{L}(K, N)$. In this paper, we are concerned with the ratio $E(K, N) = N'(K, N)/N$, which we term as the *effective reuse rate* of the data, a key quantity that characterizes how many times larger the dataset must grow to match the same performance as $K$-epoch training (see the detailed version in Definition 3.1).

In a recent study of scaling laws for multi-epoch training, Muennighoff et al. (2023) encountered this question and proposed an empirical approximation: $N'(K, N) = \left(1 + R^*(1 - e^{-(K-1)/R^*})\right) \cdot N$, where $R^*$ is a fitted constant ($R^* \approx 15.39$ in their experiments). This formula suggests that the benefit of repetition grows with $K$ but saturates exponentially at $(1 + R^*) \cdot N$ as $K$ increases. While supported by some empirical evidence in their study, this approximation still leads to a noticeable gap between scaling law predictions and empirical results (see Figure 3 in their paper). Moreover, the formula implies that the ratio $E(K, N) = N'(K, N)/N$ is independent of $N$, so the benefit of repeating the dataset $K$ times is equivalent to increasing its size by a factor that depends only on $K$, regardless of how large $N$ is. It remains unclear to what extent this independence holds in general.

**Our Contributions.** In this paper, we approach the above question on the effective reuse rate of data in the setting of linear regression, a setting that is simple enough to reveal the key mechanisms of data reuse, while still tractable for precise analysis under stochastic gradient descent (SGD). We provide a theoretical characterization of $E(K, N)$ in various regimes, and point out a neglected factor in the empirical study of Muennighoff et al. (2023): the effective reuse rate depends not only on the number of epochs $K$, but also on the dataset size $N$. In fact, we show both theoretically and empirically that **larger datasets can be repeated more**. Our main contributions are as follows:

1. In Section 4, we analyze multi-epoch SGD for linear regression under strong convexity. We show that when $K$ is small, $E(K, N) \approx K$, so every new epoch leads to a linear gain. As $K$ increases, $E(K, N)$ saturates at a problem-dependent value of order $\Theta(\log N)$, suggesting that larger datasets can be repeated for more epochs before the marginal benefit vanishes.

2. In Section 5, we go beyond the strongly convex case and study a class of Zipf-law distributed data, and show that $E(K, N)$ exhibits a similar scaling behavior to the strongly convex case, except that the saturation point scales as a power of $N$ instead of $\log N$.

3. Technically, we derive the optimal learning rate (Lemma 4.4) for multi-epoch SGD in linear regression and its corresponding approximation formula for the expected excess risk up to an $o(1)$ multiplicative error (Lemma G.1). These results may be of independent interest.

4. In Section 6, we conduct LLM pretraining experiments with up to 200B repeated tokens. Consistent with our theoretical findings, the empirical results show that $E(K, N) \approx K$ for small $K$, but eventually saturates at an epoch number that scales linearly with $\log N$. See also Figure 1 for details.

## 2 RELATED WORK

**Data Reuse in LLM Pre-Training.**    Empirically, there is a long debate over the effect of data reuse in LLM pre-training. Some works (Lee et al., 2021; Hoffmann et al., 2022; Hernandez et al., 2022; Wang et al., 2023) suggested it may be harmful, while some works (Taylor et al., 2022) reported the benefit of data reuse when the number of epochs is small ($K \leq 4$). Xue et al. (2023) then discovered a degradation phenomenon in multi-epoch training and investigated relevant factors and regularization methods to tackle it. Muennighoff et al. (2023) trained LLMs under different configurations and also found that reusing data is as good as using fresh data in the first few epochs. Yet, as the number of epochs increases, the returns for repetitions diminish. In our work, from a theoretical perspective, we rigorously analyze the effect of data reuse using non-asymptotic techniques, and we define and calculate the effective reuse rate under two cases, shedding light on the theoretical understanding of data reusing in LLM pre-training.

**Comparison with Lin et al. (2025).**    A recent study on linear regression with data reusing (Lin et al., 2025) is among the most relevant to our results.  They showed that when the number of epochs is relatively small (smaller than some power of the dataset size), the order of loss remains the same as one pass SGD for the same iterations, which aligns with our results. However, their results only imply that $E(K, N) = \Theta(K)$ for small $K$, while our analysis directly gives the explicit loss characterization with $o(1)$ relative error bound and a more exact description of the effective reuse rate, which reflects the data reusing scaling behaviour. Our analysis is across various problem setups, and further shows the general scaling trend of data reusing under different problem setups.

## 3 PRELIMINARIES

**Notations.**    We use $\|\cdot\|$ to denote the $\ell_2$-norm of vectors and the corresponding operator norm of matrices. For two sequences $(A_n)_{n=0}^{\infty}$ and $(B_n)_{n=0}^{\infty}$, we write $A_n = O(B_n)$, or alternatively $A_n \lesssim B_n$, $B_n = \Omega(A_n)$, $B_n \gtrsim A_n$, if there exist constants $C > 0, N > 0$ such that $|A_n| \leq C|B_n|$ for all $n \geq N$. We write $A_n = \Theta(B_n)$, or alternatively $A_n \asymp B_n$, if both $A_n = O(B_n)$ and $A_n = \Omega(B_n)$ hold. Moreover, for some variable $n$, we write $A_n = o_n(B_n)$ if for every constant $c > 0$, there exists $n_0 > 0$ such that $|A_n| < c|B_n|$ for all $n \geq n_0$. In this paper, when $n$ is clear from the context, we write $A_n = o(B_n)$ for short. Furthermore, we write $A_n = \omega(B_n)$ if $B_n = o(A_n)$. For matrices $\boldsymbol{A}_1, \boldsymbol{A}_2, \ldots, \boldsymbol{A}_n$, we use $\prod_{l=1}^{n} \boldsymbol{A}_l$ to denote the product $\boldsymbol{A}_1 \boldsymbol{A}_2 \ldots \boldsymbol{A}_n$. We define $\|\boldsymbol{u}\|_{\boldsymbol{S}} := \sqrt{\boldsymbol{u}^\top \boldsymbol{S} \boldsymbol{u}}$ for any vector $\boldsymbol{u}$ and any positive semi-definite (PSD) matrix $\boldsymbol{S}$.

**Linear Regression Problem.**    We focus on a linear regression setup, where data point $(\boldsymbol{x}, y) \in \mathbb{R}^d \times \mathbb{R}$ follows a joint distribution $\mathcal{D}$ and $\|\boldsymbol{x}\| \leq D$ for some constant $D$. W.L.O.G., we assume that the covariance matrix of data input is diagonal, i.e., $\boldsymbol{H} := \mathbb{E}[\boldsymbol{x}\boldsymbol{x}^\top] = \mathrm{diag}(\lambda_1, \lambda_2, \ldots, \lambda_d)$, where $\lambda_1 \geq \lambda_2 \geq \cdots \geq \lambda_d$. A direct corollary is that $\lambda_1 \leq D^2$. For a given data input $\boldsymbol{x}$, the label $y$ is generated by $y := \langle \boldsymbol{w}^*, \boldsymbol{x} \rangle + \xi$, where $\boldsymbol{w}^* \in \mathbb{R}^d$ is the ground-truth weight and $\xi$ represents the independent random label noise with $\mathbb{E}[\xi] = 0$ and $\mathbb{E}[\xi^2] = \sigma^2$. We aim to train a linear model $f(\boldsymbol{x}; \boldsymbol{w}) := \langle \boldsymbol{w}, \boldsymbol{x} \rangle$ to predict the data label, where $\boldsymbol{w} \in \mathbb{R}^d$ is the trainable parameter. We use MSE-loss $\ell(\boldsymbol{w}; \boldsymbol{x}, y) := \frac{1}{2}(f(\boldsymbol{x}; \boldsymbol{w}) - y)^2$ to measure the fitting error. Then, the population loss is defined as $\mathcal{L}(\boldsymbol{w}) := \mathbb{E}_{(\boldsymbol{x}, y) \sim \mathcal{D}}[\ell(\boldsymbol{w}; \boldsymbol{x}, y)]$. Further we define the excess risk $\mathcal{R}(\boldsymbol{w}) := \mathcal{L}(\boldsymbol{w}) - \frac{1}{2}\sigma^2$, which is the expected population loss minus the irreducible loss $\frac{1}{2}\sigma^2$.

**Multi-Epoch SGD Training Algorithm.**    Consider a finite training dataset with $N$ data points $\{(\boldsymbol{x}_0, y_0), (\boldsymbol{x}_1, y_1), \ldots, (\boldsymbol{x}_{N-1}, y_{N-1})\}$, where the data points $(\boldsymbol{x}_i, y_i)$ are i.i.d. sampled from the distribution $\mathcal{D}$. We use $K$-epoch stochastic gradient descent (SGD) with random shuffling to minimize the loss function. And the initial parameter $\boldsymbol{w}_0$ is set to $0$. Formally, we denote $K$ independent random permutations of $[N]$ by $\pi_1, \ldots, \pi_K$. And we define $j_t := \pi_{k_t}(i_t)$, where $i_t := t \bmod N$, $k_t := \lfloor t/N \rfloor + 1$. Then we have the update rule for $K$-epoch SGD with $N$ data points

$$\boldsymbol{w}_{t+1} = \boldsymbol{w}_t - \eta \nabla_{\boldsymbol{w}} \ell(\boldsymbol{w}_t; \boldsymbol{x}_{j_t}, y_{j_t}) = \left(\boldsymbol{I} - \eta \boldsymbol{x}_{j_t} \boldsymbol{x}_{j_t}^\top\right) \boldsymbol{w}_t + \eta \xi_{j_t} \boldsymbol{x}_{j_t}.$$

Next, given a $K$-epoch SGD over $N$ data points, with learning rate $\eta$, we define $\mathcal{W}_{K,N,\eta}$ to be the distribution of $\boldsymbol{w}_{KN}$. The randomness within $\boldsymbol{w}_{KN}$ comes from the random draw of the dataset, label noise $\xi$, and the shuffling in SGD. Based on this, we define the expected excess risk of a given $K$-epoch SGD over $N$ data points, with learning rate $\eta$ as $\bar{\mathcal{R}}(K, N; \eta) := \mathbb{E}_{\boldsymbol{w} \sim \mathcal{W}_{K,N,\eta}}[\mathcal{R}(\boldsymbol{w})]$. We assume $\eta \leq D^{-2}$ for training stability.

We also consider multi-epoch training under more practical training configurations, for example, using a class of polynomially decaying learning-rate schedules. See Section K for more details.

**Comparing Performance under Optimal Learning Rate Regime.** To compare the performance of one-pass and multi-epoch SGD, we consider the settings where the learning rates for both methods are optimally tuned. Formally, we introduce the notion of the *optimal expected excess risk* of $K$-epoch SGD for $N$ samples as $\bar{\mathcal{R}}^*(K, N) := \min_{\eta \in (0, \frac{1}{D^2}]} \{\bar{\mathcal{R}}(K, N; \eta)\}$. To calculate this value analytically, in the next section, we will show that we can get a learning rate choice that can approximately achieve the above optimal expected excess risk $\bar{\mathcal{R}}^*(K, N)$ both for one-pass and multi-epoch SGD. Following our discussion in the introduction, we define the *effective reuse rate* as follows:

**Definition 3.1** (Effective Reuse Rate). *Given $K$-epoch SGD trained with $N$ fresh data samples, the effective reuse rate is defined as:* $E(K, N) := \frac{1}{N} \min\{N' \geq 0 : \bar{\mathcal{R}}^*(1, N') \leq \bar{\mathcal{R}}^*(K, N)\}$.

That is, the effective reuse rate measures how many times larger the dataset must grow under one-pass training to match the performance of $K$-epoch training, both under the optimal learning rate regime.

# 4 MULTI-EPOCH SCALING IN STRONGLY CONVEX LINEAR REGRESSION

In the study of linear regression problems, the strongly convex case is a classical and central theoretical framework, serving as the standard entry point before relaxing to weaker conditions in many analyses (Hastie, 2009; Ge et al., 2019). In Section 4.1, we first give the problem setups and the main results of the effective reuse rate. In Section 4.2, we give a proof sketch for our theoretical results, and the detailed proof of this section can be found in Appendix G.

## 4.1 MAIN RESULTS

As we focus on the strongly convex case, we make the following assumption on the minimum eigenvalue of the Hessian matrix.

**Assumption 4.1** (Strong Convexity). *We assume that $\lambda_d \geq \mu$ for some constant $\mu > 0$.*

For simplicity, we make the following prior for the ground-truth weight $\boldsymbol{w}^*$.

**Assumption 4.2** (Parameter Prior). *The ground truth $\boldsymbol{w}^*$ satisfies $w_i^* \neq 0$ for all $i \in [d]$.*

As the number of samples $N$ can be very large in practice, training on the entire dataset for a large amount of epochs can be computationally expensive. This motivates us to impose an upper bound on the number of epochs $K$. Technically, this ensures that the accumulated error terms remain manageable in our analysis.

**Assumption 4.3** (Computationally feasible number of epochs). *We assume that the training dataset size $N$ and number of epochs $K$ satisfy $K = O(N^{0.1})$.*

Here, the exponent $0.1$ is chosen for ease of calculation, though it may not be tight.

To compute $E(K, N)$, we first precisely characterize the optimal expected excess risk. In particular, we derive asymptotic expansions for $\bar{\mathcal{R}}^*(K, N)$ in the regimes $K = o(\log N)$ and $K = \omega(\log N)$, each expressed as a leading term accompanied by an explicitly controlled higher-order remainder.

**Theorem 4.1** (Multi-Epoch Data Scaling Law). *Under Assumptions 4.1 to 4.3, for multi-epoch SGD with the number of epochs $K$, dataset size of $N$, it holds that*

$$\bar{\mathcal{R}}^*(K, N) = \begin{cases} \frac{\sigma^2 \operatorname{tr}(\boldsymbol{H})}{8\lambda_d}(1 + o_N(1)) \cdot \frac{\log(KN)}{KN} & \text{for } K = o(\log N), \\ \frac{\sigma^2 d}{2}(1 + o_N(1)) \cdot \frac{1}{N} & \text{for } K = \omega(\log N). \end{cases}$$

Theorem 4.1 describes how expected excess risk decays with number of epochs $K$ and dataset size $N$ when choosing the optimal learning rate. When $K \ll \log N$, then $\bar{\mathcal{R}}^*(K, N) = \Theta\left(\frac{\log T}{T}\right)$ where

$T = KN$; by contrast, when $K \gg \log N$, then $\bar{\mathcal{R}}^*(K, N) = \Theta\left(\frac{1}{N}\right)$ which does not depend on $K$, showing that endless data reuse turns out to be useless.

Next we propose the expression of $E(K, N)$ by applying Theorem 4.1.

**Theorem 4.2.** *Under Assumptions 4.1 to 4.3, for multi-epoch SGD with the number of epochs $K$, dataset size of $N$, it holds that*

$$E(K, N) = \begin{cases} (1 + o_N(1)) \cdot K & \text{for } K = o(\log N), \\ \frac{\text{tr}(\boldsymbol{H})}{4\lambda_d d}(1 + o_N(1)) \cdot \log N & \text{for } K = \omega(\log N). \end{cases}$$

Theorem 4.2 pinpoints two regimes for the effective reuse rate in the strongly convex case. The first one is the **effective-reuse regime**: when $K \ll \log N$, then $E(K, N) = K(1 + o(1))$. This suggests that each extra epoch is essentially as valuable as a fresh pass. The second one is the **limited-reuse regime**: when $K \gg \log N$, then $E(K, N) = \frac{\text{tr}(\boldsymbol{H}) \log N}{4\lambda_d d}(1 + o_N(1))$, which means additional epochs yield only logarithmic gains. This further implies that the model has effectively "seen" the dataset enough times that additional repetition is redundant.

Together, these two asymptotic descriptions expose a phase transition when the quantity $\lim_{N \to \infty} \frac{K}{\log N}$ changes from 0 to $\infty$. For the former case ($\lim_{N \to \infty} \frac{K}{\log N} = 0$), multi-epoch training behaves like unlimited data augmentation; for the latter ($\lim_{N \to \infty} \frac{K}{\log N} = \infty$), the benefits of reusing data all but vanish, capping $E(K, N)$ at $\Theta(\log N)$.

**Larger Datasets Can Be Repeated More.** Our theorem provides the following insight. Fixing the data distribution, as we collect more data, the largest possible epoch number $K$ to stay in the effective-reuse regime also increases. Consequently, larger datasets can sustain more epochs while maintaining linear gains from data reuse. Specifically, for the setup we study in this section, if we have collected $N$ data points in total, then with multi-epoch training, we can get a performance comparable to one-pass training on $\Theta(N \log N)$ data points, which is superlinear in the number of data points we collected. This finding highlights a neglected factor in the data-constrained scaling laws proposed in Muennighoff et al. (2023), which assumed a uniform effective number of epochs across different fresh data sizes. In Section 6.3, we validate this insight by showing that the effective reuse rate indeed increases with the dataset size in LLM pretraining.

**Discussion: Effect of Learning Rate Decay.** The factor $\log N$ is in fact associated with the use of constant learning rate in our setting, where $\eta$ may depend on the total number of steps but remains fixed throughout training. While our main focus is on the constant learning rate setting, introducing learning rate decay may lead to different scaling behaviors and warrants further theoretical investigation. Under learning rate decay, for example using $\eta_t = \frac{\eta_0}{1 + bt^\alpha}$ with $b > 0$ and $0.5 < \alpha \leq 1$ at step $t$, by a direct calculation based on the setup of Theorem 1 in Ge et al. (2019), it is not hard to show that one-pass training yields $\bar{\mathcal{R}}^*(1, N) = O(\kappa_{\boldsymbol{H}} \cdot \frac{\sigma^2 d}{N})$ for sufficiently large $N$ when $\alpha = 1$, where $\kappa_{\boldsymbol{H}} := \frac{\lambda_1}{\lambda_d}$ is the condition number of $\boldsymbol{H}$. This bound loses only by a constant factor compared to the statistical minimax rate $\frac{\sigma^2 d}{2N}$ implied by the Cramér–Rao lower bound (Jain et al., 2017). Therefore, under learning rate decay, the effective reuse rate of multi-epoch training is at most $O(\kappa_{\boldsymbol{H}})$. In Section K, we provide a preliminary analysis comparing one-pass and multi-epoch training in the limit of infinitely many epochs, showing that multi-epoch training can almost attain the statistical minimax rate and achieve $E(\infty, N) = \Omega(\kappa_{\boldsymbol{H}})$. It remains an interesting direction to understand how $E(K, N)$ saturates to this value and how it depends on the dataset size $N$.

### 4.2 PROOF SKETCH

We now provide a proof sketch of our main results. First, we need to compute the optimal expected excess risk $\bar{\mathcal{R}}^*(K, N)$. This requires us to compute $\bar{\mathcal{R}}(K, N; \eta)$ and then select the optimal learning rate $\eta^*$ that minimizes $\bar{\mathcal{R}}(K, N; \eta)$. However, due to the random shuffling and multi-pass processing of the training data, directly analyzing $\bar{\mathcal{R}}(K, N; \eta)$ is intractable. To overcome this, we seek an analytic approximation of $\bar{\mathcal{R}}(K, N; \eta)$, which is derived through the following steps.

**Step 1: Bias-Variance Decomposition for Training Dynamics.** Following the widely-applied bias-variance decomposition approach to analyzing the dynamics of SGD training (Neu & Rosasco, 2018; Ge et al., 2019; Zou et al., 2021; Wu et al., 2022a), we define $\boldsymbol{\theta}_t = \boldsymbol{w}_t - \boldsymbol{w}^*$ and examine the following two processes of bias and variance: $\boldsymbol{\theta}_{t+1}^{\text{bias}} = \boldsymbol{\theta}_t^{\text{bias}} - \eta \langle \boldsymbol{\theta}_t^{\text{bias}}, \boldsymbol{x}_{j_t} \rangle \boldsymbol{x}_{j_t}, \boldsymbol{\theta}_{t+1}^{\text{var}} =$

$\boldsymbol{\theta}_t^{\text{var}} - \eta \langle \boldsymbol{\theta}_t^{\text{var}}, \boldsymbol{x}_{j_t} \rangle \boldsymbol{x}_{j_t} + \eta \xi_{j_t} \boldsymbol{x}_{j_t}$, where the two processes are initialized as $\boldsymbol{\theta}_0^{\text{bias}} = \boldsymbol{w}_0 - \boldsymbol{w}^*$ and $\boldsymbol{\theta}_0^{\text{var}} = \boldsymbol{0}$. It follows that $\boldsymbol{\theta}_t = \boldsymbol{\theta}_t^{\text{bias}} + \boldsymbol{\theta}_t^{\text{var}}$, with $\mathbb{E}[\boldsymbol{\theta}_t^{\text{var}}] = \boldsymbol{0}$. We can then decompose the excess risk $\mathcal{R}(\boldsymbol{w}_t)$ into two components: the *bias term* and the *variance term*, which we formalize as follows $\mathcal{R}(\boldsymbol{w}_t) = \frac{1}{2} \|\boldsymbol{\theta}_t\|_{\boldsymbol{H}}^2 = \frac{1}{2} \|\boldsymbol{\theta}_t^{\text{bias}}\|_{\boldsymbol{H}}^2 + \frac{1}{2} \|\boldsymbol{\theta}_t^{\text{var}}\|_{\boldsymbol{H}}^2$.

**Step 2: Analytic Risk Approximation by Matrix Concentration.** A key challenge in tracking the dynamics of multi-epoch SGD training arises from the non-commutative nature of the matrices in the weight updates, which depend on randomly shuffled and multi-pass data. For example, the bias weight evolves as $\boldsymbol{\theta}_{KN}^{\text{bias}} = \left( \prod_{k=1}^{K} \left( \prod_{l=1}^{N} \left( \boldsymbol{I} - \eta \boldsymbol{x}_{\pi_k(l)} \boldsymbol{x}_{\pi_k(l)}^{\top} \right) \right) \right) \boldsymbol{\theta}_0^{\text{bias}}$, where we can see that one data point appears more than once across different epochs. Thus, the above matrix multiplication involves massive correlated data, which makes calculating the bias term $\mathbb{E}\left[ \|\boldsymbol{\theta}_{KN}^{\text{bias}}\|_{\boldsymbol{H}}^2 \right]$ intractable. To resolve this issue, we borrow tools from concentration inequalities for matrix products Huang et al. (2022). Specifically, we use the following result:

**Lemma 4.1** (Corollary of Theorem 7.1 in Huang et al. (2022)). *Given $n$ data points such that* $\boldsymbol{z}_0, \cdots \boldsymbol{z}_{n-1} \overset{i.i.d}{\sim} \mathcal{N}(0, \boldsymbol{H})$, *and defining* $\boldsymbol{A} = \prod_{j=0}^{n-1} \left( \boldsymbol{I} - \eta \boldsymbol{z}_j \boldsymbol{z}_j^{\top} \right)$, *we have* $\mathbb{E}\|\boldsymbol{A} - \mathbb{E}\boldsymbol{A}\|^l \leq \left( \sqrt{\delta_{\text{A}} \eta^2 n l} \right)^l$, *where* $\delta_{\text{A}} := \tilde{C} 8 e D^4 \log d$ *for some absolute constant* $\tilde{C} > 0$.

However, several obstacles prevent us from directly applying Lemma 4.1 to our problem. For example, we actually need to control error terms like $\mathbb{E}\left\| \prod_{i=K}^{k+1} \boldsymbol{A}^{(i)} - (\mathbb{E}\boldsymbol{A})^l \right\|$, where $\boldsymbol{A}^{(i)}$ represents the product of sequential updates through all samples in epoch $i$ (see the formal definition in Equation (1), Appendix E). To address this, our main idea is to derive a tight upper bound for the original term, and decompose this upper bound into the sum of a series of sub-terms for which we can apply Lemma 4.1. (see the detailed derivation in Appendix G.2.1 and Appendix G.2.2)

Finally, we derive an error bound on matrix deviations based on our calculations, which is a higher-order infinitesimal of the main term when $\eta \in \left[ \Omega\left(T^{-1}\right), o(T^{-\frac{3}{4}}) \right]$ and $K = o\left( \eta^{-1} T^{-\frac{3}{4}} \right)$, with $T := KN$ denoting the total number of training steps. This provides a theoretical guarantee for us to approximate the risk function with a tractable expression. For the bias term, we have

$$\mathbb{E}\left[ \|\boldsymbol{\theta}_{KN}^{\text{bias}}\|_{\boldsymbol{H}}^2 \right] = \mathbb{E}\left[ \left\| \left( \prod_{k=1}^{K} \left( \prod_{l=0}^{N-1} \left( \boldsymbol{I} - \eta \boldsymbol{x}_{\pi_k(l)} \boldsymbol{x}_{\pi_k(l)}^{\top} \right) \right) \right) \boldsymbol{\theta}_0 \right\|_{\boldsymbol{H}}^2 \right]$$

$$\approx \left\| \left( \prod_{k=1}^{K} \mathbb{E}\left[ \prod_{l=0}^{N-1} \left( \boldsymbol{I} - \eta \boldsymbol{x}_{\pi_k(l)} \boldsymbol{x}_{\pi_k(l)}^{\top} \right) \right] \right) \boldsymbol{\theta}_0 \right\|_{\boldsymbol{H}}^2$$

$$= \left\| \left( (\boldsymbol{I} - \eta \boldsymbol{H})^{KN} \right) \boldsymbol{\theta}_0 \right\|_{\boldsymbol{H}}^2,$$

where the approximation step follows from Lemma 4.1, and the last equation follows the facts that $\mathbb{E}\left[ \boldsymbol{x}_{\pi_k(l)} \boldsymbol{x}_{\pi_k(l)}^{\top} \right] = \boldsymbol{H}$ and $\boldsymbol{x}_i$ is uncorrelated with $\boldsymbol{x}_j$ for $i \neq j$. For the variance term, the data correlation issue is similar to what we met in the bias term case. Again, leveraging Lemma 4.1 and following a similar analysis, we can get an approximation formula for the variance term as shown:

$$\mathbb{E}\left[ \|\boldsymbol{\theta}_{KN}^{\text{var}}\|_{\boldsymbol{H}}^2 \right] \approx \frac{2\sigma^2}{N} tr \left( \frac{\left( \boldsymbol{I} - (\boldsymbol{I} - \eta \boldsymbol{H})^{KN} \right) \left( (\boldsymbol{I} - \eta \boldsymbol{H})^N - (\boldsymbol{I} - \eta \boldsymbol{H})^{KN} \right)}{\boldsymbol{I} + (\boldsymbol{I} - \eta \boldsymbol{H})^N} \right)$$

$$+ \eta \sigma^2 \left\langle \boldsymbol{H}, (\boldsymbol{I} - (\boldsymbol{I} - \eta \boldsymbol{H})^{2KN})(2\boldsymbol{I} - \eta \boldsymbol{H})^{-1} \right\rangle.$$

**Step 3: Narrowing the Range for Optimal Learning Rate.** However, although we have an analytic approximation for risk, it is important to note that this approximation holds only for a specific range of parameters. For a detailed discussion, refer to Lemma G.1. To mitigate this, we first determine a reasonable range for the optimal learning rate in two steps: First, we choose $\tilde{\eta} = \frac{\log KN}{2\lambda_d KN}$ as a reference learning rate; Then, by comparing the losses for the reference learning rate and other candidate learning rates, we can eliminate a large range of values. This analysis helps narrow down the potential range of learning rates (Lemma G.5 for small $K$ and Lemma G.6 for large $K$). Within

this range, we further simplify the risk approximation to make it more tractable for optimization, as shown in the following lemmas:

**Lemma 4.2** (Small $K$). *Let $\boldsymbol{H} = \boldsymbol{P}\boldsymbol{D}\boldsymbol{P}^\top$ be the canonical form of $\boldsymbol{H}$ under similarity, and let $\tilde{\theta}_d^2 := \sum_{l=d-n_d+1}^{d} (\boldsymbol{P}\boldsymbol{\theta}_0)_l^2$. Under Assumption 4.1 and 4.3, for learning rate $\eta \in \left[ \frac{\log KN}{3\lambda_d KN}, \frac{D^2 \mathrm{tr}(\boldsymbol{H}) \log KN}{\lambda_d \mathrm{tr}(\boldsymbol{H}^2) KN} \right]$, $K = o(\log N)$, we have $\bar{\mathcal{R}}(K, N; \eta) = M(K, N; \eta)(1 + o(1))$ with $M(K, N; \eta) := \frac{1}{2} \tilde{\theta}_d^2 \lambda_d \exp(-2\lambda_d \eta KN) + \frac{\eta \mathrm{tr}(\boldsymbol{H})\sigma^2}{4}$.*

**Lemma 4.3** (Large $K$). *We define $\tilde{\theta}_d^2$ as the same as Lemma 4.2. Under Assumption 4.1 and 4.3, for learning rate $\eta \in [\frac{\log KN}{3\lambda_d KN}, o\left(\frac{1}{N}\right)]$ and $K = \omega(\log N)$, we have $\bar{\mathcal{R}}(K, N; \eta) = M(K, N; \eta)(1 + o(1))$ with $M(K, N; \eta) = \frac{1}{2} \tilde{\theta}_d^2 \lambda_d \exp(-2\lambda_d \eta KN) + \frac{\eta \mathrm{tr}(\boldsymbol{H})\sigma^2}{4} + \frac{\sigma^2 d}{2N}$.*

**Step 4: Deriving the Approximately Optimal Learning Rate.** At this point, we have narrowed down the range for the optimal learning rate and simplified the risk approximation. The next step is to approximate the optimal expected excess risk. To achieve this, we differentiate the simplified risk function $M(K, N; \eta)$ in Lemma 4.2 and Lemma 4.3 with respect to the learning rate $\eta$ and give the critical point $\eta = \eta'(K, N)$, which are presented as follows:

**Lemma 4.4** (Approximately Optimal Learning Rate). *Under Assumption 4.1 and 4.3, we consider $K$-epoch SGD with $N$ fresh data and learning rate $\eta = \eta'(K, N) = \frac{\log \rho KN}{2\lambda_d KN}$, where $\rho := \frac{4\tilde{\theta}_d^2 \lambda_d}{\mathrm{tr}(\boldsymbol{H})\sigma^2}$. Then it holds for $K = o(\log N)$ or $K = \omega(\log N)$ that $\bar{\mathcal{R}}(K, N; \eta'(K, N)) = \bar{\mathcal{R}}^*(K, N)\left(1 + o(1)\right)$.*

Using Lemma 4.4, we complete the proof as follows. By evaluating the risk at the approximately optimal learning rate $\eta'(K, N) = \frac{\log \rho KN}{2\lambda_d KN}$, we obtain an approximation of the optimal risk (Theorem 4.1), based on which we derive the effective reuse rate (Theorem 4.2).

# 5 A SOLVABLE CASE WITH ZIPF-DISTRIBUTED DATA

Natural data distributions often exhibit power law structures. To capture this phenomenon, we go beyond the strongly convex case and analyze a stylized linear regression model with Zipf-distributed data, where the excess risk admits a closed-form expression and the effective reuse rate can be characterized explicitly.

Through this setup, we can see that the effective reuse rate exhibits a similar scaling behavior: as the number of epochs $K$ increases, $E(K, N)$ initially grows linearly but eventually saturates at a problem-dependent value that increases with $N$. In contrast to the strongly convex case, however, the saturation point does not scale as $\sim \log N$ but instead scales as a power of $N$.

**Problem Setup.** We use the same notation for excess risk, one-pass and multi-epoch SGD, and *i.i.d.* training data as in Section 3. We specify the data distribution as a Zipf distribution over $d$ one-hot data points, where the $i$-th data point is $\boldsymbol{x}^{(i)} = \mu_i \boldsymbol{e}_i$ for some $\mu_i > 0$ and the probability of sampling the $i$-th data point is $p_i = c \cdot i^{-\alpha}$ for some constants $c > 0$ and $\alpha > 1$. The label is generated by $y = \langle \boldsymbol{w}^*, \boldsymbol{x} \rangle$ with no label noise. The ground-truth weight $\boldsymbol{w}^* \in \mathbb{R}^d$ follows an isotropic prior distribution.

**Assumption 5.1** (Parameter Prior). *$\boldsymbol{w}^*$ is sampled from a prior distribution with $\mathbb{E}[\boldsymbol{w}^* \boldsymbol{w}^{*\top}] = \boldsymbol{I}$.*

**Interpretation.** This setup can be interpreted as a simplified model of real-world data with heavy-tailed feature distributions. Each coordinate represents an atomic feature that appears with Zipf-distributed probability, mimicking the long-tailed statistics observed in domains such as text and natural language. The scaling factors $\mu_i$ encode feature importance, which may reflect, for instance, effects introduced by feature weighting or normalization.

## 5.1 RESULTS ON POWER-LAW SPECTRUM

**Assumption 5.2** (Power-Law Spectrum). *There exist two constants $a, b > 0$ with $a - b > 1$ such that the data input distribution satisfies that $p_i = ci^{-(a-b)}$ and $\Lambda_i = i^{-b}$, where $c = \left(\sum_{i=1}^{d} \frac{1}{i^{a-b}}\right)^{-1}$.*

Here we establish matching upper and lower bounds for $\bar{\mathcal{R}}^*(K, N)$ in the small-$K$ and large-$K$ regimes, given the solvable model. Comparing with the strongly convex case, we observe a different scaling behavior: when $K \ll N^{\frac{b}{a-b}}$, $\bar{\mathcal{R}}^*(K, N)$ decays as a power law in $KN$, with exponent $\frac{a-1}{a}$; whereas when $K \gg N^{\frac{b}{a-b}}$, $\bar{\mathcal{R}}^*(K, N)$ exhibits a power-law decay in $N$ and is independent of $K$.

**Theorem 5.1.** *Consider a $K$-epoch SGD over $N$ fresh data. Under Assumptions 5.1-5.2, and given the data dimension $d = \Omega((KN)^{\frac{1}{a}})$, it holds that*

$$\bar{\mathcal{R}}^*(K, N) \asymp \begin{cases} (KN)^{-\frac{a-1}{a}} & \text{for } K = o(N^{\frac{b}{a-b}}) \\ N^{-\frac{a-1}{a-b}} & \text{for } K = \omega(N^{\frac{b}{a-b}}). \end{cases}$$

Then we derive the formula of $E(K, N)$ by first solving the equation $\bar{\mathcal{R}}^*(1, T') = \bar{\mathcal{R}}^*(K, N)$ based on Theorem 5.1, and divide $T'$ by $N$.

**Theorem 5.2** (Multi-Epoch Scaling Under Power-Law Spectrum). *Consider a $K$-epoch SGD over $N$ fresh data. Under Assumptions 5.1-5.2, and given the data dimension $d = \Omega((KN)^{\frac{1}{a}})$, it holds that*

$$E(K, N) = \begin{cases} K(1 + o(1)) & \text{for } K = o(N^{\frac{b}{a-b}}) \\ \Theta(N^{\frac{b}{a-b}}) & \text{for } K = \omega(N^{\frac{b}{a-b}}). \end{cases}$$

Under the assumption of a logarithmic power-law spectrum, the trend of the effective reuse rate as a function of $K$ approximates the phenomena described in Theorem 4.2 in the strongly convex setting and the trend described in Theorem 5.2 under the power-law spectrum assumption. We still observe an effective-reuse regime ($E(K, N) \approx K$) when $K$ is relatively small ($K \ll N^{b/(a-b)}$), and as $K$ increases, the effective reuse rate undergoes a phase transition, converging to an upper bound determined by $N$, entering the limited-reuse regime ($E(K, N) = \Theta(N^{b/(a-b)})$).

We can see that the exponent of this power of $N$ is determined by the rate of eigenvalue decay of the Hessian and the rate of norm decay of the parameter with respect to dimension. The proofs of Theorem 5.1 and Theorem 5.2 are given in Appendix I.2 and Appendix I.3 respectively.

## 5.2 Results on Logarithmic Power-law Spectrum

Further, we aim to understand under the same Hessian matrix, how the data distribution correlated with $P$ and $\Lambda$ affects the effective reusing rate. By changing the spectrum of $\Lambda$, we can also obtain matching upper lower bounds for $\bar{\mathcal{R}}^*(K, N)$ and a characterization for $E(K, N)$, which behave differently from the power-spectrum case. Here we present only the latter; the former can be seen in Appendix D.

**Assumption 5.3** (Logarithmic Power-Law Spectrum). *There exist two constants $a > 1, b > 0$ such that the data input distribution satisfies that $p_i = ci^{-a} \log^b(i + 1)$ and $\Lambda_i = 1/\log^b(i + 1)$, where $c = \left(\sum_{i=1}^{d} i^{-a} \log^b(i + 1)\right)^{-1}$.*

**Theorem 5.3** (Multi-Epoch Scaling Under Logarithmic Power-Law Spectrum). *Under Assumptions 5.1, Assumption 5.3, and given the data dimension $d = \Omega((KN)^{\frac{1}{a}})$ for a one-pass SGD and a $K$-epoch SGD over $N$ fresh data, it holds that*

$$E(K, N) = \begin{cases} K(1 + o(1)) & \text{for } K = o(\log^b N) \\ \Theta(\log^b N) & \text{for } K = \omega(\log^b N). \end{cases}$$

**The Saturation Point Varies across Different Problem Setups.** The phase transition point where the effectiveness of data reusing changes from the effective-reuse regime to the limited-reuse regime varies across different problem setups. In strongly convex linear regression problems, this phase transition happens when the limit $\lim_{K \to \infty} \frac{K}{\log N}$ changes from 0 to $\infty$. And in the above power spectrum and log-power spectrum case, the limit turns to be $\lim_{K \to \infty} \frac{K}{N^{b/(a-b)}}$ and $\lim_{K \to \infty} \frac{K}{\log^b N}$.

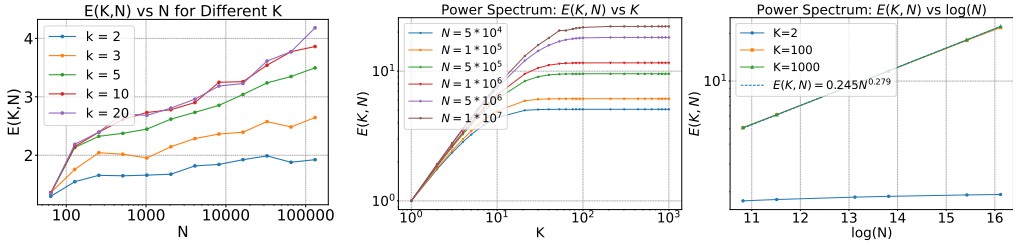

(a) Strongly convex case: $E(K, N)$ with $N$.  (b) The solvable case with Zipf-distributed data and power spectrum: $E(K, N)$ versus $K$ and $N$.

Figure 2: Simulation experiments for strongly-convex linear regression and the solvable case with Zipf-distributed data and power spectrum. Results show that $E(K, N)$ is approximately proportional to some function of $N$ when $N$ is relatively small, and $E(K, N) \approx K$ when $N$ is relatively large. For the solvable case with Zipf-distributed data and power spectrum, we also fit the effective reuse rate using the formula $E(K, N) = c_1 N^{c_2}$ suggested by Theorem 5.2, and the fitted exponent $c_2 = 0.279 \approx \frac{b}{a-b} = \frac{2}{7}$ matches our theory.

## 6 EXPERIMENTS

### 6.1 SIMULATIONS IN SECTION 4

First, we conduct our experiments on synthetic datasets with a strongly convex linear regression to verify the characterization of effective reuse rate $E(K, N)$ in Theorem 4.2.

**Experiments Setup.** We generate data pairs $(\boldsymbol{x}_i, y_i)$ where $\boldsymbol{x}_i \overset{i.i.d}{\sim} \mathcal{N}(0, \boldsymbol{I}_d)$ with dimension $d = 100$. For the label $y_i$, we generate it as $y_i = \langle \boldsymbol{w}^*, \boldsymbol{x}_i \rangle + \xi_i$, where $\boldsymbol{w}^*$ is the ground truth generated by standard Gaussian with unit variance. Also, $\xi_i \overset{i.i.d}{\sim} \mathcal{N}(0, \sigma^2 \boldsymbol{I}_d)$. Here in our simulation, we set $\sigma$ to 0.1. To make our simulation aligned with the theoretical setup, we set the learning rate $\eta \propto \frac{\log KN}{KN}$, and we grid search the ratio $c := \frac{\eta}{\log KN/KN}$ for the $c^*$ which minimizes the final loss given training steps $T = KN$.

**Results.** As shown in Figure 2a, we plot $E(K, N)$ as a function of $\log N$ for various fixed values of $K$. Each curve corresponds to a fixed number of epochs (e.g., $K = 3, 5, \ldots, 20$) and illustrates how the effective reuse rate $E(K, N)$ grows with dataset size. For small data size ($\log N \ll K$), the effective reuse factor increases roughly linearly with $\log N$, indicating that adding more data substantially boosts the one-pass equivalent performance. However, as $N$ becomes large ($\log N \gg K$), each curve flattens out and approaches an asymptote at $E(K, N) \approx K$. In other words, once the dataset is sufficiently large relative to the number of epochs, additional passes through the same data yield no further benefit beyond a factor of $K$. This behavior is exactly as predicted by Theorem 4.2: when $K$ is much smaller than $\log N$, we have $E(K, N) \approx K$ (nearly full $K$-fold data reuse), whereas when $K$ is large relative to $\log N$, the effective reuse saturates and grows only on the order of $\log N$.

### 6.2 SIMULATIONS IN SECTION 5.1

We now verify the predictions of Theorem 5.2 using synthetic data generated under the spectral assumptions of Section 5 with a power-law decay Hessian spectrum (Assumption 5.2). In all sub-figures of Figure 2b, we set the data dimension $d$ to $10^5$ and tune all the learning rates to their optimal values. Here we set $a = 4.5$ and $b = 1$.

**Results.** Figure 2b plots $E(K, N)$ versus $K$ and $\log N$ for the solvable model with Zipf-distributed data. The curves depicting $E(K, N)$ versus $K$ show that $E(K, N) \approx K$ when $K$ is relatively small and saturate to some value depending on $N$ when $K$ is large. In the right panel, which describes the relationship between $E(K, N)$ and $\log N$, we observe that when $K$ is small (namely $K = 2$), $E(K, N)$ increases and approaches $K$ as $\log N$ increases, and the plots overlap when $K$ is large. Those phenomena provide empirical confirmation of the scaling behaviors predicted by Theorem 5.2. We also fit $E(K, N)$ in the large-$K$ regime with a power-form function as stated in Theorem 5.2. The fitted exponent is $0.279 \approx \frac{b}{a-b} = \frac{2}{7}$, aligning with our theory.

### 6.3 EMPIRICAL VALIDATION ON LARGE LANGUAGE MODELS

**Experiments Setup.** We conduct experiments on a large language model to empirically validate the hypothesis that larger datasets allow for more effective repetition. We perform pretraining runs

with fresh data sizes of 0.2B, 0.5B, 0.8B, 1.0B, and 2B tokens, each trained for 100 epochs. As a control, we also include a run with 200B fresh tokens. For each fresh dataset size $N$ and training epoch $K$, we approximate the effective reuse rate $E(K, N)$ by determining the effective fresh data size $N_f(K, N)$ required to achieve the same validation loss after one pass through the data. The effective reuse rate is then computed as: $E(K, N) = \frac{N_f(K,N)}{N}$.

Our experiments utilize a 0.3B parameter model adapted from the Qwen2.5-0.5B architecture (Qwen et al., 2025) and a subset of the DCLM dataset, totaling 200B tokens. A separate subset of the DCLM dataset is reserved for validation. Crucially, we use a constant learning rate schedule across all experiments to align with our theoretical analysis and mitigate the confounding effects of learning rate schedules, as reported in prior work (Hoffmann et al., 2022; Luo et al., 2025). Figure 1a depicts the relationship between $E(K, N)$ and $K$. Figure 1b depicts the training curves for different data sizes, and marks the points of different curves where $E(K, N) = \lambda K$, where $\lambda$ controls how strict the criterion is for determining when multi-epoch training begins to underperform one-pass training. Given such $\lambda$, we denote the corresponding number of training epochs as $K(\lambda, N)$, which we refer to as *saturation points*. In our experiments, we take $\lambda = 0.75$. Further, in Figure 1c, we show the precise relationship between $K(\lambda, N)$ and $N$. More details regarding the experiment setup are available in Appendix C.1.

**Previous Work: When $K \leq 4$, $E(K, N) \approx K$.** Our theoretical analysis indicates that $E(K, N)$ should be close to $K$ when $K$ is small (e.g., $K \leq 4$). In Figure 1a, when the epoch number is small (approximately $\leq 5$), we observe that $E(K, N)$ increases at a rate comparable to the epoch number, as indicated by the black dashed line. Thus our predictions of $E(K, N)$ when $K$ is small aligns with the data-constrained scaling laws (Muennighoff et al., 2023).

**Larger Datasets Allow More Repetition.** $E(K, N)$ increases with the number of fresh data sizes and eventually saturates for sufficiently large fresh datasets. Our results challenge the data-constrained scaling laws proposed by Muennighoff et al. (2023), which assume a uniform effective number of epochs across different fresh data sizes. In Figure 1b, we show that at the critical points where one-pass training start to outperform multi-epoch training significantly, $E(K, N)$ increases as $N$ increases. This suggests the continued potential for scaling pretraining through multi-epoch training with larger datasets.

**Fitting Experiments.** In Figure 1c, to provide real-world evidence that larger datasets can be repeated more, we plot the saturation point values for different $N$ to illustrate how they vary with $N$. Then we fit them as a function of $N$; see Appendix C.2 for details of the fitting procedure.

Surprisingly, though we do not claim that $E(K, N) = \Theta(\log N)$ holds for general LLM trainings when $K$ is large, as we calculated in the strongly convex linear regression case, here we do observe that $K(\lambda, N)$ gradually increases when $N$ increases, and it follows that $K(\lambda, N) \approx 0.80 \log N + 5.21$ with the correlation coefficient being $r = 0.97$. In this formula, the dataset $N$ is measured in billions of tokens (B).

**Experiments with Learning Rate Decay.** For further investigation of the scaling behaviour of multi-epoch training, we conduct LLM experiments with a non-constant learning rate schedule, aligning with the common practice in reality. Specifically, we additionally repeat the above analysis with a WSD learning rate schedule with linear decay. The experimental setup and results are described in Appendix C.3.

# 7 CONCLUSION

In this paper, we intrdouce the concept of effective reuse rate $E(K, N)$ and show both theoretically and empirically that larger datasets can be repeated more. More specifically, $E(K, N)$ saturates at later epochs as the dataset size increases. Several directions remain open for future study. (i) Our analysis is limited to the setting of optimal constant learning rates, and it remains to understand how learning rate decay, as well as the effort spent on tuning the learning rate, affect the effective reuse rate. (ii) Our theoretical analysis currently focuses on linear models, and a natural next step would be to extend the framework to more complex settings, such as neural networks with feature learning. (iii) We focus on reusing the entire dataset across multiple epochs, but to fully explore the potential of data reuse, one may consider more refined strategies such as selectively reusing or rephrasing high-quality data. (iv) Technically, our main results rely on strong convexity. Although we analyze a solvable case under a Zipf-law data distribution, generalizing these proof ideas to broader non-strongly convex settings remains an important theoretical challenge.

ACKNOWLEDGEMENTS

This work is supported by the National Natural Science Foundation of China under Grant Number 62495062. Binghui Li is supported by the Elite Ph.D. Program in Applied Mathematics at Peking University. We would like to thank Licong Lin, Huaqing Zhang, Jingfeng Wu, Kaiyue Wen, Jingzhao Zhang and Jason D. Lee for their insightful comments, and the anonymous reviewers for their valuable feedback.

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

CONTENTS

## A    THE USE OF LARGE LANGUAGE MODELS (LLMS)

In this paper, we use LLMs (mainly GPT-5 series) to polish some of the sections in our paper, and to check the grammatical issues. Besides that, we use LLMs to debug our code in LLM experiments (Section 6.3) and simulation experiments for Section 4 and Section 5. Also, LLMs are used to help improve the plotting scripts.

## B    ADDITIONAL RELATED WORKS

**Data Reuse in Synthetic Setting.**    Besides the real-world LLM pre-training regime, many works also reported the improvement of data reusing under synthetic settings empirically (Charton & Kempe, 2024; Kazdan et al., 2024) or theoretically (Zucchet et al., 2025; Dandi et al., 2024; Arnaboldi et al., 2025).

**Empirical Findings on Scaling Laws.**    Scaling laws reveal the relationships between large-scale model training loss and various factors such as model size, data size, and compute budget. These laws were initially observed by Hestness et al. (2017), but gained significant influence through the work of Kaplan et al. (2020), and have since been further developed in a series of studies (Henighan et al., 2020; Hoffmann et al., 2022; Zhai et al., 2022; Kadra et al., 2023; Aghajanyan et al., 2023; Muennighoff et al., 2023; Bi et al., 2024; Shuai et al., 2024; Kumar et al., 2024; Tissue et al., 2024; Luo et al., 2025). Notably, Muennighoff et al. (2023) further refined these models by incorporating the number of training epochs into a more complex scaling law, which empirically describes the effect of data reuse. In our work, we provide a theoretical analysis of how the effective reuse rate $E(K, N)$ relies on the epoch number $K$ and fresh data size $N$, highlighting the role of $N$ in the scaling behavior of $E(K, N)$, a factor that was overlooked in Muennighoff et al. (2023).

**Theoretical Explanations for Scaling Laws.**    A series of studies (Sharma & Kaplan, 2020; Hutter, 2021; Maloney et al., 2022; Wei et al., 2022; Jain et al., 2024; Michaud et al., 2024; Nam et al., 2024; Atanasov et al., 2024; Dohmatob et al., 2024; Bahri et al., 2024; Bordelon et al., 2024a; Lin et al., 2024; Paquette et al., 2025; Bordelon et al., 2024b; Zhang et al., 2024; Ferbach et al., 2025; Li et al., 2025a; 2026a;b; Wang et al., 2026) have sought to theoretically explain scaling laws from various perspectives. Among these, recent works (Bordelon et al., 2024a; Paquette et al., 2025; Lin et al., 2024; Bordelon et al., 2024b) have analyzed scaling laws by tracking the training dynamics of SGD through linear regression setup. Specifically, Bordelon et al. (2024a) investigated a full-batch gradient flow setup, while Paquette et al. (2025) and Bordelon et al. (2024b) focused on online SGD with a sufficiently small constant learning rate. Additionally, Lin et al. (2024) studied a geometric decaying learning rate schedule (LRS) (Ge et al., 2019; Wu et al., 2022a). Recently, Li et al. (2025a) proposed a functional scaling law that characterizes the loss dynamics for general LRSs. However, these scaling law studies did not account for the impact of data reuse. In contrast, our work examines the scaling behavior of multi-epoch SGD training within the context of a linear regression setup.

**SGD Analysis in Linear Regression.**    The analysis of SGD in linear regression has been extensively studied over the years, encompassing both one-pass and multi-epoch SGD. In the context of one-pass SGD, Zou et al. (2021); Meterez et al. (2025) considered an SGD procedure with a constant step size and averaged iterates, offering a sharp risk bound in terms of the eigenvalues of the covariance matrix. Gurbuzbalaban et al. (2021) examined one-pass SGD with batch size and proved that the distribution of the SGD iterates will converge to a heavy-tailed stationary distribution. Zou et al. (2022) compared the performance of SGD in the absence of ridge regression. Wu et al. (2022a) and Wu et al. (2022b) studied SGD in linear regression under covariate shift. Xia et al. (2024) considered SGD updates with noisy gradient and analyzed the perfect deleted point problem. Li & Gu (2025) considered SGD with exponential moving average in the linear regression setting. For multi-epoch SGD, Lin & Rosasco (2019) examined a scenario in which gradients are sampled uniformly at random and mini-batches are allowed. They analyzed the effects of mini-batch size, number of epochs, and learning rate, carefully combining these parameters to achieve the optimal convergence rate. Pillaud-Vivien et al. (2018) showed that while single-pass averaged SGD is optimal for a certain class of "easy" problems, multiple passes are required to achieve optimal prediction performance on a different class of "hard" problems, provided that an appropriate step size is chosen. In contrast to the matching upper and lower bounds derived by our theory, however, all the above works were only able to derive an upper bound for the loss.

## C  Additional Experimental Details for LLM Training

### C.1  Pretraining Setup

In our pretraining experiments, we employ the AdamW optimizer with a weight decay of 0.1 and a gradient clip of 1.0. We set the peak learning rate to 0.001, aligning with the approximate optimal learning rate reported by Li et al. (2025b). Balancing the optimal batch size suggested by Li et al. (2025b) with training efficiency, we utilize a sequence batch size of 128, which corresponds to roughly 0.5M data points per batch. We adopt the vocabulary of Qwen2.5 (Qwen et al., 2025) models. Our pretraining model consists of approximately 117 million non-embedding parameters, consistent with the methodology of Kaplan et al. (2020), and a total of 331 million parameters following the convention of Hoffmann et al. (2022). The detailed hyperparameter configurations are presented in Table 2, and the model architecture specifications are provided in Table 1. To ensure a fair comparison by eliminating the influence of batch order variations, we fix the random seed that governs the data stream across all experiments.

Table 1: Model configurations and parameter counts. $d_h$: hidden dimension; $d_f$: feed-forward dimension; $n_l$: number of Transformer layers; $n_h$: number of attention heads; $n_{kv}$: number of key-value heads (for grouped-query attention); Vocab Size: size of tokenizer vocabulary; #NE params: number of non-embedding parameters (in millions); #Params: total number of model parameters (in millions).

| Name | $d_h$ | $d_f$ | $n_l$ | $n_h$ | $n_{kv}$ | Vocab Size | #NE params | #Params |
|------|-------|-------|-------|-------|----------|------------|------------|---------|
| 0.5B | 896 | 4864 | 24 | 14 | 2 | 151936 | 355 | 491 |
| 0.3B | 640 | 3328 | 16 | 10 | 2 | 151936 | 117 | 331 |

Table 2: LLM Experiment Settings

| Parameter | Value |
|-----------|-------|
| **Data** | |
| Sequence Batch Size | 128 |
| Sequence Length | 4096 |
| **Learning Rate** | |
| Peak Learning Rate | 0.001 |
| Schedule | Constant |
| Warmup Steps | 400 |
| **Optimizer** | |
| Optimizer | AdamW |
| Weight Decay | 0.1 |
| $\beta_1$ | 0.9 |
| $\beta_2$ | 0.95 |
| $\epsilon$ | 1e-8 |
| Gradient Clip | 1.0 |

### C.2  Fitting Experiments

To provide real-world evidence that larger datasets can be repeated more, we show how the saturation points can be used to determine the appropriate number of training epochs. Recall that the saturation points are the points at which multi-epoch training first starts to underperform the one-pass baseline. We estimate these points from the pretraining loss curves presented in Section 6.3 and fit its dependence on $N$.

To estimate this quantity from the training curves, we proceed as follows. First, to reduce the impact of noise, we smooth the loss curves with exponential moving average (EMA) with decay coefficient $\alpha = 0.9$ and a window size of 3 checkpoints. Then for each dataset size $N$, we examine the ratio

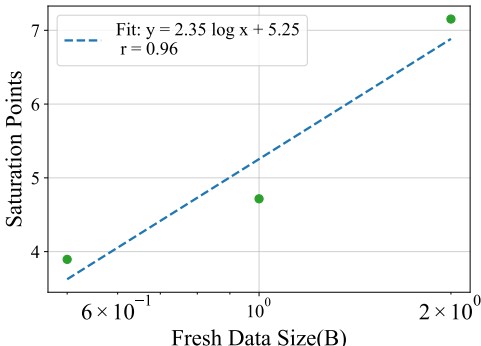

Figure 3: The saturation points $K(\lambda, N)$ as a function of the data size $N$ under a WSD learning rate schedule with linear decay.

$E(K, N)/K$. A larger ratio requires multi-epoch training to remain very close to the one-pass baseline, whereas a smaller ratio allows more deviation. Next, given a threshold hyperparameter $\lambda$, we identify the closest epoch $K$ at which this ratio first falls below $\lambda$, which we denote as $K(\lambda, N)$. Here we choose $\lambda = 0.75$, and we define $K(\lambda, N)$ as the saturation point.

We fit those points and find that $K(\lambda, N) \approx 0.80 \log N + 5.21$ with a correlation coefficient of $r = 0.97$. The fitting results are shown in Figure 1c.

### C.3 Experiments with WSD Learning Rate Schedule

Next, to make our LLM experiments more consistent with real-world pretraining practices, we repeat the LLM experiments under a warmup-stable-decay(WSD) learning rate schedule.

Concretely, we start from the checkpoints obtained in Section 6.3 for fresh data sizes $N \in \{0.2\text{B}, 0.5\text{B}, 1\text{B}, 2\text{B}\}$ after $K \in \{2, 4, 8, 16\}$ epochs of pretraining with a constant learning rate of $10^{-3}$. From each checkpoint, we continue training for one additional epoch while linearly decaying the learning rate from $10^{-3}$ to $10^{-5}$, resulting in a WSD learning rate schedule followed by a linear decay. For the one-pass baseline, we adopt the same schedule as in the $N = 2\text{B}$ run.

For each dataset size $N$, this process produces a set of four validation-loss values, each associated with one of the four selected epoch numbers $K$. We model the dependence of the final loss on the training steps $x$ using the parametric form $\ell(x) = A + \frac{B}{x^a}$, where $A, B, a$ are fitted parameters. The fitted curves are then used to predict the final validation loss under this WSD schedule for arbitrary training budgets. Using these predictions, we compute the saturation points following the same procedure as in Section 6.3. Here we still choose $\lambda = 0.75$.

The resulting saturation points are summarized in Figure 3. We observe that, even under this different learning rate schedule, the saturation points still satisfy the logarithmic scaling $K(\lambda, N) = \Theta(\log N)$. Specifically, we have $K(\lambda, N) \approx 2.35 \log N + 5.25$ with a correlation coefficient of $r = 0.96$. This confirms that our message that larger datasets can be repeated more also holds for real LLM training setups.

## D Additional Results and Simulations for Logarithmic Power-Law Spectrum

### D.1 Scaling Law for Logarithmic Power-Law Spectrum

We now present the scaling law for a logarithmic power-law spectrum. Its proof can be seen in Section I.4.

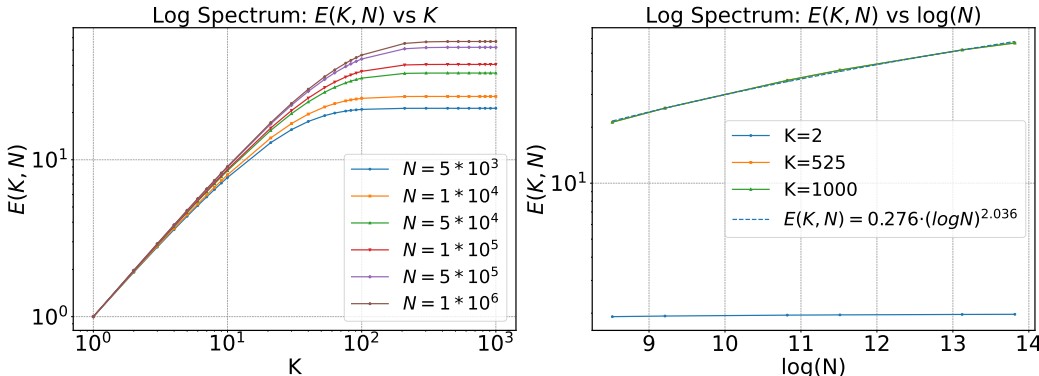

Figure 4: The solvable cases with logarithmic power-law spectrum. $E(K, N)$ exhibits a similar behavior to that presented in Figure 2. We also fit the effective reuse rate using the formula $E(K, N) = c_1 (\log N)^{c_2}$ suggested by Theorem 5.3, and the fitted exponent $c_2 = 2 \approx b = 2$ matches our theory.

**Theorem D.1.** *Consider a $K$-epoch SGD over $N$ fresh data. Under Assumptions 5.1, Assumption 5.3, and given the data dimension $d = \Omega((KN)^{\frac{1}{a}})$, it holds that*

$$\bar{\mathcal{R}}^*(K, N) \asymp \begin{cases} (KN)^{-\frac{a-1}{a}} & \text{for } K = o(\log^b N) \\ \left(N \log^b N\right)^{-\frac{a-1}{a}} & \text{for } K = \omega(\log^b N). \end{cases}$$

### D.2 SIMULATIONS IN SECTION 5.2

Now we focus on validating the predictions of Theorem 5.3 using synthetic data generated under the spectral assumptions of Section 5 and a log-power decay spectrum (Assumption 5.3).

**Experiments Setup.** Similar to Section 6.2, in all sub-figures of Figure 4, we set the data dimension $d$ to $10^5$ and tune all the learning rates to their optimal values. Here we set $a = 1.5$ and $b = 2$.

**Simulations for the Solvable Model.** Figure 4 plots $E(K, N)$ versus $K$ and $\log N$ for the solvable model. The curves depicting $E(K, N)$ versus $K$ and $E(K, N)$ versus $\log N$ show trends consistent with those in Section 6.2, aligning with Theorem 5.3. Furthermore, in the large-$K$ regime, we fit the exponent according to Theorem 5.3 and obtain $2.036 \approx b = 2$, which provides strong validation of our theory.

## E ADDITIONAL NOTATIONS

In this section, we provide some additional notations appeared in the following proof of our main results.

**Key Quantities.** We define the following key quantities to analyze the sequential updates. For each epoch $k$, let

$$\boldsymbol{A}^{(k)} := \prod_{i=N-1}^{0} \left(\boldsymbol{I} - \eta \boldsymbol{x}_{\pi_k(i)} \boldsymbol{x}_{\pi_k(i)}^\top\right) \tag{1}$$

represent the product of sequential updates through all samples in epoch $k$. More generally, we define the partial product operator:

$$\boldsymbol{Z}_{a \to b}^{(k)} := \prod_{i=a}^{b} (\boldsymbol{I} - \eta \boldsymbol{x}_{\pi_k(i)} \boldsymbol{x}_{\pi_k(i)}^\top), \quad \text{with} \quad \boldsymbol{A}^{(k)} = \boldsymbol{Z}_{N-1 \to 0}^{(k)}.$$

We further define that $\boldsymbol{Z}^{(k)}_{N-1\to N} = \boldsymbol{I}$. The cumulative effect across epochs is captured by:

$$\boldsymbol{T}^{(k)} := \prod_{i=K}^{k+1} \boldsymbol{A}^{(i)}, \quad \text{and} \quad \boldsymbol{T}^{(K)} = \boldsymbol{I}.$$

**Pseudo-expectation Notation $\tilde{\mathbb{E}}$.** Because matrix multiplication is non-commutative and the shuffling in training introduces statistical dependence, the expectations of the random matrices defined above cannot be written in a tractable closed form. To approximate the population excess risk, we therefore introduce the auxiliary notation $\tilde{\mathbb{E}}$. By construction, $\tilde{\mathbb{E}}$ computes the expectation of each factor as if the variables were independent, deliberately neglecting the correlations. We then invoke matrix-concentration inequalities to bound the gap between this "pseudo"-expectation and the true expectation of the original dependent random variables. Specifically, for the above random matrices used in our proof, here we further define that

$$\tilde{\mathbb{E}}\boldsymbol{Z}^{(k)}_{a\to b} := (\boldsymbol{I} - \eta\boldsymbol{H})^{a-b+1}, \tag{2}$$

$$\tilde{\mathbb{E}}\boldsymbol{A}^{(k)} := (\boldsymbol{I} - \eta\boldsymbol{H})^{N}, \tag{3}$$

$$\tilde{\mathbb{E}}\boldsymbol{T}^{(k)} := (\boldsymbol{I} - \eta\boldsymbol{H})^{N(K-k)}, \tag{4}$$

$$\tilde{\mathbb{E}}\boldsymbol{S}^{(ij)}_l := \tilde{\mathbb{E}}\Big[\boldsymbol{Z}^{(i)}_{N-1\to\pi_i^{-1}(l)+1}\Big]\,\mathbb{E}\big[\boldsymbol{x}_l\boldsymbol{x}_l^\top\big]\,\tilde{\mathbb{E}}\Big[\boldsymbol{Z}^{(j)}_{\pi_j^{-1}(l)+1\to N-1}\Big]. \tag{5}$$

## F    PROOF OUTLINE IN STRONGLY CONVEX LINEAR REGRESSION

In this section, we give the outline of Lemma G.1, Lemma 4.4, and Theorem 4.2. The main technical challenges and our proof insights are briefly stated in Section 4.2.

Section 4 centres on Theorem 4.2, which establishes a scaling law for the effective reuse rate $E(K, N)$ in terms of the relative magnitudes of number of epochs $K$ and dataset size $N$. Its proof unfolds in three stages.

**1. An explicit approximation of the expected excess risk.** Lemma G.1 derives a sufficiently accurate asymptotic formula for the expected excess risk of multi-epoch SGD. The argument begins with a bias–variance decomposition, splitting the expected excess risk into a variance term (Lemma G.2) and a bias term (Lemma G.3).

- **Variance term.** The closed-form approximation relies on concentration properties of matrix contractions together with a careful treatment of data shuffling.
- **Bias term.** The same contraction inequality is employed to obtain an analytic expression, after which tight error bounds are proved for the full range of relative sizes of $K$ and $N$. These bounds hold uniformly over a broad class of learning rates, necessitating detailed case-by-case analysis.

**2. Selection of a nearly optimal learning rate.** Lemma 4.4 identifies a learning rate whose resulting loss is asymptotically equivalent to the minimum excess risk attained with the optimal learning rate as stated in Section 3. This "approximately optimal learning rate" will be fixed in Appendix G.4.

**3. Proof of the effective reuse rate scaling law.** With the one-pass and multi-epoch SGD training learning rate set to the near-optimal learning rate obtained above, the proof of Theorem 4.2 proceeds to characterise the behaviour of $E(K, N)$ as $K$ and $N$ vary, yielding the desired scaling relation. Together, these three components establish Theorem 4.2 and provide a comprehensive description of how reuse efficiency depends on the interplay between $K$ and $N$.

## G    PROOF OF MAIN RESULTS IN STRONGLY CONVEX LINEAR REGRESSION

### G.1    STEP I: A CONCRETE VERSION OF BIAS-VARIANCE DECOMPOSITION

Before we begin our proof, we first present the following lemma, which provides the formal version of the loss estimate for a specific range of learning rate parameters. We define a $\widehat{\mathcal{R}}(K, N, \eta)$ as the

estimator of $\bar{\mathcal{R}}(K, N; \eta)$

$$\widehat{\mathcal{R}}(K, N; \eta) := \underbrace{\widehat{\mathcal{R}}_1(K, N; \eta)}_{\text{bias term}} + \underbrace{\widehat{\mathcal{R}}_2(K, N; \eta)}_{\text{var term across epochs}} + \underbrace{\widehat{\mathcal{R}}_3(K, N; \eta)}_{\text{var term within epoch}},$$

where

$$\widehat{\mathcal{R}}_1(K, N; \eta) := \frac{1}{2}(\boldsymbol{w}_0 - \boldsymbol{w}^*)^\top (\boldsymbol{I} - \eta \boldsymbol{H})^{2KN} \boldsymbol{H}(\boldsymbol{w}_0 - \boldsymbol{w}^*),$$

$$\widehat{\mathcal{R}}_2(K, N; \eta) := \frac{\sigma^2}{N} tr\left( \frac{(\boldsymbol{I} - (\boldsymbol{I} - \eta \boldsymbol{H})^{KN})((\boldsymbol{I} - \eta \boldsymbol{H})^N - (\boldsymbol{I} - \eta \boldsymbol{H})^{KN})}{\boldsymbol{I} + (\boldsymbol{I} - \eta \boldsymbol{H})^N} \right),$$

$$\widehat{\mathcal{R}}_3(K, N; \eta) := \frac{\eta \sigma^2}{2} \left\langle \boldsymbol{H}, (\boldsymbol{I} - (\boldsymbol{I} - \eta \boldsymbol{H})^{2KN})(2\boldsymbol{I} - \eta \boldsymbol{H})^{-1} \right\rangle.$$

### G.2 STEP II: RISK APPROXIMATION AND ERROR BOUND ANALYSIS

In this section, we rigorously formulate the analytic risk approximation in Lemma G.1 and provide its proof. Lemma G.1 indicates that the error bound is of higher order than the main term when the parameters are restricted to a limited range of values.

**Lemma G.1.** *Under Assumption 4.1 and 4.3, we further assume that for every $\boldsymbol{x}$ in the training set, $\|\boldsymbol{x}\| \le D$ for some constant $D > 0$. Consider a $K$-epoch SGD with learning rate $\eta \in \left[ \Omega\left(\frac{1}{T}\right), o(T^{-\frac{3}{4}}) \right]$, $K = o\left( \eta^{-1} T^{-\frac{3}{4}} \right)$ and data shuffling. Then, after $T = KN$ steps, the estimator of the expected excess risk satisfies:*

$$\bar{\mathcal{R}}(K, N; \eta) = \widehat{\mathcal{R}}(K, N; \eta) \left(1 + o(1)\right).$$

Recall from Section 4.2 that the risk $\bar{\mathcal{R}}(K, N; \eta)$ can be decomposed into the *bias term* $\bar{\mathcal{R}}^{\text{bias}}(K, N; \eta) := \frac{1}{2} \left\| \boldsymbol{\theta}_t^{\text{bias}} \right\|_{\boldsymbol{H}}^2$ and *variance term* $\bar{\mathcal{R}}^{\text{var}}(K, N; \eta) := \frac{1}{2} \left\| \boldsymbol{\theta}_t^{\text{var}} \right\|_{\boldsymbol{H}}^2$, which implies that Lemma G.1 is a direct corollary of the following two lemmas:

**Lemma G.2** (Variance Term). *Suppose that Assumption 4.1 holds. Then for a $K$-epoch SGD with dataset size $N$ and learning rate $\eta \in [\Omega(\frac{1}{T}), o(\frac{1}{T^{\frac{1}{2}}})]$ and shuffling, when $poly(T) \gtrsim d$, we have the estimator of the variance term $\bar{\mathcal{R}}^{\text{var}}(K, N; \eta) := \mathbb{E}_{\boldsymbol{w} \sim \mathcal{W}_{K,N,\eta}} [\mathcal{R}(\boldsymbol{w})^{\text{var}}]$ after $T := KN$ steps*

$$\tilde{\mathcal{R}}^{\text{var}}(K, N; \eta) := \frac{\sigma^2}{N} tr\left( \frac{(\boldsymbol{I} - (\boldsymbol{I} - \eta \boldsymbol{H})^{KN})((\boldsymbol{I} - \eta \boldsymbol{H})^N - (\boldsymbol{I} - \eta \boldsymbol{H})^{KN})}{\boldsymbol{I} + (\boldsymbol{I} - \eta \boldsymbol{H})^N} \right)$$
$$+ \frac{\eta \sigma^2}{2} \left\langle \boldsymbol{H}, (\boldsymbol{I} - (\boldsymbol{I} - \eta \boldsymbol{H})^{2KN})(2\boldsymbol{I} - \eta \boldsymbol{H})^{-1} \right\rangle,$$

*where the expectation is taken on the training set and shuffle, and the estimate error is*

$$\left| \tilde{\mathcal{R}}^{\text{var}}(K, N; \eta) - \bar{\mathcal{R}}^{\text{var}}(K, N; \eta) \right| = O(\eta^3 T^{\frac{3}{2}} K^2 \sqrt{\log d}).$$

*when $K \le \frac{\log 2}{\eta \sqrt{\tilde{C} 8 e D^4 T \log d}}$.*

**Lemma G.3** (Bias Term). *Under Assumption 4.1, for a $K$-epoch SGD with dataset size $N$, learning rate $\eta$ and shuffling, when $poly(T) \gtrsim d$, we have the estimator of the bias term $\bar{\mathcal{R}}^{\text{bias}}(K, N; \eta) := \mathbb{E}_{\boldsymbol{w} \sim \mathcal{W}_{K,N,\eta}} \left[ \mathcal{R}(\boldsymbol{w})^{\text{bias}} \right]$ after $T := KN$ steps*

$$\tilde{\mathcal{R}}^{\text{bias}}(K, N; \eta) := \frac{1}{2}(\boldsymbol{w}_0 - \boldsymbol{w}^*)^\top (\boldsymbol{I} - \eta \boldsymbol{H})^{2KN} \boldsymbol{H}(\boldsymbol{w}_0 - \boldsymbol{w}^*).$$

*Then we have the following estimate errors:*

*1. When $K \ge 2$ and $K = o\left( \frac{N^{\frac{1}{5}}}{(\log N)^{\frac{6}{5}}} \right)$:*

*(a) When $\eta \leq \frac{2\log T}{3\lambda_d T}$, the estimate distance is given by*

$$\left| \tilde{\mathcal{R}}^{\mathrm{bias}}(K, N; \eta) - \bar{\mathcal{R}}^{\mathrm{bias}}(K, N; \eta) \right| = O\left( (1 - \eta\lambda_d)^{N(2K-1)} K \sqrt{\eta^2 KN} \right).$$

*(b) When $\eta \geq \frac{2\log T}{3\lambda_d T}$, the estimate distance is given by*

$$\left| \tilde{\mathcal{R}}^{\mathrm{bias}}(K, N; \eta) - \bar{\mathcal{R}}^{\mathrm{bias}}(K, N; \eta) \right| = O\left( \frac{1}{T^{\frac{4}{3}}} \right).$$

2. *When $K = 1$:*

$$\left| \tilde{\mathcal{R}}^{\mathrm{bias}}(1, T; \eta) - \bar{\mathcal{R}}^{\mathrm{bias}}(1, T; \eta) \right| = O\left( \eta^2 T e^{-2\lambda_d \eta T} \right).$$

### G.2.1 VARIANCE TERM ANALYSIS: PROOF OF LEMMA G.2

We first recall some notations Appendix E that $\boldsymbol{Z}_{a \to b}^{(k)} = \prod_{i=a}^{b} (\boldsymbol{I} - \eta \boldsymbol{x}_{\pi_k(i)} \boldsymbol{x}_{\pi_k(i)}^{\top})$, $\boldsymbol{b}^{(k)} = \sum_{l=0}^{N-1} \boldsymbol{Z}_{N-1 \to l+1}^{(k)} \xi_{\pi_k(l)} \boldsymbol{x}_{\pi_k(l)}$, $\boldsymbol{A}^{(k)} = \boldsymbol{Z}_{N-1 \to 0}^{(k)}$, $\boldsymbol{T}^{(k)} = \prod_{i=K}^{k+1} \boldsymbol{A}^{(i)}$, and $\boldsymbol{T}^{(K)} = \boldsymbol{I}$. For simplicity, and if it does not cause confusion, we omit the superscript "var" in all the training parameters $\boldsymbol{\theta}^{\mathrm{var}}$ in the proof of Lemma G.2. Now we derive the recursion before and after the $k$-th epoch.

$$\begin{aligned}
\boldsymbol{\theta}_{kN} &= (\boldsymbol{I} - \eta \boldsymbol{x}_{\pi_k(N-1)} \boldsymbol{x}_{\pi_k(N-1)}^{\top}) \boldsymbol{\theta}_{kN-1} + \eta \xi_{\pi_k(N-1)} \boldsymbol{x}_{\pi_k(N-1)} \\
&= \eta \sum_{l=0}^{N-1} \boldsymbol{Z}_{N-1 \to l+1}^{(k)} \xi_{\pi_k(l)} \boldsymbol{x}_{\pi_k(l)} + \boldsymbol{A}^{(k)} \boldsymbol{\theta}_{(k-1)N} \\
&= \eta \boldsymbol{b}^{(k)} + \boldsymbol{A}^{(k)} \boldsymbol{\theta}_{(k-1)N},
\end{aligned}$$

where $\pi_k(i)$ is the $i$-th index after the permutation $\pi_k$ in the $K$-th epoch. Further writing out the above recursion gives the parameter after $K$ epochs

$$\boldsymbol{\theta}_{KN} = \eta \sum_{k=1}^{K} \boldsymbol{A}^{(K)} \cdots \boldsymbol{A}^{(k+1)} \boldsymbol{b}^{(k)}.$$

A natural move here is to replace $\boldsymbol{\theta}_{KN}$ with the expression above in the variance term

$$\begin{aligned}
\bar{\mathcal{R}}^{\mathrm{var}}(K, N; \eta) &= \mathbb{E} \frac{1}{2} \boldsymbol{\theta}_{KN}^{\top} \boldsymbol{H} \boldsymbol{\theta}_{KN} = \mathbb{E} \frac{1}{2} \left\langle \boldsymbol{H}, \boldsymbol{\theta}_{KN} \boldsymbol{\theta}_{KN}^{\top} \right\rangle \\
&= \frac{\eta^2}{2} \mathbb{E} \left\langle \boldsymbol{H}, \frac{1}{(N!)^K} \sum_{\pi_1 \cdots \pi_K} \sum_{i,j=1}^{K} \boldsymbol{T}^{(i)} \boldsymbol{b}^{(i)} \left( \boldsymbol{b}^{(j)} \right)^{\top} \left( \boldsymbol{T}^{(j)} \right)^{\top} \right\rangle \\
&= \frac{\eta^2 \sigma^2}{2} \mathbb{E} \left\langle \boldsymbol{H}, \frac{1}{(N!)^K} \sum_{\pi_1 \cdots \pi_K} \sum_{i,j=1}^{K} \boldsymbol{T}^{(i)} \left( \sum_{l=0}^{N-1} \boldsymbol{S}_l^{(ij)} \right) \left( \boldsymbol{T}^{(j)} \right)^{\top} \right\rangle \\
&= \frac{\eta^2 \sigma^2}{2} \mathbb{E} \left\langle \boldsymbol{H}, \frac{1}{(N!)^K} \sum_{\substack{i \neq j \\ i,j=1}}^{K} \sum_{\substack{\pi_1 \cdots \pi_K \\ \text{except } \pi_i, \pi_j}} \boldsymbol{T}^{(i)} \sum_{\pi_i, \pi_j} \left( \sum_{l=0}^{N-1} \boldsymbol{S}_l^{(ij)} \right) \left( \boldsymbol{T}^{(j)} \right)^{\top} \right\rangle \\
&\quad + \frac{\eta^2 \sigma^2}{2} \mathbb{E} \left\langle \boldsymbol{H}, \frac{1}{(N!)^K} \sum_{i=1}^{K} \sum_{\substack{\pi_1 \cdots \pi_K \\ \text{except } \pi_i}} \boldsymbol{T}^{(i)} \sum_{\pi_i} \left( \sum_{l=0}^{N-1} \boldsymbol{S}_l^{(ii)} \right) \left( \boldsymbol{T}^{(i)} \right)^{\top} \right\rangle. \quad (6)
\end{aligned}$$

where in the third equation, we take expectations with respect to the label noise $(\xi_l)_{l=0}^{N-1}$, and in the last equation, we decompose the variance term into two parts, according to whether the $\boldsymbol{b}^{(i)}$ and $\boldsymbol{b}^{(j)}$ are from the same epoch or not.

After explicitly writing the variance term, and to get a close-form formula for it, we then take pseudo expectations of $\boldsymbol{T}^{(i)}$, $\boldsymbol{T}^{(j)}$, $\boldsymbol{S}_l^{(ii)}$, and $\boldsymbol{S}_l^{(ij)}$ separately to get the approximation of $\bar{\mathcal{R}}^{\mathrm{var}}(K, N; \eta)$,

given as follows:

$$\tilde{\mathcal{R}}^{\mathrm{var}}(K,N;\eta) := \frac{\eta^2\sigma^2}{2}\mathbb{E}\left\langle \boldsymbol{H}, \frac{1}{(N!)^2}\sum_{\substack{i\neq j\\ i,j=1}}^{K}\tilde{\mathbb{E}}\boldsymbol{T}^{(i)}\left(\sum_{l=0}^{N-1}\sum_{\pi_i,\pi_j}\tilde{\mathbb{E}}\boldsymbol{S}_l^{(ij)}\right)\tilde{\mathbb{E}}\boldsymbol{T}^{(i)}\right\rangle$$

$$+\frac{\eta^2\sigma^2}{2}\mathbb{E}\left\langle \boldsymbol{H}, \frac{1}{N!}\sum_{i=1}^{K}\tilde{\mathbb{E}}\boldsymbol{T}^{(i)}\left(\sum_{l=0}^{N-1}\sum_{\pi_i}\tilde{\mathbb{E}}\boldsymbol{S}_l^{(ii)}\right)\tilde{\mathbb{E}}\boldsymbol{T}^{(i)}\right\rangle.$$

The intuition of the "pseudo expectation" and the related definitions are in Appendix E. Fix $l$, notice that when $i\neq j$, by Equation (5),

$$\sum_{\pi_i,\pi_j}\tilde{\mathbb{E}}\boldsymbol{S}_l^{(ij)} := \sum_{\pi_i,\pi_j}\tilde{\mathbb{E}}\left[\boldsymbol{Z}_{N-1\to\pi_i^{-1}(l)+1}^{(i)}\boldsymbol{x}_l\boldsymbol{x}_l^{\top}\boldsymbol{Z}_{\pi_j^{-1}(l)+1\to N-1}^{(j)}\right]$$

$$:= \sum_{\pi_i,\pi_j}(\boldsymbol{I}-\eta\boldsymbol{H})^{N-1-\pi_i^{-1}(l)}\boldsymbol{H}(\boldsymbol{I}-\eta\boldsymbol{H})^{N-1-\pi_j^{-1}(l)}.$$

For a fixed $i$, for all $m\in[0,N-1]$, there are $(N-1)!$ permutations $\pi_i$ that satisfies $\pi_i(m)=l$. So

$$\sum_{\pi_i,\pi_j}\tilde{\mathbb{E}}\boldsymbol{S}_l^{(ij)} = ((N-1)!)^2\sum_{m,n=0}^{N-1}(\boldsymbol{I}-\eta\boldsymbol{H})^{N-1-m}\boldsymbol{H}(\boldsymbol{I}-\eta\boldsymbol{H})^{N-1-n}. \qquad (7)$$

By applying a similar derivation to the $i=j$ case, we obtain that

$$\sum_{\pi_i}\tilde{\mathbb{E}}\boldsymbol{S}_l^{(ii)} = (N-1)!\sum_{m=0}^{N-1}(\boldsymbol{I}-\eta\boldsymbol{H})^{N-1-m}\boldsymbol{H}(\boldsymbol{I}-\eta\boldsymbol{H})^{N-1-m}. \qquad (8)$$

Plugging Equation (7) and Equation (8) into the expression of $\tilde{\mathcal{R}}^{\mathrm{var}}(K,N;\eta)$, and we have

$$\tilde{\mathcal{R}}^{\mathrm{var}}(K,N;\eta)$$

$$=\frac{\eta^2\sigma^2}{2}\mathbb{E}\left\langle \boldsymbol{H}, \frac{1}{N^2}\sum_{\substack{i\neq j\\ i,j=1}}^{K}\tilde{\mathbb{E}}\boldsymbol{T}^{(i)}\left(\sum_{l=0}^{N-1}\sum_{m,n=0}^{N-1}(\boldsymbol{I}-\eta\boldsymbol{H})^{2N-2-m-n}\boldsymbol{H}\right)\tilde{\mathbb{E}}\boldsymbol{T}^{(i)}\right\rangle$$

$$+\frac{\eta^2\sigma^2}{2}\mathbb{E}\left\langle \boldsymbol{H}, \frac{1}{N}\sum_{i=1}^{K}\tilde{\mathbb{E}}\boldsymbol{T}^{(i)}\left(\sum_{l=0}^{N-1}\sum_{m=0}^{N-1}(\boldsymbol{I}-\eta\boldsymbol{H})^{2N-2-2m}\boldsymbol{H}\right)\tilde{\mathbb{E}}\boldsymbol{T}^{(i)}\right\rangle$$

$$=\underbrace{\frac{\sigma^2}{2}\mathbb{E}\left\langle \boldsymbol{H}, \frac{1}{N}\sum_{\substack{i\neq j\\ i,j=1}}^{K}(\boldsymbol{I}-\eta\boldsymbol{H})^{N(K-i)}\left(\boldsymbol{I}-(\boldsymbol{I}-\eta\boldsymbol{H})^{N}\right)^2\boldsymbol{H}^{-1}(\boldsymbol{I}-\eta\boldsymbol{H})^{N(K-j)}\right\rangle}_{:=\Psi_1}$$

$$+\underbrace{\frac{\eta\sigma^2}{2}\mathbb{E}\left\langle \boldsymbol{H}, \sum_{i=1}^{K}(\boldsymbol{I}-\eta\boldsymbol{H})^{N(K-i)}\left(\boldsymbol{I}-(\boldsymbol{I}-\eta\boldsymbol{H})^{2N}\right)(2\boldsymbol{I}-\eta\boldsymbol{H})^{-1}(\boldsymbol{I}-\eta\boldsymbol{H})^{N(K-i)}\right\rangle}_{\Psi_2}.$$

where the second equation uses Equation (4). The quantity $\Psi_1$ accounts for the variance term across different epochs and $\Psi$ Then we calculate $\Psi_1$ and $\Psi_2$ separately. For $\Psi_1$, we have

$$\Psi_1 = \frac{\sigma^2}{2}\left\langle \boldsymbol{H}, \frac{1}{N}\sum_{i,j=1}^{K}(\boldsymbol{I}-\eta\boldsymbol{H})^{N(K-i)}\left(\boldsymbol{I}-(\boldsymbol{I}-\eta\boldsymbol{H})^{N}\right)^2\boldsymbol{H}^{-1}(\boldsymbol{I}-\eta\boldsymbol{H})^{N(K-j)}\right\rangle$$

$$-\frac{\sigma^2}{2}\left\langle \boldsymbol{H}, \frac{1}{N}\sum_{i=1}^{K}(\boldsymbol{I}-\eta\boldsymbol{H})^{N(K-i)}\left(\boldsymbol{I}-(\boldsymbol{I}-\eta\boldsymbol{H})^{N}\right)^2\boldsymbol{H}^{-1}(\boldsymbol{I}-\eta\boldsymbol{H})^{N(K-i)}\right\rangle$$

$$
= \frac{\sigma^2}{2} \left\langle \boldsymbol{H}, \frac{1}{N} \left( \left( \boldsymbol{I} - (\boldsymbol{I} - \eta\boldsymbol{H})^{KN} \right) \left( \boldsymbol{I} - (\boldsymbol{I} - \eta\boldsymbol{H})^{N} \right)^{-1} \right)^2 \boldsymbol{H}^{-1} \left( \boldsymbol{I} - (\boldsymbol{I} - \eta\boldsymbol{H})^{N} \right)^2 \right\rangle
$$

$$
- \frac{\sigma^2}{2N} \left\langle \boldsymbol{H}, \left( \boldsymbol{I} - (\boldsymbol{I} - \eta\boldsymbol{H})^{2KN} \right) \left( \boldsymbol{I} - (\boldsymbol{I} - \eta\boldsymbol{H})^{2N} \right)^{-1} \left( \left( \boldsymbol{I} - (\boldsymbol{I} - \eta\boldsymbol{H})^{N} \right)^2 \right) \boldsymbol{H}^{-1} \right\rangle
$$

$$
= \frac{\sigma^2}{2N} \mathrm{tr} \left( \left( \boldsymbol{I} - (\boldsymbol{I} - \eta\boldsymbol{H})^{KN} \right)^2 \right)
$$

$$
- \frac{\sigma^2}{2N} \mathrm{tr} \left( \left( \boldsymbol{I} - (\boldsymbol{I} - \eta\boldsymbol{H})^{N} \right)^2 \left( \boldsymbol{I} - (\boldsymbol{I} - \eta\boldsymbol{H})^{2N} \right)^{-1} \left( \boldsymbol{I} - (\boldsymbol{I} - \eta\boldsymbol{H})^{2KN} \right) \right)
$$

$$
= \frac{\sigma^2}{N} \mathrm{tr} \left( \left( \boldsymbol{I} - (\boldsymbol{I} - \eta\boldsymbol{H})^{KN} \right) \left( \boldsymbol{I} + (\boldsymbol{I} - \eta\boldsymbol{H})^{N} \right)^{-1} \left( (\boldsymbol{I} - \eta\boldsymbol{H})^{N} - (\boldsymbol{I} - \eta\boldsymbol{H})^{KN} \right) \right).
$$

The last equation is obtained by direct algebraic calculation. For $\Psi_2$, by direct matrix calculation, we get

$$
\Psi_2 = \frac{\eta\sigma^2}{2} \mathbb{E} \left\langle \boldsymbol{H}, (2\boldsymbol{I} - \eta\boldsymbol{H})^{-1} \left( \boldsymbol{I} - (\boldsymbol{I} - \eta\boldsymbol{H})^{2KN} \right) \right\rangle.
$$

Next we obtain the error bound for $\left| \tilde{\mathcal{R}}^{\mathrm{var}}(K, N; \eta) - \bar{\mathcal{R}}^{\mathrm{var}}(K, N; \eta) \right|$, which can be represented as

$$
\left| \tilde{\mathcal{R}}^{\mathrm{var}}(K, N; \eta) - \bar{\mathcal{R}}^{\mathrm{var}}(K, N; \eta) \right|
$$

$$
\leq \left| \frac{\eta^2\sigma^2}{2} \mathbb{E} \left\langle \boldsymbol{H}, \frac{1}{(N!)^K} \sum_{\substack{i \neq j \\ i,j=1}}^{K} \sum_{\substack{\pi_1 \cdots \pi_K \\ \text{except } \pi_i, \pi_j}} \boldsymbol{T}^{(i)} \sum_{\pi_i, \pi_j} \left( \sum_{l=0}^{N-1} \boldsymbol{S}_l^{(ij)} \right) \left( \boldsymbol{T}^{(j)} \right)^{\top} \right\rangle \right.
$$

$$
\left. - \frac{\eta^2\sigma^2}{2} \mathbb{E} \left\langle \boldsymbol{H}, \frac{1}{(N!)^2} \sum_{\substack{i \neq j \\ i,j=1}}^{K} (\boldsymbol{I} - \eta\boldsymbol{H})^{N(K-i)} \left( \sum_{l=0}^{N-1} \sum_{\pi_i, \pi_j} \tilde{\mathbb{E}} \boldsymbol{S}_l^{(ij)} \right) (\boldsymbol{I} - \eta\boldsymbol{H})^{N(K-j)} \right\rangle \right| =: I_1
$$

$$
+ \left| \frac{\eta^2\sigma^2}{2} \mathbb{E} \left\langle \boldsymbol{H}, \frac{1}{(N!)^K} \sum_{i=1}^{K} \sum_{\substack{\pi_1 \cdots \pi_K \\ \text{except } \pi_i}} \boldsymbol{T}^{(i)} \sum_{\pi_i} \left( \sum_{l=0}^{N-1} \boldsymbol{S}_l^{(ii)} \right) \left( \boldsymbol{T}^{(i)} \right)^{\top} \right\rangle \right.
$$

$$
\left. - \frac{\eta^2\sigma^2}{2} \mathbb{E} \left\langle \boldsymbol{H}, \frac{1}{N!} \sum_{i=1}^{K} (\boldsymbol{I} - \eta\boldsymbol{H})^{N(K-i)} \left( \sum_{l=0}^{N-1} \sum_{\pi_i} \tilde{\mathbb{E}} \boldsymbol{S}_l^{(ii)} \right) (\boldsymbol{I} - \eta\boldsymbol{H})^{N(K-i)} \right\rangle \right| =: I_2,
$$

where the first inequality uses the triangle inequality. The term $I_1$ represents the error term between epochs, and $I_2$ represents the error term within one epoch. We will bound $I_1$ and $I_2$ separately in the proof.

**Upper bound for $I_1$.** To bound $I_1$, a natural move here is to plug in a term that takes pseudo expectation over $(\boldsymbol{T}^{(i)})_{i=1}^{K}$ but does not take pseudo expectation over $(\boldsymbol{S}_l^{(ij)})_{l,i,j}$, and divide $I_1$ into two terms.

$$
I_1 \leq \left| \frac{\eta^2\sigma^2}{2} \mathbb{E} \left\langle \boldsymbol{H}, \frac{1}{(N!)^K} \sum_{\substack{i \neq j \\ i,j=1}}^{K} \sum_{\substack{\pi_1 \cdots \pi_K \\ \text{except } \pi_i, \pi_j}} \boldsymbol{T}^{(i)} \sum_{\pi_i, \pi_j} \left( \sum_{l=0}^{N-1} \boldsymbol{S}_l^{(ij)} \right) \left( \boldsymbol{T}^{(j)} \right)^{\top} \right\rangle \right.
$$

$$
\left. - \frac{\eta^2\sigma^2}{2} \mathbb{E} \left\langle \boldsymbol{H}, \frac{1}{(N!)^2} \sum_{\substack{i \neq j \\ i,j=1}}^{K} (\boldsymbol{I} - \eta\boldsymbol{H})^{N(K-i)} \sum_{\pi_i, \pi_j} \left( \sum_{l=0}^{N-1} \boldsymbol{S}_l^{(ij)} \right) (\boldsymbol{I} - \eta\boldsymbol{H})^{N(K-j)} \right\rangle \right|
$$

$$
+ \left| \frac{\eta^2\sigma^2}{2} \mathbb{E} \left\langle \boldsymbol{H}, \frac{1}{(N!)^2} \sum_{\substack{i \neq j \\ i,j=1}}^{K} (\boldsymbol{I} - \eta\boldsymbol{H})^{N(K-i)} \sum_{\pi_i, \pi_j} \left( \sum_{l=0}^{N-1} \boldsymbol{S}_l^{(ij)} \right) (\boldsymbol{I} - \eta\boldsymbol{H})^{N(K-j)} \right\rangle \right.
$$

$$- \frac{\eta^2 \sigma^2}{2} \mathbb{E} \left\langle \boldsymbol{H}, \frac{1}{(N!)^2} \sum_{\substack{i \neq j \\ i,j=1}}^{K} (\boldsymbol{I} - \eta \boldsymbol{H})^{N(K-i)} \left( \sum_{l=0}^{N-1} \sum_{\pi_i, \pi_j} \tilde{\mathbb{E}} \boldsymbol{S}_l^{(ij)} \right) (\boldsymbol{I} - \eta \boldsymbol{H})^{N(K-j)} \right\rangle \Bigg|$$

$$=: I_{11} + I_{12}.$$

Next we bound the terms $I_{11}$ and $I_{12}$ separately. Notice that

$$\sum_{\substack{i \neq j \\ i,j=1}}^{K} \sum_{\substack{\pi_1 \cdots \pi_K \\ \text{except } \pi_i, \pi_j}} (\boldsymbol{I} - \eta \boldsymbol{H})^{N(K-i)} \sum_{\pi_i, \pi_j} \left( \sum_{l=0}^{N-1} \boldsymbol{S}_l^{(ij)} \right) (\boldsymbol{I} - \eta \boldsymbol{H})^{N(K-j)}$$

$$= (N!)^{K-2} \sum_{\substack{i \neq j \\ i,j=1}}^{K} (\boldsymbol{I} - \eta \boldsymbol{H})^{N(K-i)} \sum_{\pi_i, \pi_j} \left( \sum_{l=0}^{N-1} \boldsymbol{S}_l^{(ij)} \right) (\boldsymbol{I} - \eta \boldsymbol{H})^{N(K-j)} \tag{9}$$

because the summands do not depend on the permutations except $\pi_i, \pi_j$, plugging Equation (9) into the expression of $I_1$ we have

$$I_{11} \leq \left| \frac{\eta^2 \sigma^2}{2} \mathbb{E} \left\langle \boldsymbol{H}, \frac{1}{(N!)^K} \sum_{\substack{i \neq j \\ i,j=1}}^{K} \sum_{\substack{\pi_1 \cdots \pi_K \\ \text{except } \pi_i, \pi_j}} \boldsymbol{T}^{(i)} \sum_{\pi_i, \pi_j} \left( \sum_{l=0}^{N-1} \boldsymbol{S}_l^{(ij)} \right) \left( \boldsymbol{T}^{(j)} \right)^\top \right\rangle \right.$$

$$\left. - \frac{\eta^2 \sigma^2}{2} \mathbb{E} \left\langle \boldsymbol{H}, \frac{1}{(N!)^K} \sum_{\substack{i \neq j \\ i,j=1}}^{K} \sum_{\substack{\pi_1 \cdots \pi_K \\ \text{except } \pi_i, \pi_j}} (\boldsymbol{I} - \eta \boldsymbol{H})^{N(K-i)} \sum_{\pi_i, \pi_j} \left( \sum_{l=0}^{N-1} \boldsymbol{S}_l^{(ij)} \right) (\boldsymbol{I} - \eta \boldsymbol{H})^{N(K-j)} \right\rangle \right|.$$

Then we use Equation (4) to split $I_{11}$ into three terms and by triangle inequality:

$$I_{11} \leq \left| \frac{\eta^2 \sigma^2}{2} \mathbb{E} \left\langle \boldsymbol{H}, \frac{1}{(N!)^K} \sum_{\substack{i \neq j \\ i,j=1}}^{K} \sum_{\substack{\pi_1 \cdots \pi_K \\ \text{except } \pi_i, \pi_j}} \left( \boldsymbol{T}^{(i)} - \tilde{\mathbb{E}} \boldsymbol{T}^{(i)} \right) \sum_{\pi_i, \pi_j} \left( \sum_{l=0}^{N-1} \boldsymbol{S}_l^{(ij)} \right) \tilde{\mathbb{E}} \boldsymbol{T}^{(i)} \right\rangle \right|$$

$$+ \left| \frac{\eta^2 \sigma^2}{2} \mathbb{E} \left\langle \boldsymbol{H}, \frac{1}{(N!)^K} \sum_{\substack{i \neq j \\ i,j=1}}^{K} \sum_{\substack{\pi_1 \cdots \pi_K \\ \text{except } \pi_i, \pi_j}} \tilde{\mathbb{E}} \boldsymbol{T}^{(i)} \sum_{\pi_i, \pi_j} \left( \sum_{l=0}^{N-1} \boldsymbol{S}_l^{(ij)} \right) \left( \boldsymbol{T}^{(j)} - \tilde{\mathbb{E}} \boldsymbol{T}^{(j)} \right) \right\rangle \right|$$

$$+ \left| \frac{\eta^2 \sigma^2}{2} \mathbb{E} \left\langle \boldsymbol{H}, \frac{1}{(N!)^K} \sum_{\substack{i \neq j \\ i,j=1}}^{K} \sum_{\substack{\pi_1 \cdots \pi_K \\ \text{except } \pi_i, \pi_j}} \left( \boldsymbol{T}^{(i)} - \tilde{\mathbb{E}} \boldsymbol{T}^{(i)} \right) \sum_{\pi_i, \pi_j} \left( \sum_{l=0}^{N-1} \boldsymbol{S}_l^{(ij)} \right) \left( \boldsymbol{T}^{(j)} - \tilde{\mathbb{E}} \boldsymbol{T}^{(j)} \right) \right\rangle \right|.$$

Next, we use Lemma J.1 and the fact that $\boldsymbol{S}_l^{(ij)} \lesssim \boldsymbol{I}$ to bound the matrix inner products:

$$I_{11} \leq \frac{\eta^2 \sigma^2 N D^2 \text{tr}(\boldsymbol{H})}{2(N!)^{K-2}} \sum_{\substack{i \neq j \\ i,j=1}}^{K} \sum_{\substack{\pi_1 \cdots \pi_K \\ \text{except } \pi_i, \pi_j}} \left( \mathbb{E} \left\| \boldsymbol{T}^{(i)} - \tilde{\mathbb{E}} \boldsymbol{T}^{(i)} \right\| + \mathbb{E} \left\| \boldsymbol{T}^{(j)} - \tilde{\mathbb{E}} \boldsymbol{T}^{(j)} \right\| \right.$$

$$\left. + \mathbb{E} \left\| \boldsymbol{T}^{(i)} - \tilde{\mathbb{E}} \boldsymbol{T}^{(i)} \right\| \left\| \boldsymbol{T}^{(j)} - \tilde{\mathbb{E}} \boldsymbol{T}^{(j)} \right\| \right).$$

Notice that Lemma J.2 and Lemma J.5 implies that

$$\mathbb{E}\left\|\boldsymbol{T}^{(i)} - \tilde{\mathbb{E}}\boldsymbol{T}^{(i)}\right\| \leq (\sqrt{\delta_{\mathrm{A}}\eta^2 NK} + \|\mathbb{E}\boldsymbol{A}\|)^K - \|\mathbb{E}\boldsymbol{A}\|^K$$

$$\leq (\sqrt{\delta_{\mathrm{A}}\eta^2 NK} + 1)^K - 1$$

$$\leq 2K\sqrt{\delta_{\mathrm{A}}\eta^2 NK} \quad \text{when} \quad K \leq \frac{\log 2}{\eta\sqrt{\delta_{\mathrm{A}}T}},$$

where $\delta_{\mathrm{A}} = \tilde{C}8eD^4 \log d$ is the constant appeared in Lemma J.4, and $\tilde{C}$ is some absolute constant. The second inequality uses the fact that $(\sqrt{\delta_{\mathrm{A}}\eta^2 NK} + \|\mathbb{E}\boldsymbol{A}\|)^K - \|\mathbb{E}\boldsymbol{A}\|^K$ motonously increases with $\|\mathbb{E}\boldsymbol{A}\|$. A similar approach combining Lemma J.2 and Lemma J.6 derives another concentration inequality for $\boldsymbol{T}^{(i)}$:

$$\mathbb{E}\left\|\boldsymbol{T}^{(i)} - \tilde{\mathbb{E}}\boldsymbol{T}^{(i)}\right\|^2 \leq \left(2K\sqrt{\delta_{\mathrm{A}}\eta^2 NK}\right)^2 \quad \text{when} \quad K \leq \frac{\log 2}{\eta\sqrt{\delta_{\mathrm{A}}T}}.$$

Applying Cauchy-Schwarz's inequality and the concentration inequalities for $\left(\boldsymbol{T}^{(i)}\right)_i$, we get that

$$I_{11} \leq \frac{\eta^2\sigma^2 ND^2\mathrm{tr}(\boldsymbol{H})}{2(N!)^{K-2}} \sum_{\substack{i\neq j \\ i,j=1}}^K \sum_{\substack{\pi_1\cdots\pi_K \\ \text{except } \pi_i,\pi_j}} \left( \mathbb{E}\left\|\boldsymbol{T}^{(i)} - \tilde{\mathbb{E}}\boldsymbol{T}^{(i)}\right\| + \mathbb{E}\left\|\boldsymbol{T}^{(j)} - \tilde{\mathbb{E}}\boldsymbol{T}^{(j)}\right\| \right.$$

$$\left. + \left( \mathbb{E}\left\|\boldsymbol{T}^{(i)} - \tilde{\mathbb{E}}\boldsymbol{T}^{(i)}\right\|^2 \right)^{\frac{1}{2}} \left( \mathbb{E}\left\|\boldsymbol{T}^{(j)} - \tilde{\mathbb{E}}\boldsymbol{T}^{(j)}\right\|^2 \right)^{\frac{1}{2}} \right)$$

$$\leq \frac{\eta^2\sigma^2 ND^2\mathrm{tr}(\boldsymbol{H})}{2} \sum_{\substack{i\neq j \\ i,j=1}}^K \left( 4K\sqrt{\delta_{\mathrm{A}}\eta^2 NK} + \left(2K\sqrt{2\delta_{\mathrm{A}}\eta^2 NK}\right)^2 \right).$$

Our next step is to bound $I_{12}$. We first make use of the fact that $\boldsymbol{I} - \eta\boldsymbol{H} \lesssim \boldsymbol{I}$, and get that

$$I_{12} \leq \left| \frac{\eta^2\sigma^2}{2}\mathbb{E}\left\langle \boldsymbol{H}, \frac{1}{(N!)^2} \sum_{\substack{i\neq j \\ i,j=1}}^K \sum_{\pi_i,\pi_j} \left( \sum_{l=0}^{N-1} \boldsymbol{S}_l^{(ij)} - \tilde{\mathbb{E}}\boldsymbol{S}_l^{(ij)} \right) \right\rangle \right|.$$

Recall that for a fixed $i$, for all $m \in [0, N-1]$, there are $(N-1)!$ permutations $\pi_i$ that satisfies $\pi_i(m) = l$. So

$$I_{12} \leq \left| \frac{\eta^2\sigma^2}{2}\mathbb{E}\left\langle \boldsymbol{H}, \frac{1}{(N!)^2} \sum_{\substack{i\neq j \\ i,j=1}}^K \sum_{l=0}^{N-1}\sum_{m=0}^{N-1}\sum_{n=0}^{N-1} ((N-1)!)^2 \left( \boldsymbol{Z}_{N-1\to m+1}^{(i)} \boldsymbol{H}\boldsymbol{Z}_{n+1\to N-1}^{(j)} \right. \right. \right.$$

$$\left. \left. \left. - \mathbb{E}\boldsymbol{Z}_{N-1\to m+1}^{(i)} \boldsymbol{H}\mathbb{E}\boldsymbol{Z}_{n+1\to N-1}^{(j)} \right) \right\rangle \right|.$$

Notice that

$$\boldsymbol{Z}_{N-1\to m+1}^{(i)} \boldsymbol{H}\boldsymbol{Z}_{n+1\to N-1}^{(j)} - \mathbb{E}\boldsymbol{Z}_{N-1\to m+1}^{(i)} \boldsymbol{H}\mathbb{E}\boldsymbol{Z}_{n+1\to N-1}^{(j)}$$

$$= \left( \boldsymbol{Z}_{N-1\to m+1}^{(i)} - \mathbb{E}\boldsymbol{Z}_{N-1\to m+1}^{(i)} \right) \boldsymbol{H}\mathbb{E}\boldsymbol{Z}_{n+1\to N-1}^{(j)} + \mathbb{E}\boldsymbol{Z}_{N-1\to m+1}^{(i)} \boldsymbol{H} \left( \boldsymbol{Z}_{n+1\to N-1}^{(j)} - \mathbb{E}\boldsymbol{Z}_{n+1\to N-1}^{(j)} \right)$$

$$+ \left( \boldsymbol{Z}_{N-1\to m+1}^{(i)} - \mathbb{E}\boldsymbol{Z}_{N-1\to m+1}^{(i)} \right) \boldsymbol{H} \left( \boldsymbol{Z}_{n+1\to N-1}^{(j)} - \mathbb{E}\boldsymbol{Z}_{n+1\to N-1}^{(j)} \right).$$

Applying Lemma J.1 and using the fact that $\mathbb{E}\boldsymbol{Z}_{N-1\rightarrow m+1}^{(i)} \lesssim \boldsymbol{I}$,

$$I_{12} \leq \frac{\eta^2\sigma^2\mathrm{tr}(\boldsymbol{H})\|\boldsymbol{H}\|N}{2N^2}\mathbb{E}\sum_{\substack{i\neq j\\i,j=1}}^{K}\left(\sum_{m=0}^{N-2}\left\|\boldsymbol{Z}_{N-1\rightarrow m+1}^{(i)} - \mathbb{E}\boldsymbol{Z}_{N-1\rightarrow m+1}^{(i)}\right\|\right.$$

$$+ \sum_{n=0}^{N-2}\left\|\boldsymbol{Z}_{n+1\rightarrow N-1}^{(j)} - \mathbb{E}\boldsymbol{Z}_{n+1\rightarrow N-1}^{(j)}\right\|$$

$$\left.+ \sum_{m=0}^{N-2}\sum_{n=0}^{N-2}\left\|\boldsymbol{Z}_{N-1\rightarrow m+1}^{(i)} - \mathbb{E}\boldsymbol{Z}_{N-1\rightarrow m+1}^{(i)}\right\|\left\|\boldsymbol{Z}_{n+1\rightarrow N-1}^{(j)} - \mathbb{E}\boldsymbol{Z}_{n+1\rightarrow N-1}^{(j)}\right\|\right).$$

Applying Cauchy-Schwarz inequality and Lemma J.4 gives

$$I_{12} \leq \frac{\eta^2\sigma^2\mathrm{tr}(\boldsymbol{H})\|\boldsymbol{H}\|N}{2N^2}\sum_{\substack{i\neq j\\i,j=1}}^{K}\left(\sum_{m=0}^{N-2}\mathbb{E}\left\|\boldsymbol{Z}_{N-1\rightarrow m+1}^{(i)} - \mathbb{E}\boldsymbol{Z}_{N-1\rightarrow m+1}^{(i)}\right\|\right.$$

$$+ \sum_{n=0}^{N-2}\mathbb{E}\left\|\boldsymbol{Z}_{n+1\rightarrow N-1}^{(j)} - \mathbb{E}\boldsymbol{Z}_{n+1\rightarrow N-1}^{(j)}\right\|$$

$$\left.+ \sum_{m=0}^{N-2}\sum_{n=0}^{N-2}\left(\mathbb{E}\left\|\boldsymbol{Z}_{N-1\rightarrow m+1}^{(i)} - \mathbb{E}\boldsymbol{Z}_{N-1\rightarrow m+1}^{(i)}\right\|^2\right)^{\frac{1}{2}}\left(\mathbb{E}\left\|\boldsymbol{Z}_{n+1\rightarrow N-1}^{(j)} - \mathbb{E}\boldsymbol{Z}_{n+1\rightarrow N-1}^{(j)}\right\|^2\right)^{\frac{1}{2}}\right)$$

$$\leq \frac{\eta^2\sigma^2\mathrm{tr}(\boldsymbol{H})\|\boldsymbol{H}\|N}{2N^2}\sum_{\substack{i\neq j\\i,j=1}}^{K}\left(\sum_{m=0}^{N-2}\left(\sqrt{\delta_{\mathrm{A}}\eta^2(N-1-m)}\right) + \sum_{n=0}^{N-2}\left(\sqrt{\delta_{\mathrm{A}}\eta^2(N-1-n)}\right)\right.$$

$$\left.+ \sum_{m=0}^{N-2}\sum_{n=0}^{N-2}\left(\sqrt{2\delta_{\mathrm{A}}\eta^2(N-1-m)}\right)\left(\sqrt{2\delta_{\mathrm{A}}\eta^2(N-1-n)}\right)\right)$$

$$\lesssim \eta^3K^2\sqrt{N\log d} + \eta^4K^2N^2\log d \quad\text{when}\quad \eta = o(\frac{1}{\sqrt{T}}).$$

**Upper bound for $I_2$.** We bound $I_2$ using a similar technique as what we did for $I_1$. We first plug in a term that takes pseudo expectation over $(\boldsymbol{T}^{(i)})_{i=1}^{K}$ but does not take pseudo expectation over $\boldsymbol{S}_l^{(ii)}$ for every $l$ and $i$, and decompose $I_2$ into two terms:

$$I_2 \leq \left|\frac{\eta^2\sigma^2}{2}\mathbb{E}\left\langle\boldsymbol{H}, \frac{1}{(N!)^K}\sum_{i=1}^{K}\sum_{\substack{\pi_1\cdots\pi_K\\\text{except }\pi_i}}\boldsymbol{T}^{(i)}\sum_{\pi_i}\left(\sum_{l=0}^{N-1}\boldsymbol{S}_l^{(ii)}\right)\left(\boldsymbol{T}^{(i)}\right)^{\top}\right\rangle\right.$$

$$\left.- \frac{\eta^2\sigma^2}{2}\mathbb{E}\left\langle\boldsymbol{H}, \frac{1}{N!}\sum_{i=1}^{K}(\boldsymbol{I}-\eta\boldsymbol{H})^{N(K-i)}\sum_{\pi_i}\left(\sum_{l=0}^{N-1}\boldsymbol{S}_l^{(ii)}\right)(\boldsymbol{I}-\eta\boldsymbol{H})^{N(K-i)}\right\rangle\right|$$

$$+ \left|\frac{\eta^2\sigma^2}{2}\mathbb{E}\left\langle\boldsymbol{H}, \frac{1}{N!}\sum_{i=1}^{K}(\boldsymbol{I}-\eta\boldsymbol{H})^{N(K-i)}\sum_{\pi_i}\left(\sum_{l=0}^{N-1}\boldsymbol{S}_l^{(ii)}\right)(\boldsymbol{I}-\eta\boldsymbol{H})^{N(K-i)}\right\rangle\right.$$

$$\left.- \frac{\eta^2\sigma^2}{2}\mathbb{E}\left\langle\boldsymbol{H}, \frac{1}{N!}\sum_{i=1}^{K}(\boldsymbol{I}-\eta\boldsymbol{H})^{N(K-i)}\left(\sum_{l=0}^{N-1}\sum_{\pi_i}\tilde{\mathbb{E}}\boldsymbol{S}_l^{(ii)}\right)(\boldsymbol{I}-\eta\boldsymbol{H})^{N(K-i)}\right\rangle\right|$$

$$=: I_{21} + I_{22}.$$

Next we bound the terms $I_{21}$ and $I_{22}$ separately. Notice that

$$\sum_{i=1}^{K}\sum_{\substack{\pi_1\cdots\pi_K\\\text{except }\pi_i}}(\boldsymbol{I}-\eta\boldsymbol{H})^{N(K-i)}\sum_{\pi_i}\left(\sum_{l=0}^{N-1}\boldsymbol{S}_l^{(ii)}\right)(\boldsymbol{I}-\eta\boldsymbol{H})^{N(K-i)}$$

$$= (N!)^{K-1} \sum_{i=1}^{K} (\boldsymbol{I} - \eta\boldsymbol{H})^{N(K-i)} \sum_{\pi_i} \left( \sum_{l=0}^{N-1} \boldsymbol{S}_l^{(ii)} \right) (\boldsymbol{I} - \eta\boldsymbol{H})^{N(K-i)}$$

because the summands do not depend on the permutations except $\pi_i$, we have

$$I_{21} = \left| \frac{\eta^2\sigma^2}{2} \mathbb{E} \left\langle \boldsymbol{H}, \frac{1}{(N!)^K} \sum_{i=1}^{K} \sum_{\substack{\pi_1 \cdots \pi_K \\ \text{except } \pi_i}} \boldsymbol{T}^{(i)} \sum_{\pi_i} \left( \sum_{l=0}^{N-1} \boldsymbol{S}_l^{(ii)} \right) \left( \boldsymbol{T}^{(i)} \right)^{\top} \right\rangle \right.$$

$$\left. - \frac{\eta^2\sigma^2}{2} \mathbb{E} \left\langle \boldsymbol{H}, \frac{1}{(N!)^K} \sum_{i=1}^{K} \sum_{\substack{\pi_1 \cdots \pi_K \\ \text{except } \pi_i}} (\boldsymbol{I} - \eta\boldsymbol{H})^{N(K-i)} \sum_{\pi_i} \left( \sum_{l=0}^{N-1} \boldsymbol{S}_l^{(ii)} \right) (\boldsymbol{I} - \eta\boldsymbol{H})^{N(K-i)} \right\rangle \right|.$$

Then we use the fact that $\tilde{\mathbb{E}}\boldsymbol{T}^{(i)} = (\boldsymbol{I} - \eta\boldsymbol{H})^{N(K-i)}$ to split $I_{21}$ into three terms:

$$I_{21} \le \left| \frac{\eta^2\sigma^2}{2} \mathbb{E} \left\langle \boldsymbol{H}, \frac{1}{(N!)^K} \sum_{i=1}^{K} \sum_{\substack{\pi_1 \cdots \pi_K \\ \text{except } \pi_i}} \left( \boldsymbol{T}^{(i)} - \tilde{\mathbb{E}}\boldsymbol{T}^{(i)} \right) \sum_{\pi_i} \left( \sum_{l=0}^{N-1} \boldsymbol{S}_l^{(ii)} \right) (\boldsymbol{I} - \eta\boldsymbol{H})^{N(K-i)} \right\rangle \right|$$

$$+ \left| \frac{\eta^2\sigma^2}{2} \mathbb{E} \left\langle \boldsymbol{H}, \frac{1}{(N!)^K} \sum_{i=1}^{K} \sum_{\substack{\pi_1 \cdots \pi_K \\ \text{except } \pi_i}} (\boldsymbol{I} - \eta\boldsymbol{H})^{N(K-i)} \sum_{\pi_i} \left( \sum_{l=0}^{N-1} \boldsymbol{S}_l^{(ii)} \right) \left( \boldsymbol{T}^{(i)} - \tilde{\mathbb{E}}\boldsymbol{T}^{(i)} \right) \right\rangle \right|$$

$$+ \left| \frac{\eta^2\sigma^2}{2} \mathbb{E} \left\langle \boldsymbol{H}, \frac{1}{(N!)^K} \sum_{i=1}^{K} \sum_{\substack{\pi_1 \cdots \pi_K \\ \text{except } \pi_i}} \left( \boldsymbol{T}^{(i)} - \tilde{\mathbb{E}}\boldsymbol{T}^{(i)} \right) \sum_{\pi_i} \left( \sum_{l=0}^{N-1} \boldsymbol{S}_l^{(ii)} \right) \left( \boldsymbol{T}^{(i)} - \tilde{\mathbb{E}}\boldsymbol{T}^{(i)} \right) \right\rangle \right|.$$

Next, we use Lemma J.1 and the fact that $\boldsymbol{S}_l^{(ij)} \lesssim \boldsymbol{I}$ to bound the matrix inner products, and apply the concentration inequalities we derived for $\left( (\boldsymbol{T})^{(i)} \right)_i$:

$$I_{21} \le \frac{\eta^2\sigma^2 N D^2 \text{tr}(\boldsymbol{H})}{2(N!)^{K-1}} \sum_{i=1}^{K} \sum_{\substack{\pi_1 \cdots \pi_K \\ \text{except } \pi_i}} \left( \mathbb{E} \left\| \boldsymbol{T}^{(i)} - \tilde{\mathbb{E}}\boldsymbol{T}^{(i)} \right\| + \mathbb{E} \left\| \boldsymbol{T}^{(i)} - \tilde{\mathbb{E}}\boldsymbol{T}^{(i)} \right\| \right.$$

$$\left. + \mathbb{E} \left\| \boldsymbol{T}^{(i)} - \tilde{\mathbb{E}}\boldsymbol{T}^{(i)} \right\|^2 \right)$$

$$\le \frac{\eta^2\sigma^2 N D^2 \text{tr}(\boldsymbol{H})}{2} \sum_{i=1}^{K} \left( 4K\sqrt{\delta_{\text{A}}\eta^2 KN} + \left( 2K\sqrt{2\delta_{\text{A}}\eta^2 KN} \right)^2 \right).$$

Then we bound $I_{22}$. Recall that $\boldsymbol{I} - \eta\boldsymbol{H} \lesssim \boldsymbol{I}$, we get

$$I_{22} \le \left| \frac{\eta^2\sigma^2}{2} \mathbb{E} \left\langle \boldsymbol{H}, \frac{1}{N!} \sum_{i=1}^{K} \sum_{\pi_i} \left( \sum_{l=0}^{N-1} \boldsymbol{S}_l^{(ii)} - \tilde{\mathbb{E}}\boldsymbol{S}_l^{(ii)} \right) \right\rangle \right|.$$

Recall that for a fixed $i$, for all $m \in [0, N-1]$, there are $(N-1)!$ permutations $\pi_i$ that satisfies $\pi_i(m) = l$. So

$$I_{22} \le \left| \frac{\eta^2\sigma^2}{2} \mathbb{E} \left\langle \boldsymbol{H}, \frac{1}{N!} \sum_{i=1}^{K} \sum_{l=0}^{N-1} \sum_{m=0}^{N-1} (N-1)! \left( \boldsymbol{Z}_{N-1 \to m+1}^{(i)} \boldsymbol{H} \boldsymbol{Z}_{m+1 \to N-1}^{(i)} \right. \right. \right.$$

$$\left. \left. \left. - \mathbb{E}\boldsymbol{Z}_{N-1 \to m+1}^{(i)} \boldsymbol{H} \mathbb{E}\boldsymbol{Z}_{m+1 \to N-1}^{(i)} \right) \right\rangle \right|$$

$$= \left| \frac{\eta^2\sigma^2}{2} \mathbb{E} \left\langle \boldsymbol{H}, \frac{1}{N!} \sum_{i=1}^{K} \sum_{l=0}^{N-1} (N-1)! \sum_{m=0}^{N-2} \left( \left( \boldsymbol{Z}_{N-1 \to m+1}^{(i)} - \mathbb{E}\boldsymbol{Z}_{N-1 \to m+1}^{(i)} \right) \boldsymbol{H} \right. \right. \right.$$

$$\left(\boldsymbol{Z}_{N-1\to m+1}^{(i)} - \mathbb{E}\boldsymbol{Z}_{N-1\to m+1}^{(i)}\right)\right)\right\rangle\right|.$$

Using Lemma J.4, we have

$$I_{22} \le \frac{\eta^2\sigma^2\mathrm{tr}(\boldsymbol{H})\|\boldsymbol{H}\|N}{2N}\mathbb{E}\sum_{i=1}^{K}\left(\sum_{m=0}^{N-2}\left\|\boldsymbol{Z}_{N-1\to m+1}^{(i)} - \mathbb{E}\boldsymbol{Z}_{N-1\to m+1}^{(i)}\right\|^2\right)$$

$$\le \frac{\eta^2\sigma^2\mathrm{tr}(\boldsymbol{H})\|\boldsymbol{H}\|N}{2N}\sum_{i=1}^{K}\sum_{m=0}^{N-2}\left(\sqrt{2\delta_{\mathrm{A}}\eta^2(N-1-m)}\right)^2$$

$$\lesssim \eta^4 N^2 K\log d \quad \text{when} \quad \eta = o(\frac{1}{\sqrt{T}}).$$

Combining all the arguments above, we derive that

$$\left|\tilde{\mathcal{R}}^{\mathrm{var}}(K,N;\eta) - \bar{\mathcal{R}}^{\mathrm{var}}(K,N;\eta)\right|$$

$$\le I_{11} + I_{12} + I_{21} + I_{22}$$

$$\le C\frac{\eta^2\sigma^2 N D^2\mathrm{tr}(\boldsymbol{H})}{2}\sum_{i,j=1}^{K}\left(4K\sqrt{\delta_{\mathrm{A}}\eta^2 NK} + \left(2K\sqrt{\delta_{\mathrm{A}}\eta^2 NK}\right)^2\right)$$

$$+ O(\eta^3 K^2\sqrt{N\log d} + \eta^4 K^2 N^2\log d) + O(\eta^4 N^2 K\log d)$$

$$= O(\eta^3 N^{\frac{3}{2}} K^{\frac{7}{2}}\sqrt{\log d}) \quad \text{when} \quad \eta = o(\frac{1}{\sqrt{T}}).$$

The above equation completes the proof.

### G.2.2 BIAS TERM ANALYSIS: PROOF OF LEMMA G.3

For simplicity, and as we did in the proof of Lemma G.2, in this section we omit the superscript "bias" for all the training paramters $\boldsymbol{\theta}^{\mathrm{bias}}$. Analogous to the proof of Lemma G.2, we can derive the parameter recursion as

$$\boldsymbol{\theta}_{kN} = (\boldsymbol{I} - \eta\boldsymbol{x}_{\pi_k(N-1)}\boldsymbol{x}_{\pi_k(N-1)}^\top)\boldsymbol{\theta}_{kN-1}$$

$$= \cdots$$

$$= (\boldsymbol{I} - \eta\boldsymbol{x}_{\pi_k(N-1)}\boldsymbol{x}_{\pi_k(N-1)}^\top)\cdots(\boldsymbol{I} - \eta\boldsymbol{x}_{\pi_k(0)}\boldsymbol{x}_{\pi_k(0)}^\top)\boldsymbol{\theta}_{(k-1)N}$$

$$= \boldsymbol{A}^{(k)}\boldsymbol{\theta}_{(k-1)N}.$$

For the parameter after $K$-epochs updates, we have

$$\boldsymbol{\theta}_{KN} = \boldsymbol{A}^{(K)}\cdots\boldsymbol{A}^{(1)}\boldsymbol{\theta}_0 = \prod_{l=K}^{1}\boldsymbol{A}^{(l)}\boldsymbol{\theta}_0.$$

We also have the approximation for the bias term

$$\bar{\mathcal{R}}^{\mathrm{bias}}(K,N;\eta) = \frac{1}{2}\left\langle\boldsymbol{H}, \mathbb{E}\boldsymbol{\theta}_{KN}^2\right\rangle$$

$$= \mathbb{E}\frac{1}{2}\boldsymbol{\theta}_{KN}^\top\boldsymbol{H}\boldsymbol{\theta}_{KN}$$

$$= \mathbb{E}\frac{1}{2}\boldsymbol{\theta}_0^\top\left(\prod_{l=K}^{1}\boldsymbol{A}^{(l)}\right)^\top\boldsymbol{H}\left(\prod_{l=K}^{1}\boldsymbol{A}^{(l)}\right)\boldsymbol{\theta}_0$$

$$\approx \frac{1}{2}\boldsymbol{\theta}_0^\top\left(\prod_{l=K}^{1}\mathbb{E}\boldsymbol{A}^{(l)}\right)^\top\boldsymbol{H}\left(\prod_{l=K}^{1}\mathbb{E}\boldsymbol{A}^{(l)}\right)\boldsymbol{\theta}_0$$

$$= \underbrace{\frac{1}{2}\boldsymbol{\theta}_0^\top\left((\boldsymbol{I} - \eta\boldsymbol{H})^{KN}\right)\boldsymbol{H}\left((\boldsymbol{I} - \eta\boldsymbol{H})^{KN}\right)\boldsymbol{\theta}_0}_{=:\tilde{\mathcal{R}}^{\mathrm{var}}(K,N;\eta)}.$$

The estimate error can be given as

$$
\left| \tilde{\mathcal{R}}^{\text{bias}}(K,N;\eta) - \bar{\mathcal{R}}^{\text{bias}}(K,N;\eta) \right|
$$

$$
= \left| \mathbb{E} \frac{1}{2} \boldsymbol{\theta}_0^\top \left( \prod_{l=K}^{1} \boldsymbol{A}^{(l)} \right)^\top \boldsymbol{H} \left( \prod_{l=K}^{1} \boldsymbol{A}^{(l)} \right) \boldsymbol{\theta}_0 - \frac{1}{2} \boldsymbol{\theta}_0^\top \left( \prod_{l=K}^{1} \mathbb{E}\boldsymbol{A}^{(l)} \right)^\top \boldsymbol{H} \left( \prod_{l=K}^{1} \mathbb{E}\boldsymbol{A}^{(l)} \right) \boldsymbol{\theta}_0 \right|
$$

$$
= \left| \mathbb{E} \frac{1}{2} \boldsymbol{\theta}_0^\top \left( \prod_{l=K}^{1} \boldsymbol{A}^{(l)} - \|\mathbb{E}\boldsymbol{A}\|^K \right)^\top \boldsymbol{H} \left( \prod_{l=K}^{1} \boldsymbol{A}^{(l)} - \|\mathbb{E}\boldsymbol{A}\|^K \right) \boldsymbol{\theta}_0 \right|
$$

$$
+ 2 \left| \mathbb{E} \frac{1}{2} \boldsymbol{\theta}_0^\top \|\mathbb{E}\boldsymbol{A}\|^K \boldsymbol{H} \left( \prod_{l=K}^{1} \boldsymbol{A}^{(l)} - \|\mathbb{E}\boldsymbol{A}\|^K \right) \boldsymbol{\theta}_0 \right|
$$

$$
\leq \mathbb{E} \frac{1}{2} \|\boldsymbol{H}\| \|\boldsymbol{\theta}_0\|^2 \left( \left\| \boldsymbol{A}^K - (\mathbb{E}\boldsymbol{A})^K \right\|^2 + 2\|\mathbb{E}\boldsymbol{A}\|^K \left\| \boldsymbol{A}^K - (\mathbb{E}\boldsymbol{A})^K \right\| \right). \tag{10}
$$

where the last equation uses the fact that $\|\mathbb{E}\boldsymbol{A}\| \leq 1$. Next, we discuss the approximation error bound for the bias term in Equation (10), with different categorizations based on the range of $K$.

1. Under Assumption 4.1 and $K = o\left( \frac{N^{\frac{1}{5}}}{(\log N)^{\frac{6}{5}}} \right)$:

   (a) $\eta \leq \frac{2\log T}{3\lambda_d T}$. We now verify that $K = o\left( \frac{\|\mathbb{E}\boldsymbol{A}\|}{\eta\sqrt{T}} \right)$ under given conditions. We have

   $$
   \|\mathbb{E}\boldsymbol{A}\| = (1 - \eta\lambda_d)^N = (1 - \eta\lambda_d)^{\frac{T}{K}} \geq (1 - \eta\lambda_d)^{\frac{T}{2}}
   $$
   $$
   \geq \left( 1 - \frac{2\log T}{3T} \right)^{\frac{T}{2}} = e^{\frac{T}{2} \log\left(1 - \frac{2\log T}{3T}\right)}
   $$
   $$
   = e^{-\frac{\log T}{3} + O\left(\frac{2\log^2 T}{9T}\right)} = \Theta\left(\frac{1}{T^{\frac{1}{3}}}\right).
   $$

   thus

   $$
   \frac{\|\mathbb{E}\boldsymbol{A}\|}{\eta\sqrt{T}} = \Omega\left(\frac{T^{\frac{1}{6}}}{\log T}\right).
   $$

   Also, given $K = o\left( \frac{N^{\frac{1}{5}}}{(\log N)^{\frac{6}{5}}} \right)$, we obtain that

   $$
   K = o\left( \frac{T^{\frac{1}{6}}}{\log N} \right) = o\left( \frac{T^{\frac{1}{6}}}{\log T} \right).
   $$

   The second equality uses $\log T = \log N + \log K = \Theta(\log N)$. Now we use the results in Lemma J.5 and Lemma J.6, and then the estimated distance can be given as

   $$
   \left| \tilde{\mathcal{R}}^{\text{bias}}(K,N;\eta) - \bar{\mathcal{R}}^{\text{bias}}(K,N;\eta) \right|
   $$
   $$
   \leq \frac{1}{2}\|\boldsymbol{H}\|\|\boldsymbol{\theta}_0\|^2 \|\mathbb{E}\boldsymbol{A}\|^{2K} \left( \left( (\frac{\sqrt{2\delta_A\eta^2 NK}}{\|\mathbb{E}\boldsymbol{A}\|} + 1)^K - 1 \right)^2 + 2\left( (\frac{\sqrt{2\delta_A\eta^2 NK}}{\|\mathbb{E}\boldsymbol{A}\|} + 1)^K - 1 \right) \right)
   $$
   $$
   \leq \frac{1}{2}\|\boldsymbol{H}\|\|\boldsymbol{\theta}_0\|^2 \|\mathbb{E}\boldsymbol{A}\|^{2K} \left( \frac{8K^2\delta_A\eta^2 NK}{\|\mathbb{E}\boldsymbol{A}\|^2} + 4K\frac{\sqrt{2\delta_A\eta^2 NK}}{\|\mathbb{E}\boldsymbol{A}\|} \right)
   $$
   $$
   = O\left( \|\mathbb{E}\boldsymbol{A}\|^{2K-1} K\sqrt{\eta^2 NK} \right),
   $$

   where the second inequality is by Lemma J.2.

(b) $\eta \geq \frac{2 \log T}{3 \lambda_d T}$. We have

$$\left| \tilde{\mathcal{R}}^{\text{bias}}(k, N; \eta) - \bar{\mathcal{R}}^{\text{bias}}(k, N; \eta) \right| \leq \tilde{\mathcal{R}}^{\text{bias}}(k, N; \eta) + \bar{\mathcal{R}}^{\text{bias}}(k, N; \eta)$$

$$\leq \left[ \tilde{\mathcal{R}}^{\text{bias}}(k, N; \eta) + \bar{\mathcal{R}}^{\text{bias}}(k, N; \eta) \right] \Big|_{\eta = \frac{2 \log T}{3 \lambda_d T}}$$

$$\leq \left[ \left| \bar{\mathcal{R}}^{\text{bias}}(k, N; \eta) - \tilde{\mathcal{R}}^{\text{bias}}(k, N; \eta) \right| + 2 \tilde{\mathcal{R}}^{\text{bias}}(k, N; \eta) \right] \Big|_{\eta = \frac{2 \log T}{3 \lambda_d T}}$$

$$\leq \left[ O \left( \| \mathbb{E} \boldsymbol{A} \|^{2K-1} K \sqrt{\eta^2 K N} \right) + 2 \times \frac{1}{2} \| \boldsymbol{H} \| \| \boldsymbol{\theta}_0 \|^2 \| \mathbb{E} \boldsymbol{A} \|^{2K} \right] \Big|_{\eta = \frac{2 \log T}{3 \lambda_d T}}$$

$$= O \left( \| \mathbb{E} \boldsymbol{A} \|^{2K} \right) \Big|_{\eta = \frac{2 \log T}{3 \lambda_d T}} = O \left( \left( 1 - \frac{2 \log T}{3T} \right)^{2KN} \right)$$

$$= O(\frac{1}{T^{\frac{4}{3}}}) \quad \text{when} \quad K = o \left( \frac{N^{\frac{1}{5}}}{(\log N)^{\frac{6}{5}}} \right),$$

where the first equality uses the fact that $K = o \left( \frac{\| \mathbb{E} \boldsymbol{A} \|}{\eta \sqrt{T}} \right)$ when $\eta = \frac{2 \log T}{3 \lambda_d T}$.

2. For the $K = 1$ case, which is equivalent to one-pass (OP) SGD, we derive a different upper bound for bias term error. In this scenario, we have the update rule as

$$\boldsymbol{\theta}_t = (\boldsymbol{I} - \eta \boldsymbol{x}_t \boldsymbol{x}_t^\top) \boldsymbol{\theta}_{t-1}.$$

We can denote the covariance as $\boldsymbol{B}_t$, which is

$$\begin{aligned}
\boldsymbol{B}_t &:= \mathbb{E} \boldsymbol{\theta}_t \boldsymbol{\theta}_t^\top \\
&= \mathbb{E}(\boldsymbol{I} - \eta \boldsymbol{x}_t \boldsymbol{x}_t^\top) \boldsymbol{\theta}_{t-1} \boldsymbol{\theta}_{t-1}^\top (\boldsymbol{I} - \eta \boldsymbol{x}_t \boldsymbol{x}_t^\top) \\
&= \boldsymbol{B}_{t-1} - \eta \boldsymbol{H} \boldsymbol{B}_{t-1} - \eta \boldsymbol{B}_{t-1} \boldsymbol{H} + \eta^2 \mathbb{E} \boldsymbol{x}_t \boldsymbol{x}_t^\top \boldsymbol{\theta}_{t-1} \boldsymbol{\theta}_{t-1}^\top \boldsymbol{x}_t \boldsymbol{x}_t^\top \\
&= (\boldsymbol{I} - \eta \boldsymbol{H}) \boldsymbol{B}_{t-1} (\boldsymbol{I} - \eta \boldsymbol{H}) + \eta^2 \mathbb{E}(\boldsymbol{x}_t \boldsymbol{x}_t^\top - \boldsymbol{H}) \boldsymbol{\theta}_{t-1} \boldsymbol{\theta}_{t-1}^\top (\boldsymbol{x}_t \boldsymbol{x}_t^\top - \boldsymbol{H}). \quad (11)
\end{aligned}$$

Since the bias term in the excess risk can be represented as

$$\bar{\mathcal{R}}^{\text{bias}}(1, T; \eta) = \frac{1}{2} \langle \boldsymbol{H}, \boldsymbol{B}_T \rangle.$$

We then get the lower and upper bounds for $\boldsymbol{B}_t$, and derive the corresponding lower and upper bounds for the bias term in the excess risk.

**Lower bound.** By Equation (11), we get a lower bound of $\boldsymbol{B}_t$

$$\begin{aligned}
\boldsymbol{B}_T &\succeq (\boldsymbol{I} - \eta \boldsymbol{H}) \boldsymbol{B}_{T-1} (\boldsymbol{I} - \eta \boldsymbol{H}) \\
&\succeq \cdots \succeq (\boldsymbol{I} - \eta \boldsymbol{H})^T \boldsymbol{B}_0 (\boldsymbol{I} - \eta \boldsymbol{H})^T
\end{aligned}$$

and

$$\begin{aligned}
\bar{\mathcal{R}}^{\text{bias}}(1, T; \eta) &= \frac{1}{2} \langle \boldsymbol{H}, \boldsymbol{B}_T \rangle \\
&\geq \frac{1}{2} \langle \boldsymbol{H}, (\boldsymbol{I} - \eta \boldsymbol{H})^T \boldsymbol{B}_0 (\boldsymbol{I} - \eta \boldsymbol{H})^T \rangle \\
&= \frac{1}{2} \boldsymbol{\theta}_0^\top \left( (\boldsymbol{I} - \eta \boldsymbol{H})^T \right) \boldsymbol{H} \left( (\boldsymbol{I} - \eta \boldsymbol{H})^T \right) \boldsymbol{\theta}_0.
\end{aligned}$$

**Upper bound.** By the recursion of $\boldsymbol{B}_t$, we have

$$\begin{aligned}
\boldsymbol{B}_t &\preceq (\boldsymbol{I} - \eta \boldsymbol{H}) \boldsymbol{B}_{t-1} (\boldsymbol{I} - \eta \boldsymbol{H}) + \eta^2 \mathbb{E}_{\boldsymbol{x}_{T-1}, \cdots \boldsymbol{x}_0} \mathbb{E}_{\boldsymbol{x}_T} (\boldsymbol{x}_t \boldsymbol{x}_t^\top - \boldsymbol{H}) \boldsymbol{\theta}_{t-1} \boldsymbol{\theta}_{t-1}^\top (\boldsymbol{x}_t \boldsymbol{x}_t^\top - \boldsymbol{H}) \\
&= (\boldsymbol{I} - \eta \boldsymbol{H}) \boldsymbol{B}_{t-1} (\boldsymbol{I} - \eta \boldsymbol{H}) + \eta^2 \mathbb{E}_{\boldsymbol{x}_{T-1}, \cdots \boldsymbol{x}_0} \left[ \mathbb{E}_{\boldsymbol{x}_T} \left[ \boldsymbol{x}_T \boldsymbol{x}_T^\top \boldsymbol{\theta}_{T-1} \boldsymbol{\theta}_{T-1}^\top \boldsymbol{x}_T \boldsymbol{x}_T^\top \right] - \boldsymbol{H} \boldsymbol{\theta}_{T-1} \boldsymbol{\theta}_{T-1}^\top \boldsymbol{H} \right] \\
&\preceq (\boldsymbol{I} - \eta \boldsymbol{H}) \boldsymbol{B}_{t-1} (\boldsymbol{I} - \eta \boldsymbol{H}) + \eta^2 \mathbb{E}_{\boldsymbol{x}_{T-1}, \cdots \boldsymbol{x}_0} \mathbb{E}_{\boldsymbol{x}_T} \left[ \boldsymbol{x}_T \boldsymbol{x}_T^\top \boldsymbol{\theta}_{T-1} \boldsymbol{\theta}_{T-1}^\top \boldsymbol{x}_T \boldsymbol{x}_T^\top \right].
\end{aligned}$$

Then, combining Assumption 4.1 and Lemma J.9 gives

$$
\begin{aligned}
\boldsymbol{B}_T &\preceq (\boldsymbol{I} - \eta\boldsymbol{H})\boldsymbol{B}_{T-1}(\boldsymbol{I} - \eta\boldsymbol{H}) + \eta^2\alpha\mathbb{E}_{\boldsymbol{x}_{T-1},\cdots\boldsymbol{x}_0}\mathrm{tr}(\boldsymbol{H}\boldsymbol{\theta}_{T-1}\boldsymbol{\theta}_{T-1}^{\top})\boldsymbol{H} \\
&= (\boldsymbol{I} - \eta\boldsymbol{H})\boldsymbol{B}_{T-1}(\boldsymbol{I} - \eta\boldsymbol{H}) + \eta^2\alpha\left\langle \boldsymbol{H}, \boldsymbol{B}_{T-1}\right\rangle \boldsymbol{H} \\
&\preceq \cdots \\
&\preceq (\boldsymbol{I} - \eta\boldsymbol{H})^T\boldsymbol{B}_0(\boldsymbol{I} - \eta\boldsymbol{H})^T + \eta^2\alpha\sum_{i=0}^{T-1}\left\langle \boldsymbol{B}_i, \boldsymbol{H}\right\rangle (\boldsymbol{I} - \eta\boldsymbol{H})^{2(T-i-1)}\boldsymbol{H},
\end{aligned}
$$

and

$$
\left\langle \boldsymbol{H}, \boldsymbol{B}_T\right\rangle \leq \left\langle \boldsymbol{H}, (\boldsymbol{I} - \eta\boldsymbol{H})^T\boldsymbol{B}_0(\boldsymbol{I} - \eta\boldsymbol{H})^T\right\rangle + \eta^2\alpha\sum_{i=0}^{T-1}\left\langle \boldsymbol{H}, \boldsymbol{B}_i\right\rangle \left\langle (\boldsymbol{I} - \eta\boldsymbol{H})^{2(T-i-1)}\boldsymbol{H}, \boldsymbol{H}\right\rangle.
$$

We also have

$$
\begin{aligned}
\left\langle \boldsymbol{H}, \boldsymbol{B}_i\right\rangle &\leq \left\langle \boldsymbol{H}, (\boldsymbol{I} - \eta\boldsymbol{H})\boldsymbol{B}_{i-1}(\boldsymbol{I} - \eta\boldsymbol{H})\right\rangle + \eta^2\alpha\mathrm{tr}(\boldsymbol{H}^2)\left\langle \boldsymbol{H}, \boldsymbol{B}_{i-1}\right\rangle \\
&\leq (1 - \eta\lambda_d)^2\left\langle \boldsymbol{H}, \boldsymbol{B}_{i-1}\right\rangle + \eta^2\alpha\mathrm{tr}(\boldsymbol{H}^2)\left\langle \boldsymbol{H}, \boldsymbol{B}_{i-1}\right\rangle \\
&\leq \cdots \\
&\leq [(\lambda_d^2 + \alpha\mathrm{tr}(\boldsymbol{H}^2))\eta^2 - 2\lambda_d\eta + 1]^i\left\langle \boldsymbol{H}, \boldsymbol{B}_0\right\rangle \\
&\leq e^{T\log[(\lambda_d^2 + \alpha\mathrm{tr}(\boldsymbol{H}^2))\eta^2 - 2\lambda_d\eta + 1]}\left\langle \boldsymbol{H}, \boldsymbol{B}_0\right\rangle \\
&= e^{-2\lambda_d\eta i + O(\eta^2 i)}\left\langle \boldsymbol{H}, \boldsymbol{B}_0\right\rangle \\
&\leq C_1 e^{-2\lambda_d\eta i}\left\langle \boldsymbol{H}, \boldsymbol{B}_0\right\rangle
\end{aligned}
$$

and

$$
\begin{aligned}
\left\langle (\boldsymbol{I} - \eta\boldsymbol{H})^{2(T-i-1)}\boldsymbol{H}, \boldsymbol{H}\right\rangle &= \left\langle (\boldsymbol{I} - \eta\boldsymbol{H})^{2(T-i-1)}, \boldsymbol{H}^2\right\rangle \\
&\leq \mathrm{tr}\left(\boldsymbol{H}^2\right)(1 - \eta\lambda_d)^{2(T-1-i)} \\
&\leq \mathrm{tr}\left(\boldsymbol{H}^2\right)e^{2(T-1-i)\log(1-\eta\lambda_d)} \\
&= \mathrm{tr}\left(\boldsymbol{H}^2\right)e^{-2(T-1-i)\eta\lambda_d + O\left(\eta^2(T-1-i)\right)} \\
&\leq C_2 e^{-2(T-1-i)\eta\lambda_d}
\end{aligned}
$$

So

$$
\begin{aligned}
\left\langle \boldsymbol{H}, \boldsymbol{B}_i\right\rangle &\leq \left\langle \boldsymbol{H}, (\boldsymbol{I} - \eta\boldsymbol{H})^T\boldsymbol{B}_0(\boldsymbol{I} - \eta\boldsymbol{H})^T\right\rangle + \eta^2\alpha\sum_{i=0}^{T-1}C_1 e^{-2\lambda_d\eta i}\left\langle \boldsymbol{H}, \boldsymbol{B}_0\right\rangle C_2 e^{-2\lambda_d\eta(T-1-i)}\mathrm{tr}\left(\boldsymbol{H}^2\right) \\
&= \left\langle \boldsymbol{H}, (\boldsymbol{I} - \eta\boldsymbol{H})^T\boldsymbol{B}_0(\boldsymbol{I} - \eta\boldsymbol{H})^T\right\rangle + C_3\eta^2 T e^{-2\lambda_d\eta T}
\end{aligned}
$$

And finally we get

$$
\left|\bar{\mathcal{R}}^{\mathrm{bias}}(1, T; \eta) - \frac{1}{2}\left\langle \boldsymbol{H}, (\boldsymbol{I} - \eta\boldsymbol{H})^{\top}\boldsymbol{B}_0(\boldsymbol{I} - \eta\boldsymbol{H})\right\rangle\right| = O(\eta^2 T e^{-2\lambda_d\eta T}).
$$

### G.3 STEP III: NARROWING THE RANGE FOR OPTIMAL LEARNING RATE

We recap that our goal is to derive the scaling law formula for strongly convex linear regression with multi-epoch SGD, and the formula of the effective reuse rate. Before we start our proof, we first give a technical lemma below.

**Lemma G.4.** *Given $\eta \in \left[\omega\left(\frac{1}{T}\right), o\left(\frac{1}{\sqrt{T}}\right)\right]$, and define $n_d$ to be the number of the minimal eigenvalue $\lambda_d$ in $\boldsymbol{H}$, then it holds that*

$$
\sum_{i=1}^{d}(\boldsymbol{P}\boldsymbol{\theta}_0)_i^2\lambda_i(1 - \eta\lambda_i)^{2T} = \tilde{\theta}_d^2\lambda_d\exp(-2\lambda_d\eta T)(1 + o(1)),
$$

$$
\sum_{i=1}^{d}\lambda_i(1 - \eta\lambda_i)^{2T} = n_d\lambda_d\exp(-2\lambda_d\eta T)(1 + o(1)).
$$

*Proof of Lemma G.4.* For the first equation, for any $\lambda_i > \lambda_d$, we define $\rho_i = \frac{\lambda_i}{\lambda_d} > 1$, then we have

$$
\begin{aligned}
(1 - \eta\lambda_i)^{2T} &= \exp\left(2T\log(1 - \eta\lambda_i)\right) = \exp\left(2T(-\eta\lambda_i + O(\eta^2\lambda_i^2))\right) \\
&= \exp(-2\lambda_i\eta T)\exp(O(\eta^2 T)) = \exp(-2\lambda_d\rho_i\eta T)(1 + o(1)) \\
&= (\exp(-2\lambda_d\eta T))^{\rho_i}(1 + o(1)) = o(\exp(-2\lambda_d\eta T)). \tag{12}
\end{aligned}
$$

Since $\lambda_i \leq D^2$, we have

$$
\sum_{i=1}^{d-n_d} (\boldsymbol{P}\boldsymbol{\theta}_0)_i^2 \lambda_i (1 - \eta\lambda_i)^{2T} = o(\exp(-2\lambda_d\eta T)),
$$

From Equation (12), we can also directly get the second equation, which completes the proof of Lemma G.4. $\qquad\square$

### G.3.1 A DESCRIPTION OF THE RANGE OF OPTIMAL LEARNING RATE, SMALL-$K$ CASE

**Lemma G.5.** *Under the conditions in Lemma 4.4, and when $K = o(\log N)$, we have $\eta^* \in [\frac{\log T}{3\lambda_d T}, \frac{\alpha\log T}{T}]$, where the constant $\alpha := \frac{D^2\mathrm{tr}(\boldsymbol{H})}{\lambda_d\mathrm{tr}(\boldsymbol{H}^2)}$.*

*Proof.* We first prove the upper bound. Given a learning rate $\eta$, Equation (6) gives

$$
\bar{\mathcal{R}}(K,N;\eta) \geq \bar{\mathcal{R}}^{\mathrm{var}}(K,N;\eta) =
$$

$$
\underbrace{\frac{\eta^2\sigma^2}{2}\mathbb{E}\left\langle \boldsymbol{H}, \frac{1}{(N!)^K}\sum_{\substack{i\neq j \\ i,j=1}}^{K}\sum_{\substack{\pi_1\cdots\pi_K \\ \text{except }\pi_i,\pi_j}}\boldsymbol{T}^{(i)}\sum_{\pi_i,\pi_j}\left(\sum_{l=0}^{N-1}\boldsymbol{S}_l^{(ij)}\right)\left(\boldsymbol{T}^{(j)}\right)^\top\right\rangle}_{=:\psi_1}
$$

$$
+ \underbrace{\frac{\eta^2\sigma^2}{2}\mathbb{E}\left\langle \boldsymbol{H}, \frac{1}{(N!)^K}\sum_{i=1}^{K}\sum_{\substack{\pi_1\cdots\pi_K \\ \text{except }\pi_i}}\boldsymbol{T}^{(i)}\sum_{\pi_i}\left(\sum_{l=0}^{N-1}\boldsymbol{S}_l^{(ii)}\right)\left(\boldsymbol{T}^{(i)}\right)^\top\right\rangle}_{=:\psi_2}.
$$

For $\psi_1$, using the fact that $(\boldsymbol{I} - \eta\boldsymbol{x}\boldsymbol{x}^\top) \succeq (\boldsymbol{I} - \eta D^2\boldsymbol{I})$, we replace all the terms $(\boldsymbol{I} - \eta\boldsymbol{x}\boldsymbol{x}^\top)$ with $(\boldsymbol{I} - \eta D^2\boldsymbol{I})$ thus we have a lower bound for $\psi_1$

$$
\psi_1 \geq \frac{\eta^2\sigma^2}{2}\left\langle \boldsymbol{H}, \frac{N((N-1)!)^2}{(N!)^K}\sum_{\substack{i\neq j \\ i,j=1}}^{K}\sum_{\substack{\{\pi_1\cdots\pi_K\} \\ \backslash\{\pi_i,\pi_j\}}}(1-\eta D^2)^{(2K-i-j)N}\left(\sum_{m,n=0}^{N-1}(1-\eta D^2)^{2N-2-m-n}\mathbb{E}[\boldsymbol{x}\boldsymbol{x}^\top]\right)\right\rangle
$$

$$
= \frac{\eta^2\sigma^2}{2ND^4}\left\langle \boldsymbol{H}, \sum_{\substack{i\neq j \\ i,j=1}}(1-\eta D^2)^{(K-i)N}(1-\eta D^2)^{(K-j)N}\left(1-(1-\eta D^2)^N\right)^2\boldsymbol{H}\right\rangle
$$

$$
= \frac{\sigma^2}{2ND^4}\mathrm{tr}\left(\boldsymbol{H}^2\left(1-(1-\eta D^2)^{KN}\right)^2\right) - \frac{\sigma^2}{2ND^4}\mathrm{tr}\left(\boldsymbol{H}^2\frac{1-\left(1-(1-\eta D^2)^N\right)^{2KN}}{1-(1-\eta D^2)^{2N}}\right)
$$

$$
= \frac{\sigma^2}{ND^4}\mathrm{tr}\left(\boldsymbol{H}^2\frac{1-(1-\eta D^2)^{KN}}{1+(1-\eta D^2)^N}\left((1-\eta D^2)^N-(1-\eta D^2)^{KN}\right)\right).
$$

For $\psi$, we use a similar argument to get its lower bound

$$
\psi_2 \geq \frac{\eta^2\sigma^2}{2}\left\langle \boldsymbol{H}, \sum_{i=1}^{K}(1-\eta D^2)^{2N(K-i)}\frac{1-(1-\eta D^2)^{2N}}{1-(1-\eta D^2)^2}\boldsymbol{H}\right\rangle
$$

$$
= \frac{\eta\sigma^2}{2D^2}\left\langle \boldsymbol{H}, \frac{1-(1-\eta D^2)^{2KN}}{1-(1-\eta D^2)^{2N}}\frac{1-(1-\eta D^2)^{2N}}{1-(1-\eta D^2)^2}\boldsymbol{H}\right\rangle
$$

$$
= \frac{\eta\sigma^2\mathrm{tr}(\boldsymbol{H}^2)}{4D^2}(1+o(1)).
$$

Notice that from the above lower bound, when $K = o(\log N)$, we have

$$\bar{\mathcal{R}}(K, N; \eta) \geq \psi_1 + \psi_2$$
$$\geq O(\frac{1}{N}) + \frac{\eta\sigma^2\mathrm{tr}(\boldsymbol{H}^2)}{4D^2}(1 + o(1))$$
$$= \frac{\eta\sigma^2\mathrm{tr}(\boldsymbol{H}^2)}{4D^2}(1 + o(1)). \tag{13}$$

Taking $\eta > \frac{\alpha \log T}{T}$, and $\alpha = \frac{D^2\mathrm{tr}(\boldsymbol{H})}{\lambda_d\mathrm{tr}(\boldsymbol{H}^2)}$ gives

$$\bar{\mathcal{R}}(K, N; \eta) \geq \frac{\sigma^2\mathrm{tr}(\boldsymbol{H})\log T}{4\lambda_d T}(1 + o(1)).$$

Now we recall that

$$\bar{\mathcal{R}}^*(K, N) \leq \bar{\mathcal{R}}(K, N; \eta') = M(K, N; \eta')(1 + o(1))$$
$$= \frac{\sigma^2\mathrm{tr}(\boldsymbol{H})\log T}{8\lambda_d T}(1 + o(1)) < \frac{\sigma^2\mathrm{tr}(\boldsymbol{H})\log T}{4\lambda_d T}(1 + o(1))$$

Thus we have that $\eta^* \leq \frac{\alpha \log T}{T}$. Next, we give the lower bound of $\eta^*$.

When $\eta < \frac{\log T}{3\lambda_d T}$, we have that

$$\exp(-2\lambda_d\eta T) = \frac{1}{T^{2/3}} = \omega(\frac{\log T}{T}) = \omega(\bar{\mathcal{R}}(K, N; \eta')) = \omega(\bar{\mathcal{R}}^*(K, N)).$$

The above equation shows $\eta^* > \frac{\log T}{3\lambda_d T}$, which completes the proof. $\qquad\square$

### G.3.2 A DESCRIPTION OF THE RANGE OF OPTIMAL LEARNING RATE, LARGE-$K$ CASE

**Lemma G.6.** *Under the conditions in Lemma 4.4, and when $K = \omega(\log N)$, we have $\eta^* \in [\frac{\log T}{3\lambda_d T}, o(\frac{1}{N})]$.*

*Proof.* The proof comprises of three parts. First, we prove that $\eta^* \geq \frac{\log T}{3\lambda_d T}$ when $T$ is large. Second, we verify that $\eta^* \leq \frac{c}{N}$ for sufficiently large $N$. Finally, we refine the proof in the second step and justify that $\eta^* = o(\frac{1}{N})$. All proofs are carried out by contradiction. The method proceeds as follows: we take a specific $\eta = \eta'$ and compute its loss, then prove that $\bar{\mathcal{R}}^*(K, N) > \bar{\mathcal{R}}(K, N; \eta')$ when $N$ is sufficiently large if $\eta^*$ does not fall into some interval.

First, by Equation (15), we have

$$\bar{\mathcal{R}}(K, N; \eta') = \frac{\sigma^2 d}{2N}(1 + o(1)).$$

Then we begin our main part of the proof.

*Proof Step I: $\eta^* \geq \frac{\log T}{3\lambda_d T}$.*

We assume that $\eta^* < \frac{\log T}{3\lambda_d T}$. Observe that $\bar{\mathcal{R}}^{\mathrm{bias}}(K, N; \eta)$ decreases with $\eta$. So

$$\bar{\mathcal{R}}^*(K, N) \geq \bar{\mathcal{R}}^{\mathrm{bias}}(K, N; \eta^*) \geq \bar{\mathcal{R}}^{\mathrm{bias}}(K, N; \eta = \frac{\log T}{3\lambda_d T})$$
$$= \frac{1}{2}(\boldsymbol{w}_0 - \boldsymbol{w}^*)^\top(\boldsymbol{I} - \eta\boldsymbol{H})^{2T}\boldsymbol{H}(\boldsymbol{w}_0 - \boldsymbol{w}^*)(1 + o(1))\Big|_{\eta=\frac{\log T}{3\lambda_d T}}$$
$$= \left(\frac{1}{2}\tilde{\theta}_d^2\lambda_d\exp(-2\lambda_d\eta T)\right)(1 + o(1))\Big|_{\eta=\frac{\log T}{3\lambda_d T}}$$
$$= \Theta\left(\frac{1}{T^{\frac{2}{3}}}\right) = \omega\left(\frac{1}{N}\right),$$

where the first equality is due to Lemma G.3, the second equality is due to Lemma G.4, and the last equality is due to Assumption 4.1.

*Proof Step II:* $\eta^* \leq \frac{4D^2d}{\sigma^2 \text{tr}(\boldsymbol{H}^2)N}$. We assume that $\eta^* > \frac{4D^2d}{\sigma^2 \text{tr}(\boldsymbol{H}^2)N}$ By Equation (13), we have

$$\widehat{\mathcal{R}}(K, N; \eta) \geq \frac{\eta\sigma^2 \text{tr}(\boldsymbol{H}^2)}{4D^2}(1 + o(1)) > \frac{\sigma^2 d}{N}(1 + o(1)) > \frac{\sigma^2 d}{2N}(1 + o(1)),$$

which is a contradiction.

A direct corollary is that

$$\bar{\mathcal{R}}^*(K, N) = \widehat{\mathcal{R}}(K, N; \eta^*)(1 + o(1))$$

$$\widehat{\mathcal{R}}(K, N; \eta^*) = \frac{1}{2}(\boldsymbol{w}_0 - \boldsymbol{w}^*)^\top (\boldsymbol{I} - \eta^*\boldsymbol{H})^{2T}\boldsymbol{H}(\boldsymbol{w}_0 - \boldsymbol{w}^*)$$

$$+ \frac{\sigma^2}{N}tr\left(\frac{(\boldsymbol{I} - (\boldsymbol{I} - \eta^*\boldsymbol{H})^{KN})((\boldsymbol{I} - \eta^*\boldsymbol{H})^N - (\boldsymbol{I} - \eta^*\boldsymbol{H})^{KN})}{\boldsymbol{I} + (\boldsymbol{I} - \eta^*\boldsymbol{H})^N}\right)$$

$$+ \frac{\eta^*\sigma^2}{2}\left\langle \boldsymbol{H}, \left(\boldsymbol{I} - (\boldsymbol{I} - \eta^*\boldsymbol{H})^{2T}\right)(2\boldsymbol{I} - \eta^*\boldsymbol{H})^{-1}\right\rangle$$

$$= \frac{1}{2}\sum_{i=1}^{d}(\boldsymbol{P}\boldsymbol{\theta}_0)_l^2 \lambda_i(1 - \eta^*\lambda_i)^{2T} + \sum_{i=1}^{d}\frac{\sigma^2}{N}\frac{(1 - \eta^*\lambda_i)^N}{1 + (1 - \eta^*\lambda_i)^N}$$

$$+ \frac{\eta^*\sigma^2}{4}\text{tr}(\boldsymbol{H}) - \frac{\eta^*\sigma^2}{4}\sum_{i=1}^{d}\lambda_i(1 - \eta^*\lambda_i)^{2T} + O\left((\eta^*)^2\right)$$

$$= \left(\frac{1}{2}\tilde{\theta}_d^2\lambda_d \exp(-2\lambda_d\eta^*T) + \sum_{i=1}^{d}\frac{\sigma^2}{N}\frac{e^{-N\eta^*\lambda_i}}{1 + e^{-N\eta^*\lambda_i}} + \frac{\eta^*\sigma^2}{4}\text{tr}(\boldsymbol{H})\right)(1 + o(1)).$$

*Proof Step III:* $\eta^* = o\left(\frac{1}{N}\right)$.

We assume that there exists a constant $\epsilon > 0$ and a sequence $(N_i)_{i=1}^{\infty}$ that satisfies $N_i \to \infty$ when $i \to \infty$ and $\eta^*(N_i) \geq \frac{\epsilon}{N_i}$ for all $i$. As we only conduct our analysis on the sequence $(N_i)_{i=1}^{\infty}$, without loss of generality, we take $(N_i)_{i=1}^{\infty} = \mathbb{N}$.

We define $f(\delta) = \sum_{i=1}^{d} \sigma^2 \frac{e^{-\delta\lambda_i}}{1 + e^{-\delta\lambda_i}} + \frac{\delta\sigma^2}{4}\text{tr}(\boldsymbol{H})$. Then we have

$$f'(\delta) = \frac{\sigma^2}{4}\sum_{i=1}^{d}\lambda_i - \sum_{i=1}^{d}\sigma^2\frac{\lambda_i e^{-\delta\lambda_i}}{(1 + e^{-\delta\lambda_i})^2} = \frac{\sigma^2}{4}\sum_{i=1}^{d}\lambda_i\frac{(1 - e^{-\delta\lambda_i})^2}{(1 + e^{-\delta\lambda_i})^2} > 0 \text{ when } \delta > 0.$$

So

$$f(\epsilon) > f(0) = \frac{\sigma^2 d}{2},$$

and

$$\bar{\mathcal{R}}^*(K, N) \geq \frac{1}{N}f(\eta^*N)(1 + o(1)) \geq \frac{1}{N}f(\epsilon)(1 + o(1)) > \frac{\sigma^2 d}{2N}(1 + o(1)) = \bar{\mathcal{R}}(K, N; \eta'),$$

which is a contradiction. $\qquad\square$

### G.3.3 AN APPROXIMATION OF THE EXCESS RISK, SMALL-$K$ CASE

**Lemma G.7.** *Let* $\tilde{\theta}_d^2 = \sum_{l=d-n_d+1}^{d}(\boldsymbol{P}\boldsymbol{\theta}_0)_l^2$, $\boldsymbol{H} = \boldsymbol{P}\boldsymbol{D}\boldsymbol{P}^\top$ *to be the canonical form under similarity of* $\boldsymbol{H}$. *Under the conditions in Lemma 4.4, for learning rate* $\eta \in \left[\frac{\log KN}{3\lambda_d KN}, \frac{\alpha \log KN}{KN}\right]$ *for constant* $\alpha = \frac{D^2 \text{tr}(\boldsymbol{H})}{\lambda_d \text{tr}(\boldsymbol{H}^2)}$ *and* $K = o(\log N)$, *then we have the approximation of* $\bar{\mathcal{R}}(K, N; \eta)$ *as*

$$\bar{\mathcal{R}}(K, N; \eta) = M(K, N; \eta)(1 + o(1)),$$

$$M(K, N; \eta) := \frac{1}{2}\tilde{\theta}_d^2\lambda_d \exp(-2\lambda_d\eta T) + \frac{\eta \text{tr}(\boldsymbol{H})\sigma^2}{4},$$

*where steps* $T = KN$.

*Proof.* From Lemma G.1, we have that $\bar{\mathcal{R}}(K, N; \eta) = \widehat{\mathcal{R}}(K, N; \eta)(1 + o(1))$, where $\widehat{\mathcal{R}}(K, N; \eta)$ can be written as

$$
\begin{aligned}
\widehat{\mathcal{R}}(K, N; \eta) &= \frac{1}{2}(\boldsymbol{w}_0 - \boldsymbol{w}^*)^\top (\boldsymbol{I} - \eta \boldsymbol{H})^{2T} \boldsymbol{H}(\boldsymbol{w}_0 - \boldsymbol{w}^*) \\
&\quad + \frac{\sigma^2}{N} tr\left( \frac{\left(\boldsymbol{I} - (\boldsymbol{I} - \eta \boldsymbol{H})^{KN}\right)\left((\boldsymbol{I} - \eta \boldsymbol{H})^N - (\boldsymbol{I} - \eta \boldsymbol{H})^{KN}\right)}{\boldsymbol{I} + (\boldsymbol{I} - \eta \boldsymbol{H})^N} \right) \\
&\quad + \frac{\eta \sigma^2}{2} \left\langle \boldsymbol{H}, \left(\boldsymbol{I} - (\boldsymbol{I} - \eta \boldsymbol{H})^{2T}\right)(2\boldsymbol{I} - \eta \boldsymbol{H})^{-1} \right\rangle \\
&= \frac{1}{2} \sum_{i=1}^d (\boldsymbol{P}\boldsymbol{\theta}_0)_l^2 \lambda_i (1 - \eta \lambda_i)^{2T} + \sum_{i=1}^d \frac{\sigma^2}{N} \frac{(1 - \eta \lambda_i)^N}{1 + (1 - \eta \lambda_i)^N} \\
&\quad + \frac{\eta \sigma^2}{4} tr(\boldsymbol{H}) - \frac{\eta \sigma^2}{4} \sum_{i=1}^d \lambda_i (1 - \eta \lambda_i)^{2T} + O\left(\eta^2\right) \\
&= \underbrace{\left( \frac{1}{2} \tilde{\theta}_d^2 \lambda_d \exp(-2\lambda_d \eta T) + \frac{\eta \sigma^2}{4} tr(\boldsymbol{H}) \right)}_{M(K,N;\eta)} (1 + o(1)) + O(\frac{1}{N}) \\
&= \underbrace{\left( \frac{1}{2} \tilde{\theta}_d^2 \lambda_d \exp(-2\lambda_d \eta T) + \frac{\eta \sigma^2}{4} tr(\boldsymbol{H}) \right)}_{M(K,N;\eta)} (1 + o(1)),
\end{aligned}
\tag{14}
$$

where the second to last equation uses Lemma G.4 and the fact that $\eta(1 - \eta\lambda_d)^{2T} = o\left(M(K, N, ; \eta)\right)$ for $\eta \in [\frac{\log T}{3\lambda_d T}, \frac{\alpha \log T}{T}]$, and the last equation uses the fact that when $K = o(\log N)$, $O\left(\frac{1}{N}\right) = o\left(\frac{\log(N)}{K,N}\right) = o\left(M(T; \eta)\right)$. $\qquad \square$

### G.3.4 An Approximation of the Excess Risk, Large-$K$ Case

**Lemma G.8.** *Under the conditions in Lemma 4.4, for $\eta \in [\frac{\log T}{3\lambda_d T}, o\left(\frac{1}{N}\right)]$, and $K = \omega\left(\log N\right)$, we have*

$$
\mathbb{E}[\bar{\mathcal{R}}(K, N; \eta)] = M(K, N; \eta)(1 + o(1)),
$$
$$
M(K, N; \eta) = \frac{1}{2} \tilde{\theta}_d^2 \lambda_d \exp(-2\lambda_d \eta T) + \frac{\eta tr(\boldsymbol{H}) \sigma^2}{4} + \frac{\sigma^2 d}{2N},
$$

*where $\tilde{\theta}_d^2 := \sum_{l=d-n_d+1}^d (\boldsymbol{P}\boldsymbol{\theta}_0)_l^2$, and $\boldsymbol{P}\boldsymbol{D}\boldsymbol{P}^\top$ is the canonical form under similarity of $\boldsymbol{H}$.*

*Proof.* Given $K = O(N^{0.1})$, one can verify that

$$
\lim_{N \to \infty} K\eta T^{\frac{3}{4}} = \lim_{N \to \infty} \frac{K^{\frac{7}{4}} N^{\frac{3}{4}}}{N} \eta N = 0.
$$

So condition $K = o\left(\eta^{-1} T^{-\frac{3}{4}}\right)$ is satisfied, thus by invoking Lemma G.1, we have $\bar{\mathcal{R}}(K, N; \eta) = \widehat{\mathcal{R}}(K, N; \eta)(1 + o(1))$.

Note that when $\eta = o\left(\frac{1}{N}\right)$, for any $i \in [1, d]$, we have

$$
(1 - \lambda_i \eta)^N = e^{-\lambda_i \eta N + O(\eta^2 N)} = 1 + o(1).
$$

Combining this with Lemma G.4, we have

$$
\begin{aligned}
\widehat{\mathcal{R}}(K, N; \eta) &= \frac{1}{2}(\boldsymbol{w}_0 - \boldsymbol{w}^*)^\top (\boldsymbol{I} - \eta \boldsymbol{H})^{2T} \boldsymbol{H} (\boldsymbol{w}_0 - \boldsymbol{w}^*) \\
&\quad + \frac{\sigma^2}{N} tr \left( \frac{\left(\boldsymbol{I} - (\boldsymbol{I} - \eta\boldsymbol{H})^{KN}\right)\left((\boldsymbol{I} - \eta\boldsymbol{H})^N - (\boldsymbol{I} - \eta\boldsymbol{H})^{KN}\right)}{\boldsymbol{I} + (\boldsymbol{I} - \eta\boldsymbol{H})^N} \right) \\
&\quad + \frac{\eta\sigma^2}{2} \left\langle \boldsymbol{H}, \left(\boldsymbol{I} - (\boldsymbol{I} - \eta\boldsymbol{H})^{2T}\right)(2\boldsymbol{I} - \eta\boldsymbol{H})^{-1} \right\rangle \\
&= \frac{1}{2}\sum_{i=1}^d (\boldsymbol{P}\boldsymbol{\theta}_0)_i^2 \lambda_i (1 - \eta\lambda_i)^{2T} + \sum_{i=1}^d \frac{\sigma^2}{N} \frac{(1 - \eta\lambda_i)^N}{1 + (1 - \eta\lambda_i)^N} \\
&\quad + \frac{\eta\sigma^2}{4} tr(\boldsymbol{H}) - \frac{\eta\sigma^2}{4} \sum_{i=1}^d \lambda_i (1 - \eta\lambda_i)^{2T} + O\left(\eta^2\right) \\
&= \underbrace{\left(\frac{1}{2}\tilde{\theta}_d^2 \lambda_d \exp(-2\lambda_d\eta T) + \frac{\eta\sigma^2}{4} tr(\boldsymbol{H})\right) + \frac{\sigma^2 d}{2N}}_{M(K,N;\eta)} \left(1 + o(1)\right), \qquad (15)
\end{aligned}
$$

which concludes the proof. $\qquad\square$

### G.4 Step IV: Deriving the Approximately Optimal Learning Rate, Proof of Lemma 4.4

The proof of Lemma 4.4 for the small-$K$ case and large-$K$ case follows a similar pattern. First, we minimize the aproximate excess risk obtained in Section G.3.3 and Section G.3.4. Then we conduct an error bound analysis and complete the proof.

#### G.4.1 Proof of Lemma 4.4, small $K$

**Part I: Minimizing the Approximation of the Excess Risk**

**Lemma G.9.** *Under Assumption 4.1 and 4.3, we consider $K$-epoch SGD with $N$ fresh data and learning rate $\eta$ satisfying $\eta \in [\frac{\log T}{3\lambda_d T}, \frac{\alpha \log T}{T}]$, where steps $T := KN$ and $\alpha$ is some constant independent of $T$, but can depend on $D$ and $\lambda_1, \lambda_2, \ldots, \lambda_d$. Then when $K = o(\log N)$, the chosen learning rate $\eta' = \frac{\log \rho T}{2\lambda_d T} = \arg\min_{\eta \in [\frac{\log T}{3\lambda_d T}, \frac{\alpha \log T}{T}]} M(K, N; \eta)$.*

*Proof.* Given Lemma G.7, we take the derivative of $M(K, N; \eta)$ with respect to $\eta$

$$
\frac{\partial M}{\partial \eta} = -\tilde{\theta}_d^2 \lambda_d^2 T \exp(-2\lambda_d\eta T) + \frac{tr(\boldsymbol{H})\sigma^2}{4}.
$$

Define $\rho := \frac{4\tilde{\theta}_d^2 \lambda_d}{tr(\boldsymbol{H})\sigma^2}$, and we let $\frac{\partial M}{\partial \eta} = 0$, then we get

$$
\begin{aligned}
0 &= -\rho T \exp(-2\lambda_d\eta T) + 1 \\
\rho T &= \exp(2\lambda_d\eta T) \\
\eta &= \frac{\log \rho T}{2\lambda_d T}.
\end{aligned}
$$

The above equation completes the proof. $\qquad\square$

**Part II: Error Bound Analysis**

**Lemma G.10.** *Consider $K$-epoch SGD with $N$ fresh data and learning rate $\eta$. Given a set of learning rate values $\Gamma$, and an excess risk estimate that satisfies $\bar{\mathcal{R}}(K, N; \eta) = M(K, N; \eta)(1 + o(1))$ when $\eta \in \Gamma$. Assume that $\eta' = \arg\min_\Gamma M(K, N; \eta)$ and $\eta^* \in \Gamma$. Then we have $\bar{\mathcal{R}}(K, N; \eta'(K, N)) = \bar{\mathcal{R}}^*(K, N)(1 + o(1))$.*

*Proof.* According to the optimality of $\eta^*$, it holds that

$$\bar{\mathcal{R}}^*(K,N) \leq \bar{\mathcal{R}}(K,N;\eta') = M(K,N;\eta)(1+o(1)).$$

Also, according to the optimality of $\eta'$, it holds that

$$M(K,N;\eta')(1+o(1)) \leq M(K,N;\eta^*)(1+o(1)) = \bar{\mathcal{R}}^*(K,N)$$

Combining the above two equations gives

$$\bar{\mathcal{R}}(K,N;\eta') = \bar{\mathcal{R}}^*(K,N)(1+o(1)).$$

$\square$

Combine the above two lemmas and we finish the whole proof.

### G.4.2 PROOF OF LEMMA 4.4, LARGE $K$

**Part I: Minimizing the Approximation of the Excess Risk**

**Lemma G.11.** *Under Assumption 4.1 and 4.3, we consider $K$-epoch SGD with $N$ fresh data and learning rate $\eta$ satisfying $\eta \in [\frac{\log T}{3\lambda_d T}, o\left(\frac{1}{N}\right)]$. Then when $K = \omega\left(\log N\right)$, the chosen learning rate $\eta' = \frac{\log \rho T}{2\lambda_d T} = \arg\min_{[\frac{\log T}{3\lambda_d T}, o\left(\frac{1}{N}\right)]} M(K,N;\eta)$.*

*Proof.* Given Lemma G.8, we compute the global minima of $M(K,N;\eta)$, we have $\eta' = \frac{\log T}{2\lambda_d T} + O\left(\frac{1}{T}\right) = \arg\min_{\eta\in\mathbb{R}} M(K,N;\eta)$, which lies in the regime $[\frac{\log T}{3\lambda_d T}, o\left(\frac{1}{N}\right)]$ when $N$ is sufficiently large. $\square$

**Part II: Error Bound Analysis** The proof of Lemma 4.4 concludes directly by applying Lemmas G.6, G.8, G.10 and G.11.

Combine the above two parts and we finish the whole proof.

### G.5 PROOF OF THEOREM 4.1

*Proof.* Notice from Lemma G.1 and Lemma G.4, we have that

$$\bar{\mathcal{R}}(K,N;\eta) = \underbrace{\frac{1}{2}\tilde{\theta}_d^2\lambda_d(1-\eta\lambda_d)^{2KN}\left(1+o(1)\right)}_{\widehat{\mathcal{R}}_1(K,N,\eta)}$$

$$+ \underbrace{\sum_{i=1}^d \frac{\sigma^2}{N}\frac{(1-\eta\lambda_i)^N}{1+(1-\eta\lambda_i)^N}}_{\widehat{\mathcal{R}}_2(K,N,\eta)}$$

$$+ \underbrace{\frac{\eta\sigma^2}{4}\mathrm{tr}(\boldsymbol{H}) - \frac{n_d\eta\sigma^2}{4}\lambda_d(1-\eta\lambda_d)^{2KN}\left(1+o(1)\right)}_{\widehat{\mathcal{R}}_3(K,N,\eta)} \text{ when } \eta \in \left[\omega\left(\frac{1}{T}\right), o\left(\frac{1}{T^{\frac{3}{4}}}\right)\right].$$

Next, we carefully analyze the magnitude of $\widehat{\mathcal{R}}_1(K,N,\eta)$, $\widehat{\mathcal{R}}_2(K,N,\eta)$, and $\widehat{\mathcal{R}}_3(K,N,\eta)$, and using these results, we can simplify the excess risk expression.

Now, We take $\eta = \frac{\log \rho T}{2\lambda_d T} = \frac{\log KN}{2\lambda_d KN} + O\left(\frac{1}{T}\right)$ in Lemma 4.4, then

$$(1-\lambda_d\eta)^{2KN} = \exp\left(2KN\log\left(1 - \frac{\log KN}{2KN} - O\left(\frac{1}{T}\right)\right)\right)$$

$$= \exp\left(-\log KN + O(1)\right)$$

$$= O\left(\frac{1}{T}\right).$$

Thus

$$\widehat{\mathcal{R}}_1(K, N, \eta) = \frac{1}{2}\tilde{\theta}_d^2 \lambda_d \left(1 - \lambda_d \eta\right)^{2KN}$$
$$= O\left(\frac{1}{T}\right),$$

and

$$\widehat{\mathcal{R}}_3(K, N, \eta) = \frac{\sigma^2 \text{tr}(\boldsymbol{H}) \log T}{8\lambda_d T} - \frac{n_d \sigma^2 \log T}{8\lambda_d T}\lambda_d \left(1 - \lambda_d \eta\right)^{2KN}(1 + o(1))$$
$$= \frac{\sigma^2 \text{tr}(\boldsymbol{H}) \log T}{8\lambda_d T}\left(1 + O\left(\frac{1}{T}\right)\right)$$
$$= \frac{\sigma^2 \text{tr}(\boldsymbol{H}) \log T}{8\lambda_d T}(1 + o(1))$$
$$= \omega(\widehat{\mathcal{R}}_1(K, N, \eta)).$$

Next, we discuss two scenarios where $K$ is relatively small and $K$ is relatively large, to be specific, $K = o(\log N)$ and $K = \omega(\log N)$.

**When $K = o(\log N)$,** We have

$$(1 - \lambda_i \eta)^N = \left(1 - \frac{\log KN}{2KN}\rho_i + O\left(\frac{1}{KN}\right)\right)^N$$
$$= \exp\left(N \log\left(1 - \frac{\log KN}{2KN}\rho_i + O\left(\frac{1}{KN}\right)\right)\right)$$
$$= \exp\left(-\frac{\log KN}{2K}\rho_i(1 + o(1))\right)$$
$$= o(1).$$

As a consequence,

$$\widehat{\mathcal{R}}_2(K, N, \eta) = \sum_{i=1}^{d} \frac{\sigma^2}{N}\frac{o(1)}{1 + o(1)}$$
$$= o\left(\frac{1}{N}\right).$$

Meanwhile,

$$\widehat{\mathcal{R}}_3(K, N, \eta) = O\left(\frac{\log KN}{KN}\right) = O\left(\frac{1}{N}\right) = \omega\left(\widehat{\mathcal{R}}_2(K, N, \eta)\right).$$

So

$$\bar{\mathcal{R}}^*(K, N) = \widehat{\mathcal{R}}(K, N; \eta)(1 + o(1)) = \frac{\sigma^2 \text{tr}(\boldsymbol{H}) \log T}{8\lambda_d T}(1 + o(1)).$$

**When $K = \omega(\log N)$,** we have

$$(1 - \lambda_i \eta)^N = \left(1 - \frac{\log KN}{2KN}\rho_i + O\left(\frac{1}{KN}\right)\right)^N$$
$$= \exp\left(N \log\left(1 - \frac{\log KN}{2KN}\rho_i + O\left(\frac{1}{KN}\right)\right)\right)$$
$$= \exp\left(-\frac{\log KN}{2K}\rho_i + O\left(\frac{1}{K}\right)\right) = \exp\left(o(1)\right)$$
$$= 1 + o(1).$$

So

$$\widehat{\mathcal{R}}_2(K, N, \eta) = \sum_{i=1}^{d} \frac{\sigma^2}{N} \frac{1 + o(1)}{2 + o(1)} = \frac{\sigma^2 d}{2N}(1 + o(1))$$

$$= O\left(\frac{1}{N}\right).$$

$$\widehat{\mathcal{R}}_3(K, N, \eta) = O\left(\frac{\log KN}{KN}\right) = o\left(\frac{1}{N}\right) = o\left(\widehat{\mathcal{R}}_2(K, N, \eta)\right).$$

As a result, we have

$$\bar{\mathcal{R}}^*(K, N) = \widehat{\mathcal{R}}(K, N; \eta)(1 + o(1)) = \frac{\sigma^2 d}{2N}(1 + o_N(1)).$$

### G.6    PROOF OF THEOREM 4.2

Now we establish the formulation of $E(K, N)$ by solving the equation $\bar{\mathcal{R}}^*(1, T') = \bar{\mathcal{R}}^*(K, N)$.

**When** $K = o(\log N)$,    solving $\bar{\mathcal{R}}^*(1, T') = \bar{\mathcal{R}}^*(K, N)$, we get

$$\frac{\sigma^2 \text{tr}(\boldsymbol{H}) \log T'}{8\lambda_d T'}(1 + o_{T'}(1)) = \frac{\sigma^2 \text{tr}(\boldsymbol{H}) \log T}{8\lambda_d T}(1 + o_T(1))$$

$$\frac{\log T'}{T'}(1 + o_{T'}(1)) = \frac{\log T}{T}(1 + o_T(1)). \tag{16}$$

According to the definition of the small $o$ notation, there exists a constant $\tilde{T}_0$ such that when $T > \tilde{T}_0$, the right hand side is smaller than $\max_{T' \in 1,2,3} \frac{\log T'}{T'}(1 + o_{T'}(1))$. So W.L.O.G, we could assume that $T' \geq 3$ in the following and use the fact that the function $\frac{\log x}{x}$ is monotonously decreasing when $x > 3$.

**Lemma G.12.** *Given $T'$ and $N$ satisfying Equation (16), it holds that $T' \approx T$ when $T > T_0$ for some constant $T_0$.*

*Proof.* Notice that there exists $T_1$ such that $|o_T(1)| < \frac{1}{2}$ when $T > T_1$, and $o_{T'}(1)$ is bounded. Furthermore, $o_{T'}(1) > -1$, because the left hand side is strictly greater than zero due to the fact that $\eta < \frac{1}{D^2}$. So when $T > T_1$, we have

$$c_4 \frac{\log T'}{T'} \leq \frac{3}{2} \frac{\log T}{T} \tag{17}$$

$$c_5 \frac{\log T'}{T'} \geq \frac{1}{2} \frac{\log T}{T} \tag{18}$$

for two constants $c_4 \leq 1 \leq c_5$. We claim that $T' \geq \frac{c_4}{3} T =: \alpha T$ when $T \geq \frac{1}{\alpha^2}$; otherwise,

$$c_4 \frac{\log T'}{T'} \geq c_4 \frac{\log \alpha T}{\alpha T}$$

$$= \frac{3 \log \alpha T}{T}$$

$$\geq \frac{3 \log T}{2T} \quad \text{when} \quad T \geq \frac{1}{\alpha^2},$$

which contradicts Equation (17). We also have $T' \leq 3c_5 T =: \beta T$ when $T \geq \beta^2$ by a similar argument; otherwise,

$$c_5 \frac{\log T'}{T'} \leq c_5 \frac{\log \beta T}{\beta T}$$

$$= \frac{\log \beta T}{3T}$$

$$\leq \frac{\log T}{2T} \quad \text{when} \quad T \geq \beta^2,$$

which contradicts Equation (18). So $T' \approx T$ when $T \geq \min(T_1, \frac{1}{\alpha^2}, \beta^2, \tilde{T}_0) = T_0$.

Next, we prove the first part in Theorem 4.2, which is $\mathbb{E}(K, N) = K(1 + o(1))$ when $K = o(\log N)$. We define $F(T) = \frac{\log T}{T}, \delta(T) = |o_T(1)|$, and $\epsilon(T') = |o_{T'}(1)|$, so

$$F(T')(1 - \epsilon(T')) \leq F(T)(1 + \delta(T))$$
$$F(T')(1 + \epsilon(T')) \geq F(T)(1 - \delta(T))$$

Consequently, we have

$$-F(T)\delta(T) - F(T')\epsilon(T') \leq F(T') - F(T) \leq F(T)\delta(T) + F(T')\epsilon(T'). \tag{19}$$

So due to the convexity of $F(T)$,

$$-\frac{\log T - 1}{T^2}(T' - T) \leq F'(T)(T' - T) \leq F(T') - F(T) \leq F(T)\delta(T) + F(T')\epsilon(T') = \frac{\log T}{T}|o(1)|.$$

Thus we have

$$T' \geq T(1 - o(1)).$$

The above equation completes the proof. □

Combining Equation (16) and Lemma G.12 gets

$$-\frac{\log T - 1}{T^2}(T - T') \approx -\frac{\log T' - 1}{T'^2}(T - T'). \tag{20}$$

Further using Equation (19),

$$F'(T')(T - T') \leq F(T) - F(T') \leq F(T)\delta(T) + F(T')\epsilon(T') \tag{21}$$

Combining Equation (20) and Equation (21) gives

$$T' \leq T(1 + o(1)).$$

Substituding the definition of $E(K, N)$ and we get the first part in Theorem 4.2.

**When $K = \omega(\log N)$,** solving $\bar{\mathcal{R}}^*(1, T') = \bar{\mathcal{R}}^*(K, N)$, we get

$$\frac{\sigma^2 \text{tr}(\boldsymbol{H}) \log T'}{8\lambda_d T'}(1 + o_{T'}(1)) = \frac{\sigma^2 d}{2N}(1 + o_N(1)). \tag{22}$$

There exists a constant $\tilde{N}_0$ such that when $N > \tilde{N}_0$, the right hand side is smaller than the minimal value of $\bar{\mathcal{R}}^*(1, T')$ when $T'$ is finite, that is, $\min_{T' \in 1,2,3} \frac{\sigma^2 \text{tr}(\boldsymbol{H}) \log T'}{8\lambda_d T'}(1 + o_{T'}(1))$. So W.L.O.G, we could assume that $T' \geq 3$ in the following and use the fact that the function $\frac{\log x}{x}$ is monotonously decreasing when $x > 3$.

Now we provide a lemma to give a loose bound of $T'$ fisrt, and then we apply the lemma to get the formula of $E(K, N)$.

**Lemma G.13.** *Given $T'$ and $N$ satisfying Equation (22). It holds that $N \leq T' \leq N^{\frac{3}{2}}$ when $N \geq N_0$ for some constant $N_0$.*

*Proof.* Notice that there exists $N_1$ such that $|o_N(1)| < \frac{1}{2}$ when $N > N_1$, and $o_{T'}(1)$ is bounded. Furthermore, $o_{T'}(1) > -1$, because the left hand side is strictly greater than zero due to the fact that $\eta < \frac{1}{D^2}$. So when $N > N_1$, for the left side in Equation (22), we have

$$c_6 \frac{\log T'}{T'} \leq \frac{\sigma^2 \text{tr}(\boldsymbol{H}) \log T'}{8\lambda_d T'}(1 + o_{T'}(1)) \leq c_7 \frac{\log T'}{T'},$$

where $c_6 < c_7$ are two positive constants. And for the right side,

$$\frac{c_8}{N} \leq \frac{\sigma^2 d}{2N}(1 + o_N(1)) \leq \frac{c_9}{N},$$

where $c_8 < c_9$ are two positive constants. Then we prove that $T' \geq N$ when $N \geq \max\left(e^{\frac{c_9}{c_6}}, 3\right)$. Otherwise, we have

$$\frac{\sigma^2 \text{tr}(\boldsymbol{H}) \log T'}{8\lambda_d T'}(1 + o_{T'}(1)) \geq c_6 \frac{\log T'}{T'} \geq c_6 \frac{\log N}{N} \geq \frac{c_9}{N} \geq \frac{\sigma^2 d}{2N}(1 + o_N(1)),$$

which is a contradiction. Then we prove that $T' \leq N^{\frac{3}{2}}$ when $N \geq \left(\frac{c_{10}}{c_8}\right)^4$ for some constant $c_{10}$. Otherwise, we have

$$\frac{\sigma^2 \text{tr}(\boldsymbol{H}) \log T'}{8\lambda_d T'}(1 + o_{T'}(1)) \leq c_7 \frac{\log T'}{T'} \leq c_7 \frac{\log N^{\frac{3}{2}}}{N^{\frac{3}{2}}} = \frac{3c_7}{2} \frac{\log N}{N^{\frac{3}{2}}} \leq \frac{c_{10}}{N^{\frac{5}{4}}} \leq \frac{c_8}{N} \leq \frac{\sigma^2 d}{2N}(1 + o_N(1)),$$

which is another contradiction. The third inequality uses the fact that $\frac{\log N}{N^{\frac{1}{4}}}$ is bounded. We take $N_0 = \max\left(N_1, e^{\frac{c_9}{c_6}}, \left(\frac{c_{10}}{c_8}\right)^4, \tilde{N}_0\right)$ and we prove the claim. $\qquad\square$

Combining Equation (22) and Lemma G.13 gives $T' = \Theta\left(N \log T'\right) = \Theta(N \log N)$, and

$$\log T' = \log N + \log \log N + \Theta(1). \tag{23}$$

Again, combining Equation (23) and Equation (22), and we get

$$T' = \frac{\text{tr}(\boldsymbol{H}) N \log T'}{4\lambda_d d}(1 + o_N(1)) = \frac{\text{tr}(\boldsymbol{H}) N \log N}{4\lambda_d d}(1 + o_N(1)),$$

and

$$E(K, N) = \frac{T'}{N} = \frac{\text{tr}(\boldsymbol{H}) \log N}{4\lambda_d d}(1 + o_N(1))$$

as a direct corollary.

The above equation immediately finish the proof. $\qquad\square$

## H    PROOF OUTLINE FOR THE SOLVABLE CASE WITH ZIPF-DISTRIBUTED DATA

In this section, we give the proof sketch of Lemma I.1 and Theorems 5.2 and 5.3. Lemma I.1 gives a general expression of the excess risk, Theorem 5.2 and Theorem 5.3 characterise the behavior of $E(K, N)$ respectively under power spectrum and logarithm power spectrum assumption. Their proof outlines are given separately as follows.

**1. Proof sketch of Lemma I.1.**    We exploit the properties that the sequential updates are commutative and all finite-order moments of data are computable, and we obtain the result through a direct calculation.

**2. Proof sketch of Theorem 5.2 and Theorem 5.3.**    For Theorem 5.2, we consider two cases when $K$ is relatively small and $K$ is relatively large. As a special case, one-pass scenario belongs to the small-$K$ case. We first derive matching upper bounds and lower bounds for high-dimensional cases for both the two regimes. The core of the proof lies in determining $E(K, N)$ by solving $\mathbb{ER}(\boldsymbol{w}_{T'}) = \mathbb{ER}(\boldsymbol{w}_{K,N})$, which requires an asymptotic analysis. We tackle this issue with two steps. First we prove a loose bound for $T'$ for $N$ beyond a threshold, then we refine the obtained results and utilize the convexity of loss approximation to derive more precise estimates. The proof of Theorem 5.3 is similar to that of Theorem 5.2.

## I    PROOF OF MAIN RESULTS FOR THE SOLVABLE CASE WITH ZIPF-DISTRIBUTED DATA

Similar to the proof insights in Section 4, the first move to get the formula of the effective reuse rate is to get an accurate proxy of the excess risk. Here, leveraging the simplicity of the setting, we can derive a general closed formula for the excess risk.

**Lemma I.1.** *Under Assumption 5.1, the excesss risk for $K$-epoch training over $N$ fresh data , with learning rate $\eta$ can be given by*

$$\bar{\mathcal{R}}(K, N; \eta) = \frac{1}{2} \left\langle \boldsymbol{P}\Lambda, \left( \boldsymbol{I} - \boldsymbol{P} + \boldsymbol{P} \left( \boldsymbol{I} - \eta\Lambda \right)^{2K} \right)^N \right\rangle,$$

*where the expectation is over the randomness of $\boldsymbol{w}^*$ and training datasets $\{\boldsymbol{x}_i, y_i\}_{i=0}^{N-1}$.*

The above lemma states that we can explicitly write out the exact expression for the excess risk. From the above expression for the excess risk, we can observe that, in the absence of label noise interference, and under the condition that the absolute values of all elements of the diagonal matrix $\boldsymbol{I} - \eta\Lambda$ are less than 1, the optimal learning rate can be of the constant order. Therefore, in the subsequent study of the effective reuse rate, we consider using the same learning rate $\eta = \Theta(1)$ for both multi-epoch and one-pass SGD.

It is worth noting that here we are actually describing a more general problem setting than the Zipf law, as we only impose constraints on the power spectrum of the Hessian matrix $\boldsymbol{H}$. In contrast, the probability matrix $\boldsymbol{P}$ can follow Zipf's law or any other law. In the remainder of this section, we first consider the classic Zipf's law setting, where $\boldsymbol{P}$ follows a power law, and the data matrix $\Lambda$ also follows a power law, which is consistent with the previous power law analysis. In Section 5.2, we explore the case where $\boldsymbol{P}$ follows a log-power spectrum (Lin et al., 2024), and investigate the impact of changing the spectrum's properties on the resulting effective reuse rate formula.

## I.1 A CLOSED FORMULA FOR THE EXCESS RISK: PROOF OF LEMMA I.1

We first write out the update of parameter after $K$ epochs

$$\boldsymbol{\theta}_{KN} = \boldsymbol{A}^{(K)} \cdots \boldsymbol{A}^{(1)} \boldsymbol{\theta}_0 = \prod_{l=K}^{1} \boldsymbol{A}^{(l)} \boldsymbol{\theta}_0$$
$$= \left( \boldsymbol{I} - \eta \boldsymbol{x}_{N-1} \boldsymbol{x}_{N-1}^\top \right)^K \cdots \left( \boldsymbol{I} - \eta \boldsymbol{x}_0 \boldsymbol{x}_0^\top \right)^K \boldsymbol{\theta}_0.$$

Then we get the excess risk expression as

$$\bar{\mathcal{R}}(K, N; \eta) = \mathbb{E} \frac{1}{2} \boldsymbol{\theta}_{K,N}^\top \boldsymbol{H} \boldsymbol{\theta}_{K,N}$$
$$= \mathbb{E} \frac{1}{2} \boldsymbol{\theta}_0^T \boldsymbol{P}\Lambda \left( \boldsymbol{I} - \eta \boldsymbol{x}_{N-1} \boldsymbol{x}_{N-1}^\top \right)^{2K} \cdots \left( \boldsymbol{I} - \eta \boldsymbol{x}_0 \boldsymbol{x}_0^\top \right)^{2K} \boldsymbol{\theta}_0.$$

Assumption 5.1 gives

$$\bar{\mathcal{R}}(K, N; \eta) = \mathbb{E} \frac{1}{2} \left\langle \boldsymbol{\theta}_0 \boldsymbol{\theta}_0^T, \boldsymbol{P}\Lambda \left( \boldsymbol{I} - \eta \boldsymbol{x}_{N-1} \boldsymbol{x}_{N-1}^\top \right)^{2K} \cdots \left( \boldsymbol{I} - \eta \boldsymbol{x}_0 \boldsymbol{x}_0^\top \right)^{2K} \right\rangle$$
$$= \frac{1}{2} \left\langle \boldsymbol{I}, \boldsymbol{P}\Lambda \left( \mathbb{E} \left( \boldsymbol{I} - \eta \boldsymbol{x} \boldsymbol{x}^\top \right)^{2K} \right)^N \right\rangle.$$

Direct calculation gives

$$\mathbb{E} \left( \boldsymbol{x} \boldsymbol{x}^\top \right)^j = \sum_{i=1}^{d} \mu_i^{2j-2} p_i \mu_i^2 e_i e_i^\top = \boldsymbol{P}\Lambda^j,$$

and

$$\mathbb{E} \left[ \left( \boldsymbol{I} - \eta \boldsymbol{x} \boldsymbol{x}^\top \right)^{2K} \right] = \boldsymbol{I} + \sum_{j=1}^{2K} \binom{2K}{j} (-1)^j \eta^j \boldsymbol{P}\Lambda^j = \boldsymbol{I} - \boldsymbol{P} + \boldsymbol{P}(\boldsymbol{I} - \eta\Lambda)^{2K}.$$

Then we can write out the excess risk as

$$\bar{\mathcal{R}}(K, N; \eta) = \frac{1}{2} \left\langle \boldsymbol{P}\Lambda, \left( \boldsymbol{I} - \boldsymbol{P} + \boldsymbol{P}(\boldsymbol{I} - \eta\Lambda)^{2K} \right)^N \right\rangle.$$

The above equation completes the proof.

## I.2 SCALING LAWS FOR POWER-LAW SPECTRUM: PROOF OF THEOREM 5.1

Before we begin our main part of the proof, note that for all $\eta = \Theta(1)$ and $\eta \leq 2$, there exists $d_1 = \Theta(1) > 0$ such that $1 - \frac{\eta}{i^b} > 0$ when $i > d_1$. Then we divide the expected excess risk into two parts:

$$
\begin{aligned}
\bar{\mathcal{R}}(K, N; \eta) &= \frac{1}{2} \sum_{i=1}^{d} \frac{c}{i^a} \left( 1 - \frac{c}{i^{a-b}} \left( 1 - \left( 1 - \frac{\eta}{i^b} \right)^{2K} \right) \right)^N \\
&= \underbrace{\frac{1}{2} \sum_{i=1}^{d_1} \frac{c}{i^a} \left( 1 - \frac{c}{i^{a-b}} \left( 1 - \left( 1 - \frac{\eta}{i^b} \right)^{2K} \right) \right)^N}_{S_1(K,N;\eta)} \\
&+ \underbrace{\frac{1}{2} \sum_{i=d_1+1}^{d} \frac{c}{i^a} \left( 1 - \frac{c}{i^{a-b}} \left( 1 - \left( 1 - \frac{\eta}{i^b} \right)^{2K} \right) \right)^N}_{S_2(K,N;\eta)}.
\end{aligned}
$$

The intuition behind our proof here is quite similar to what we do in Appendix G.5. We first separately simplify the expression of the excess risk when $K = o(N^{\frac{b}{a-b}})$ and $K = \omega(N^{\frac{b}{a-b}})$. The proofs for both the small-$K$ and large-$K$ regimes proceed in parallel. We first control $S_2(K, N; \eta)$ over a broad range of learning rates and identify a near-optimal $\eta'$ for which $S_1$ is negligible compared to $S_2$. This allows us to approximate $\bar{\mathcal{R}}^*(K, N)$ via $\bar{\mathcal{R}}(K, N; \eta')$ and $S_2(K, N; \eta^*)$.

### I.2.1 PROOF OF THEOREM 5.1: SMALL-$K$ CASE

**The Expected Excess Risk Approximation.**

**Lemma I.2.** *Suppose the assumptions in Theorem 5.2 hold. When $K = o(N^{\frac{b}{a-b}})$ and $\eta = \Theta(1)$, we define the estimator of $S_2(K, N; \eta)$ as*

$$
\tilde{S}_2(K, N; \eta) := \frac{1}{2} \sum_{i=d_1+1}^{d} \frac{c}{i^a} e^{\frac{-2KNc\eta}{i^a}}.
$$

*Then we have $S_2(K, N; \eta) = \tilde{S}_2(K, N; \eta)(1 + o(1))$, and $\tilde{S}_2(K, N; \eta) \asymp \frac{1}{(KN)^{\frac{a-1}{a}}}$.*

*Proof.* By the fact that $K = o(N^{\frac{b}{a-b}})$, there exists a constant $N_2$ such that when $N \geq N_2$, $K \leq N^{\frac{b}{a-b}}$. And we define $F(x) := \frac{c}{x^a} \left( 1 - \frac{c}{x^{a-b}} \left( 1 - \left( 1 - \frac{\eta}{x^b} \right)^{2K} \right) \right)^N$. Direct observation gives us that under Assumption 5.2, $\bar{\mathcal{R}}(K, N; \eta) \propto \sum_{i=1}^{d} F(i)$. Next we take the derivative of $F$ and analyze its maximizer.

$$
\begin{aligned}
F'(x) &= -\frac{ac}{x^{a+1}} \left( 1 - \frac{c}{x^{a-b}} + \frac{c}{x^{a-b}} \left( 1 - \frac{\eta}{x^b} \right)^{2K} \right)^N \\
&+ \frac{cN}{x^a} \left( 1 - \frac{c}{x^{a-b}} + \frac{c}{x^{a-b}} \left( 1 - \frac{\eta}{x^b} \right)^{2K} \right)^{N-1} \cdot \Phi(x) \\
&= \frac{c}{x^a} \left( 1 - \frac{c}{x^{a-b}} + \frac{c}{x^{a-b}} \left( 1 - \frac{\eta}{x^b} \right)^{2K} \right)^{N-1} \\
&\quad \left( -\frac{a}{x} \left( 1 - \frac{c}{x^{a-b}} + \frac{c}{x^{a-b}} \left( 1 - \frac{\eta}{x^b} \right)^{2K} \right) + N\Phi(x) \right) \\
&= \frac{c}{x^{2a-b+1}} \left( 1 - \frac{c}{x^{a-b}} + \frac{c}{x^{a-b}} \left( 1 - \frac{\eta}{x^b} \right)^{2K} \right)^{N-1} \cdot G(x).
\end{aligned}
$$

where we define

$$G(x) := -a \left( x^{a-b} - c + c \left( 1 - \frac{\eta}{x^b} \right)^{2K} \right)$$
$$+ N \left( (a-b)c - (a-b)c \left( 1 - \frac{\eta}{x^b} \right)^{2K} + \frac{2cKb\eta}{x^b} \left( 1 - \frac{\eta}{x^b} \right)^{2K-1} \right),$$

and

$$\Phi(x) := \left( \frac{(a-b)c}{x^{a-b+1}} - \frac{(a-b)c}{x^{a-b+1}} \left( 1 - \frac{\eta}{x^b} \right)^{2K} + \frac{2cKb\eta}{x^{a+1}} \left( 1 - \frac{\eta}{x^b} \right)^{2K-1} \right).$$

We denote the maximizer of $F(x)$ by $x_0$, so $G(x_0) = 0$. We claim that:

when $N \geq N_2$, $x_0 \geq \min \left( \left( \frac{KN(a-b)c\eta}{2a} \right)^{\frac{1}{a}}, 6^{\frac{1}{b}} (KN)^{\frac{1}{a}} \right) =: x_1$.

*Proof of the claim.* Notice that when $N \geq N_2$,

$$\frac{\eta}{x^b} \leq \frac{1}{6(KN)^{\frac{b}{a}}} \leq \frac{1}{6K}.$$

We assume that the claim is wrong, then

$$G(x_0) \geq N \left( (a-b)c - (a-b)c \left( 1 - \frac{\eta}{x^b} \right)^{2K} \right) - ax^{a-b}$$
$$\geq \frac{KN(a-b)c\eta - ax^a}{x^b}$$
$$\geq \frac{KN(a-b)c\eta}{2x_1^b}$$
$$> 0,$$

which is a contradiction. The third inequality comes from Lemma J.3. $\square$

So $x_0 = \Omega \left( (KN)^{\frac{1}{a}} \right)$. Further pluging this into $G(x_0) = 0$ that

$$G(x_0) = -ax_0^{a-b}(1+o(1)) + N \left( \frac{2K(a-b)c\eta}{x_0^b}(1+o(1)) + \frac{2K(a-b)c\eta}{x_0^b}(1+o(1)) \right) = 0.$$

gives

$$x_0 = \Theta \left( (KN)^{\frac{1}{a}} \right), \quad F(x_0) = \Theta \left( \frac{1}{KN} \right).$$

Then we have

$$S_2(K,N;\eta) = \frac{1}{2} \sum_{i=d_1+1}^{K^{\frac{0.5}{a+0.5b}} (KN)^{\frac{1}{a+0.5b}}} \frac{c}{i^a} \left( 1 - \frac{c}{i^{a-b}} + \frac{c}{i^{a-b}} \left( 1 - \frac{\eta}{i^b} \right)^{2K} \right)^N$$
$$+ \frac{1}{2} \sum_{K^{\frac{0.5}{a+0.5b}} (KN)^{\frac{1}{a+0.5b}}+1}^{d} \frac{c}{i^a} \left( 1 - \frac{c}{i^{a-b}} + \frac{c}{i^{a-b}} \left( 1 - \frac{\eta}{i^b} \right)^{2K} \right)^N$$
$$:= J_1 + J_2.$$

Furthermore, we have

$$J_1 \lesssim K^{\frac{0.5}{a+0.5b}} (KN)^{\frac{1}{a+0.5b}} F(x_0) \lesssim \frac{K^{\frac{0.5}{a+0.5b}} (KN)^{\frac{1}{a+0.5b}}}{KN},$$

and

$$J_2 = \frac{1}{2} \sum_{i=K^{\frac{0.5}{a+0.5b}}(KN)^{\frac{1}{a+0.5b}}+1}^{d} \frac{c}{i^a}\left(1 - \frac{c}{i^{a-b}} + \frac{c}{i^{a-b}}\left(1-\frac{\eta}{i^b}\right)^{2K}\right)^N$$

$$= \frac{1}{2} \sum_{i=K^{\frac{0.5}{a+0.5b}}(KN)^{\frac{1}{a+0.5b}}+1}^{d} \frac{c}{i^a}\left(1 - \frac{2Kc\eta}{i^a} + O\left(\frac{K^2}{i^{a+b}}\right)\right)^N$$

$$= \frac{1}{2} \sum_{K^{\frac{0.5}{a+0.5b}}(KN)^{\frac{1}{a+0.5b}}+1}^{d} \frac{c}{i^a}e^{N\log\left(1-\frac{2Kc\eta}{i^a}+O\left(\frac{K^2}{i^{a+b}}\right)\right)}$$

$$= \frac{1}{2} \sum_{i=K^{\frac{0.5}{a+0.5b}}(KN)^{\frac{1}{a+0.5b}}+1}^{d} \frac{c}{i^a}e^{\frac{-2KNc\eta}{i^a}+O\left(\frac{K^2N}{i^{a+b}}\right)}$$

$$= \frac{1}{2} \sum_{i=K^{\frac{0.5}{a+0.5b}}(KN)^{\frac{1}{a+0.5b}}+1}^{d} \frac{c}{i^a}e^{\frac{-2KNc\eta}{i^a}}(1+o(1)).$$

We define $K_1(x) = \frac{c}{x^a}e^{\frac{-2KNc\eta}{x^a}}$. We can derive that $\arg\max K_1(x) = \Theta\left((KN)^{\frac{1}{a}}\right)$, and $\max K_1(x) = \Theta\left(\frac{1}{KN}\right)$. So when $d \geq 3(KN)^{\frac{1}{a}}$, we have

$$J_2 \geq \frac{1}{2}\sum_{i=(KN)^{\frac{1}{a}}}^{3(KN)^{\frac{1}{a}}} \frac{c}{i^a}e^{\frac{-2KNc\eta}{i^a}}(1+o(1))$$

$$\gtrsim (KN)^{\frac{1}{a}} \times \frac{ce^{-2c\eta}}{KN} \gtrsim \frac{(KN)^{\frac{1}{a}}}{KN}.$$

We can verify that $J_1 = o(J_2)$ as a direct consequence. We define

$$\tilde{S}_2(K,N;\eta) = \frac{1}{2}\sum_{i=d_1+1}^{d}\frac{c}{i^a}e^{\frac{-2KNc\eta}{i^a}}$$

$$= \frac{1}{2}\sum_{i=d_1+1}^{K^{\frac{0.5}{a+0.5b}}(KN)^{\frac{1}{a+0.5b}}}\frac{c}{i^a}e^{\frac{-2KNc\eta}{i^a}} + \frac{1}{2}\sum_{i=K^{\frac{0.5}{a+0.5b}}(KN)^{\frac{1}{a+0.5b}}+1}^{d}\frac{c}{i^a}e^{\frac{-2KNc\eta}{i^a}}$$

$$:= \tilde{J}_1 + \tilde{J}_2.$$

We have $J_2 = \tilde{J}_2(1+o(1))$, and

$$\tilde{J}_1 \leq K^{\frac{0.5}{a+0.5b}}(KN)^{\frac{1}{a+0.5b}} \times \max K_1(x) \lesssim \frac{K^{\frac{0.5}{a+0.5b}}(KN)^{\frac{1}{a+0.5b}}}{KN} = o(\tilde{J}_2).$$

So $S_2(K,N;\eta) = \tilde{S}_2(K,N;\eta)(1+o(1))$.

The matching upper and lower bounds for $\tilde{S}_2(K,N;\eta)$ comes directly from Lemma J.7. $\qquad\square$

By combining the expression of $\tilde{S}_2(K,N;\eta)$ with Lemma J.7, we get another lemma:

**Lemma I.3.** *Suppose the assumptions in Theorem 5.2 hold, and the expression of $\tilde{S}_2(K,N;\eta)$ is given in Lemma I.2. Then we have $\frac{\partial}{\partial\eta}\tilde{S}_2(K,N;\eta) \approx -\frac{1}{(KN)^{\frac{a-1}{a}}}$.*

*Proof.*

$$\frac{\partial}{\partial \eta} \tilde{S}_2(K, N; \eta) = -KN \sum_{i=d_1+1}^{d} \frac{c}{i^{2a}} e^{\frac{-2KNc\eta}{i^a}}$$

$$\overline{\approx} -\frac{1}{(KN)^{\frac{a-1}{a}}},$$

where the second line comes from Lemma J.7. $\qquad \square$

**Lemma I.4.** *Suppose the assumptions in Theorem 5.2 hold, and the expression of $\tilde{S}_2(K, N; \eta)$ is given in Lemma I.2. Consider two learning rate options $\eta, \eta' = \Theta(1)$ that satisfy $\eta - \eta' = o(1)$. Then we have $\tilde{S}_2(K, N; \eta) = \tilde{S}_2(K, N; \eta')(1 + o(1))$.*

*Proof.*

$$\left| \tilde{S}_2(K, N; \eta) - \tilde{S}_2(K, N; \eta') \right| = \left| \frac{\partial}{\partial \eta} \tilde{S}_2(K, N; \tilde{\eta}) \right| |(\eta - \eta')|$$

$$\overline{\approx} \frac{1}{(KN)^{\frac{a-1}{a}}} |(\eta - \eta')|$$

$$= \tilde{S}_2(K, N; \eta') o(1),$$

where $\tilde{\eta} \in [\min(\eta, \eta'), \max(\eta, \eta')] = \Theta(1)$, and the first line comes from Lagrange's Mean Value Theorem. The secome line comes from Lemma I.3, and the last line comes from Lemma I.2. $\qquad \square$

**The Range of Optimal Learning Rate.** First, take $\eta' = 2 - \frac{(a-1)d_1^{a-b}}{ac} \frac{\log KN}{KN}$, and we have

$$S_1(K, N; \eta') \leq \frac{d_1 c}{2} \left( 1 - \frac{c}{d_1^{a-b}} + \frac{c}{d_1^{a-b}} \left( 1 - \frac{(a-1)d_1^{a-b}}{ac} \frac{\log KN}{KN} \right)^{2K} \right)^N.$$

By a Taylor expansion argument, we have

$$S_1(K, N; \eta') = \frac{d_1 c}{2} \left( 1 - \frac{2Kc}{d_1^{a-b}} \times \frac{(a-1)d_1^{a-b}}{ac} \frac{\log KN}{KN}(1 + o(1)) \right)^N$$

$$= \frac{d_1 c}{2} \left( 1 - \frac{2(a-1)}{a} \frac{\log KN}{N}(1 + o(1)) \right)^N$$

$$= \frac{d_1 c}{2} e^{N \log\left(1 - \frac{2(a-1)}{a} \frac{\log KN}{N}(1 + o(1))\right)}$$

$$\overline{\approx} \frac{1}{(KN)^{\frac{2(a-1)}{a}}} = o(S_2(K, N; \eta')),$$

where the last inequality comes from Lemma I.2. Then we have

$$\bar{\mathcal{R}}(K, N; \eta') = S_1(K, N; \eta') + S_2(K, N; \eta')$$

$$= \tilde{S}_2(K, N; \eta')(1 + o(1))$$

$$= \tilde{S}_2(K, N; 2)(1 + o(1))$$

$$= \left( \frac{1}{2} \sum_{i=d_1+1}^{d} \frac{c}{i^a} e^{\frac{-4KNc}{i^a}} \right)(1 + o(1)).$$

Then we prove that $\eta^* \in [2 - o(1), 2]$. We prove by contradiction, and assume that there exist a constant $\epsilon > 0$ and a sequence $(N_i)_{i=1}^{\infty} \to \infty$ such that $\eta^*(N_i) \leq 2 - \epsilon$ for all $i \geq 1$. As we only

analyze with respect to the sequence $(N_i)_{i=1}^\infty$, without loss of generality, we take $(N_i)_{i=1}^\infty = \mathbb{N}$. By Lemma I.2 and the convexity of $\tilde{S}_2(K, N; \eta)$ with respect to $\eta$, we have

$$\bar{\mathcal{R}}^*(K, N) \geq S_2(K, N; \eta^*) = \tilde{S}_2(K, N; \eta^*)(1 + o(1))$$

$$\geq \left[ \tilde{S}_2(K, N; 2) + \epsilon \frac{\partial}{\partial \eta} \tilde{S}_2(K, N; 2) \right] (1 + o(1)) > \bar{\mathcal{R}}(K, N; \eta')$$

when $N$ is sufficiently large, which is a contradiction. So

$$\bar{\mathcal{R}}^*(K, N) = S_1(K, N; \eta^*) + S_2(K, N; \eta^*)$$

$$= S_1(K, N; \eta^*) + \tilde{S}_2(K, N; \eta^*)(1 + o(1))$$

$$= S_1(K, N; \eta^*) + \tilde{S}_2(K, N; 2)(1 + o(1)) \leq \bar{\mathcal{R}}(K, N; \eta').$$

Thus, $S_1(K, N; \eta^*) = o\left( \tilde{S}_2(K, N; 2) \right)$, and $\bar{\mathcal{R}}^*(K, N) = \tilde{S}_2(K, N; 2)(1 + o(1))$.

By Lemma I.2 and Lemma J.7, there exist two constants $C_1$ and $C_2$ such that $\bar{\mathcal{R}}^*(K, N) \leq \frac{C_1}{(KN)^{\frac{a-1}{a}}}$ and $\bar{\mathcal{R}}^*(K, N) \geq \frac{C_2}{(KN)^{\frac{a-1}{a}}}$ when the condition $d = \Omega(T^{\frac{1}{a}})$ holds. For one-pass case, by Lemma I.2 and Lemma J.7, we have

$$\bar{\mathcal{R}}^*(1, T') = \bar{\mathcal{R}}(1, T'; \eta^*(1, T')|_{d=d})$$

$$\leq \bar{\mathcal{R}}(1, T'; \eta^*(1, T')|_{d=\infty})$$

$$= \bar{\mathcal{R}}^*(1, T')\big|_{d=\infty} = \frac{1}{2} \sum_{i=d_1+1}^\infty \frac{c}{i^a} e^{-\frac{4KNc}{i^a}} (1 + o(1)) \leq \frac{C_3}{T'^{\frac{a-1}{a}}} \quad (24)$$

and

$$\bar{\mathcal{R}}^*(1, T') = \frac{1}{2} \sum_{i=d_1+1}^d \frac{c}{i^a} e^{-\frac{4KNc}{i^a}} (1 + o(1)) \geq \frac{C_4}{T'^{\frac{a-1}{a}}} \quad \text{when } d = \Omega\left( T'^{\frac{1}{a}} \right). \quad (25)$$

### I.2.2 PROOF OF THEOREM 5.1: LARGE-$K$ CASE

**The Expected Excess Risk Approximation.**

**Lemma I.5.** *Suppose the assumptions in Theorem 5.2 hold. When $K = \omega(N^{\frac{b}{a-b}})$ and $\eta = \Theta(1)$, we have $S_2(K, N; \eta) \asymp \frac{1}{N^{\frac{a-1}{a-b}}}$.*

*Proof.* There exists $N_3$ such that when $N \geq N_3$, we have $K \geq N^{\frac{b}{a-b}}$. Then when $d \geq 3(KN)^{\frac{1}{a}} \geq 3N^{\frac{1}{a-b}}$, we give the lower bound of the loss:

$$S_2(K, N; \eta) \geq \frac{1}{2} \sum_{i=N^{\frac{1}{a-b}}}^{3N^{\frac{1}{a-b}}} \frac{c}{i^a} \left( 1 - \frac{c}{i^{a-b}} \right)^N$$

$$\geq \frac{1}{2} \frac{2N^{\frac{1}{a-b}}}{(3N^{\frac{1}{a-b}})^a} (1 - \frac{c}{N})^N$$

$$\gtrsim \frac{1}{N^{\frac{a-1}{a-b}}}.$$

Then we derive the upper bound of the loss:

$$S_2(K, N; \eta) \leq \frac{1}{2} \sum_{i=1}^\infty \frac{c}{i^a} \left( 1 - \frac{c}{i^{a-b}} + \frac{c}{i^{a-b}} \left( 1 - \frac{\eta}{i^b} \right)^{2K} \right)^N$$

$$\leq \frac{1}{2} \sum_{i=1}^{N^{\frac{1}{a-b}}} \frac{c}{i^a} \left( 1 - \frac{c}{i^{a-b}} + \frac{c}{i^{a-b}} \left( 1 - \frac{\eta}{i^b} \right)^{2K} \right)^N + \frac{1}{2} \sum_{i=N^{\frac{1}{a-b}}+1}^\infty \frac{c}{i^a}.$$

When $K = \omega(N^{\frac{b}{a-b}})$ and $i \leq N^{\frac{1}{a-b}}$,

$$
\left(1 - \frac{\eta}{i^b}\right)^{2K} \leq \left(1 - \frac{\eta}{N^{\frac{b}{a-b}}}\right)^{2K} = e^{2K \log\left(1 - \frac{\eta}{N^{\frac{b}{a-b}}}\right)}
$$
$$
\leq e^{-2K \frac{\eta}{N^{\frac{b}{a-b}}}} = o(1).
$$

Then there exists $N_4$ such that $\left(1 - \frac{\eta}{i^b}\right)^{2K} \leq \frac{1}{2}$ when $N \geq N_4$. So when $N \geq \max(N_3, N_4)$, we have

$$
S_2(K, N; \eta) \leq \frac{1}{2} \sum_{i=1}^{N^{\frac{1}{a-b}}} \frac{c}{i^a} \left(1 - \frac{c}{2i^{a-b}}\right)^N + \frac{1}{2} \sum_{i=N^{\frac{1}{a-b}}+1}^{\infty} \frac{c}{i^a}.
$$

One can derive that $\max \frac{c}{i^a}\left(1 - \frac{c}{2i^{a-b}}\right)^N = \Theta\left(\frac{1}{N^{\frac{a}{a-b}}}\right)$. So

$$
\bar{\mathcal{R}}^*(K, N) \lesssim \frac{1}{N^{\frac{a-1}{a-b}}} + \frac{1}{N^{\frac{a-1}{a-b}}}
$$
$$
\lesssim \frac{1}{N^{\frac{a-1}{a-b}}}.
$$

And we complete the proof. $\qquad\square$

**The Range of Optimal Learning Rate.** First, take $\eta' = 1.5$, and we have

$$
S_1(K, N; \eta') \leq \frac{d_1 c}{2} \left(1 - \frac{c}{d_1^{a-b}} + \frac{c}{d_1^{a-b}}\left(\max\left(0.5, 1 - \frac{1.5}{d_1^b}\right)\right)^{2K}\right)^N
$$
$$
= \frac{d_1 c}{2}(1 - \Theta(1))^N
$$
$$
= o(S_2(K, N; \eta')),
$$

where the last inequality comes from Lemma I.5. Then we have

$$
\bar{\mathcal{R}}(K, N; \eta') = S_1(K, N; \eta') + S_2(K, N; \eta')
$$
$$
= S_2(K, N; \eta')(1 + o(1))
$$

It is obvious that $\eta^* \in [1, 2]$. We know that

$$
\bar{\mathcal{R}}^*(K, N) = S_1(K, N; \eta^*) + S_2(K, N; \eta^*) \leq \bar{\mathcal{R}}(K, N; \eta') = S_2(K, N, \eta')(1 + o(1)).
$$

By Lemma I.5, we have

$$
S_2(K, N; \eta^*) = \Theta\left(N^{-\frac{a-1}{a-b}}\right) \quad \text{and} \quad S_2(K, N; \eta') = \Theta\left(N^{-\frac{a-1}{a-b}}\right),
$$

which directly implies that

$$
S_1(K, N; \eta^*) = O\left(N^{-\frac{a-1}{a-b}}\right), \qquad \bar{\mathcal{R}}^*(K, N) = \Theta\left(N^{-\frac{a-1}{a-b}}\right).
$$

### I.3  $E(K, N)$ FOR POWER-LAW SPECTRUM: PROOF OF THEOREM 5.2

#### I.3.1  PROOF OF THEOREM 5.2, SMALL-$K$ CASE

Let $T'$ be defined implicitly by equating the averaged risks at their optimal step sizes:

$$
\bar{\mathcal{R}}^*(1, T') = \bar{\mathcal{R}}^*(K, N). \tag{26}
$$

We claim that

$$
\left(\frac{C_4}{C_1}\right)^{\frac{a}{a-1}} T \ \leq \ T' \ \leq \ \left(\frac{C_3}{C_2}\right)^{\frac{a}{a-1}} T. \tag{27}
$$

*Proof.* We argue by contradiction, considering two exclusive violations of Equation (27).

1. **Case 1:** $T' > \left(\frac{C_3}{C_2}\right)^{\frac{a}{a-1}} T$. By the risk bounds encoded by $(C_2, C_3)$ for one-pass training with $T'$ fresh data and by $(C_1, C_4)$ for K-epoch training with $N$ fresh data, this inequality forces
$$\bar{\mathcal{R}}^*(1, T') < \bar{\mathcal{R}}^*(K, N),$$
which contradicts the defining equality Equation (26).

2. **Case 2:** $T' < \left(\frac{C_4}{C_1}\right)^{\frac{a}{a-1}} T$. given $d = \Omega(T^{\frac{1}{a}})$ we still have $d = \Omega\big((T')^{1/a}\big)$. The same risk comparisons then yield
$$\bar{\mathcal{R}}^*(1, T') > \bar{\mathcal{R}}^*(K, N),$$
again contradicting Equation (26).

Both contradictions rule out violations; hence Equation (27) holds. □

Therefore, the desired characterization of $E(K, N)$ follows directly from Lemma J.8.

### I.3.2 PROOF OF THEOREM 5.2, LARGE-$K$ CASE

By Theorem 5.1, there exist constants $C_5, C_6 > 0$ such that, given $d = \Omega(T^{\frac{1}{a}})$,

$$\frac{C_6}{N^{\frac{a-1}{a-b}}} \ \leq \ \bar{\mathcal{R}}^*(K, N) \ \leq \ \frac{C_5}{N^{\frac{a-1}{a-b}}}. \tag{28}$$

Let $T'$ be defined by equating the averaged risks at their optimal step sizes:

$$\bar{\mathcal{R}}^*(K, N) \ = \ \bar{\mathcal{R}}^*(1, T'). \tag{29}$$

Combining Equation (28), Equation (29) with Equation (24), Equation (25), we claim that

$$\left(\frac{C_4}{C_5}\right)^{\frac{a}{a-1}} N^{\frac{a}{a-b}} \ \leq \ T' \ \leq \ \left(\frac{C_3}{C_6}\right)^{\frac{a}{a-1}} N^{\frac{a}{a-b}}. \tag{30}$$

*Proof of the claim.* We argue by contradiction.

1. **Upper violation.** If $T' > \left(\frac{C_3}{C_6}\right)^{\frac{a}{a-1}} N^{\frac{a}{a-b}}$, then by Equation (24) and Equation (28) (lower bound),
$$\bar{\mathcal{R}}^*(1, T') \ \leq \ \frac{C_3}{(T')^{\frac{a-1}{a}}} \ < \ \frac{C_6}{N^{\frac{a-1}{a-b}}} \ \leq \ \bar{\mathcal{R}}^*(K, N),$$
which contradicts Equation (29).

2. **Lower violation.** If $T' < \left(\frac{C_4}{C_5}\right)^{\frac{a}{a-1}} N^{\frac{a}{a-b}}$, then the condition $d = \Omega(T^{\frac{1}{a}})$ gives
$$d = \Omega\left(N^{\frac{1}{a-b}}\right) = \Omega\left((T')^{\frac{1}{a}}\right).$$

Using Equation (25) and Equation (28) (upper bound),
$$\bar{\mathcal{R}}^*(1, T') \ \geq \ \frac{C_4}{(T')^{\frac{a-1}{a}}} \ > \ \frac{C_5}{N^{\frac{a-1}{a-b}}} \ \geq \ \bar{\mathcal{R}}^*(K, N),$$
again contradicting Equation (29).

Both contradictions are impossible; hence Equation (30) holds. □

The characterization of $E(K, N)$ follows directly by the claim.

I.4 SCALING LAWS FOR LOGARITHMIC POWER-LAW SPECTRUM: PROOF OF THEOREM D.1

Similar to the proof of Theorem 5.1, the proof of Theorem 5.3 consists of two parts: First part is the case when $K = o(\log^b N)$, and the second part is the case when $K = \omega(\log^b N)$.

Before we begin our main part of the proof, note that for all $\eta = \Theta(1)$ and $\eta \leq 2$, there exists $d_2 = \Theta(1) > 0$ such that $1 - \frac{\eta}{\log^b(i+1)} > 0$ when $i > d_2$. Then we divide the loss into two parts:

$$\bar{\mathcal{R}}(K, N; \eta) = \frac{1}{2} \sum_{i=1}^{d} \frac{c}{i^a} \left(1 - \frac{c \log^b(i+1)}{i^a} + \frac{c \log^b(i+1)}{i^a} \left(1 - \left(1 - \frac{\eta}{\log^b(i+1)}\right)^{2K}\right)\right)^N$$

$$= \underbrace{\frac{1}{2} \sum_{i=1}^{d_2} \frac{c}{i^a} \left(1 - \frac{c \log^b(i+1)}{i^a} + \frac{c \log^b(i+1)}{i^a} \left(1 - \left(1 - \frac{\eta}{\log^b(i+1)}\right)^{2K}\right)\right)^N}_{V_1(K,N;\eta)}$$

$$+ \underbrace{\frac{1}{2} \sum_{i=d_2+1}^{d} \frac{c}{i^a} \left(1 - \frac{c \log^b(i+1)}{i^a} + \frac{c \log^b(i+1)}{i^a} \left(1 - \left(1 - \frac{\eta}{\log^b(i+1)}\right)^{2K}\right)\right)^N}_{V_2(K,N;\eta)}.$$

I.4.1 PROOF OF THEOREM D.1: SMALL-$K$ CASE

**The Expected Excess Risk Approximization.**

**Lemma I.6.** *Suppose the assumptions in Theorem 5.3 hold. When $K = o(\log^b N)$, we define the estimate of $V(K, N; \eta)$ as*

$$\tilde{V}_2(K, N; \eta) := \frac{1}{2} \sum_{i=1}^{d} \frac{c}{i^a} e^{\frac{-2KNc\eta}{i^a}}.$$

*Then we have $V_2(K, N; \eta) = \tilde{V}(K, N; \eta)(1 + o(1))$, and $\tilde{V}_2(K, N; \eta) \asymp \frac{1}{(KN)^{\frac{a-1}{a}}}$.*

*Proof of Lemma I.6.* We first define a function

$$W(x) := \frac{c}{x^a} \left(1 - \frac{c \log^b(x+1)}{x^a} \left(1 - \left(1 - \frac{\eta}{\log^b(x+1)}\right)^{2K}\right)\right)^N.$$

Direct observation gives us that under Assumption 5.3, $\bar{\mathcal{R}}(K, N; \eta) \propto \sum_{i=1}^{d} W(i)$. Simliarly we take the derivative of $W$.

$$W'(x) = -\frac{ac}{x^{a+1}} \left(1 - \frac{c \log^b(x+1)}{x^a} + \frac{c \log^b(x+1)}{x^a} \left(1 - \left(1 - \frac{\eta}{\log^b(x+1)}\right)^{2K}\right)\right)^N$$

$$+ \frac{cN}{x^a} \left(1 - \frac{c \log^b(x+1)}{x^a} + \frac{c \log^b(x+1)}{x^a} \left(1 - \left(1 - \frac{\eta}{\log^b(x+1)}\right)^{2K}\right)\right)^{N-1}$$

$$\left(\left(\frac{ac \log^b(x+1)}{x^{a+1}} - \frac{bc \log^{b-1}(x+1)}{x^a(x+1)}\right) \left(1 - \left(1 - \frac{\eta}{\log^b(x+1)}\right)^{2K}\right)\right.$$

$$\left.+ \frac{2cK \log^b(x+1)}{x^a} \left(1 - \frac{\eta}{\log^b(x+1)}\right)^{2K-1} \frac{b\eta}{(x+1) \log^{b+1}(x+1)}\right)$$

$$= \frac{c}{x^{2a+1}} \left(1 - \frac{c \log^b(x+1)}{x^a} + \frac{c \log^b(x+1)}{x^a} \left(1 - \left(1 - \frac{\eta}{\log^b(x+1)}\right)^{2K}\right)\right)^{N-1}$$

$$\left(-a \left(x^a - c \log^b(x+1) + c \log^b(x+1) \left(1 - \frac{\eta}{\log^b(x+1)}\right)^{2K}\right)\right.$$

$$+ N\left(\left(ac\log^b(x+1) - bc\log^{b-1}(x+1)\frac{x}{x+1}\right)\left(1 - \left(1 - \frac{\eta}{\log^b(x+1)}\right)^{2K}\right)\right.$$

$$\left.+ \frac{2cKb\eta}{\log(x+1)}\left(1 - \frac{\eta}{\log^b(x+1)}\right)^{2K-1}\frac{x}{x+1}\right)\right).$$

We define

$$G(x) = -a\left(x^a - c\log^b(x+1) + c\log^b(x+1)\left(1 - \frac{\eta}{\log^b(x+1)}\right)^{2K}\right)$$

$$+ N\left(\left(ac\log^b(x+1) - bc\log^{b-1}(x+1)\frac{x}{x+1}\right)\left(1 - \left(1 - \frac{\eta}{\log^b(x+1)}\right)^{2K}\right)\right.$$

$$\left.+ \frac{2cKb\eta}{\log(x+1)}\left(1 - \frac{\eta}{\log^b(x+1)}\right)^{2K-1}\frac{x}{x+1}\right),$$

and $x_0$ is defined to be the maximum of $W(x)$, so $G(x_0) = 0$.

$$G(x) \geq N\log^b(x+1)\left(ac - \frac{bc}{\log(x+1)}\frac{x}{x+1}\right)\left(1 - \left(1 - \frac{\eta}{\log^b x}\right)^{2K}\right) - ax^a$$

$$\geq N(a-b)c\log^b(x+1)\left(1 - \left(1 - \frac{\eta}{\log^b(x+1)}\right)^{2K}\right) - ax^a$$

$$= N(a-b)c\log^b(x+1) \times \frac{\eta}{\log^b(x+1)}\left(\sum_{i=0}^{2K-1}\left(1 - \frac{\eta}{\log^b(x+1)}\right)^i\right) - ax^a$$

$$\geq N(a-b)c\eta - ax^a.$$

So $x_0 = \Omega\left(N^{\frac{1}{a}}\right)$ is an direct conclusion by $G(x_0) = 0$. Also , by solving $G(x_0) = 0$, we can get the approximation of $x_0$ as

$$G(x_0) = -ax_0^a(1 + o(1))$$

$$+ N\left(ac\log^b(x_0+1)(1 + o(1)) \times \frac{2K\eta}{\log^b(x_0+1)}(1 + o(1)) + O\left(\frac{K}{\log N}\right)\right)$$

$$= -ax_0^a(1 + o(1)) + 2KNac\eta(1 + o(1)) = 0,$$

thus we have

$$x_0 = \Theta\left((KN)^{\frac{1}{a}}\right), \quad W(x_0) = \Theta\left(\frac{1}{KN}\right).$$

There exists a constant $N_5$ such that $K \leq \log^b N$ when $N \geq N_5$. So when $N \geq N_5$ and $d \geq 3(KN)^{\frac{1}{a}} \geq 3(KN)^{\frac{1}{a}}\left(\frac{K}{\log^b N}\right)^{\frac{1}{2a}}$, we have

$$V_2(K, N; \eta) = \frac{1}{2}\sum_{i=d_2+1}^{(KN)^{\frac{1}{a}}\left(\frac{K}{\log^b N}\right)^{\frac{1}{2a}}} \frac{c}{i^a}\left(1 - \frac{c\log^b(i+1)}{i^a}\left(1 - \left(1 - \frac{\eta}{\log^b(i+1)}\right)^{2K}\right)\right)^N$$

$$+ \frac{1}{2}\sum_{(KN)^{\frac{1}{a}}\left(\frac{K}{\log^b N}\right)^{\frac{1}{2a}}}^{d} \frac{c}{i^a}\left(1 - \frac{c\log^b(i+1)}{i^a}\left(1 - \left(1 - \frac{\eta}{\log^b(i+1)}\right)^{2K}\right)\right)^N$$

$$:= \psi_1 + \psi_2.$$

Furthermore,

$$\psi_1 \lesssim (KN)^{\frac{1}{a}}\left(\frac{K}{\log^b N}\right)^{\frac{1}{2a}} \times W(x_0) \lesssim \frac{(KN)^{\frac{1}{a}}\left(\frac{K}{\log^b N}\right)^{\frac{1}{2a}}}{KN},$$

and

$$\psi_2 = \frac{1}{2} \sum_{i=(KN)^{\frac{1}{a}}\left(\frac{K}{\log^b N}\right)^{\frac{1}{2a}}}^{d} \frac{c}{i^a}\left(1 - \frac{c\log^b(i+1)}{i^a} + \frac{c\log^b(i+1)}{i^a}\left(1 - \frac{\eta}{\log^b(i+1)}\right)^{2K}\right)^N$$

$$= \frac{1}{2} \sum_{i=(KN)^{\frac{1}{a}}\left(\frac{K}{\log^b N}\right)^{\frac{1}{2a}}}^{d} \frac{c}{i^a}\left(1 - \frac{2Kc\eta}{i^a} + O\left(\frac{K^2}{i^a\log^b(i+1)}\right)\right)^N$$

$$= \frac{1}{2} \sum_{i=(KN)^{\frac{1}{a}}\left(\frac{K}{\log^b N}\right)^{\frac{1}{2a}}}^{d} \frac{c}{i^a}e^{N\log\left(1 - \frac{2Kc\eta}{i^a} + O\left(\frac{K^2}{i^a\log^b(i+1)}\right)\right)}$$

$$= \frac{1}{2} \sum_{i=(KN)^{\frac{1}{a}}\left(\frac{K}{\log^b N}\right)^{\frac{1}{2a}}}^{d} \frac{c}{i^a}e^{\frac{-2KNc\eta}{i^a} + O\left(\frac{K^2 N}{i^{2a}}\right) + O\left(\frac{K^2 N}{i^a\log^b(i+1)}\right)}$$

$$= \frac{1}{2} \sum_{i=(KN)^{\frac{1}{a}}\left(\frac{K}{\log^b N}\right)^{\frac{1}{2a}}}^{d} \frac{c}{i^a}e^{\frac{-2KNc\eta}{i^a}}(1 + o(1)).$$

We recall $K_1(x) = \frac{c}{x^a}e^{\frac{-2KNc\eta}{x^a}}$. We can verify that $\arg\max K_1(x) = \Theta\left((KN)^{\frac{1}{a}}\right)$ and $\max K_1(x) = \Theta\left(\frac{1}{KN}\right)$ through a direct calculation. So for $\psi_2$ we have

$$\psi_2 \geq \frac{1}{2}\sum_{i=(KN)^{\frac{1}{a}}}^{3(KN)^{\frac{1}{a}}} \frac{c}{i^a}e^{\frac{-2KNc\eta}{i^a}}(1 + o(1))$$

$$\gtrsim \frac{(KN)^{\frac{1}{a}}}{KN}.$$

We can verify that $\psi_1 = o(\psi_2)$ as a direct consequence. We define

$$\tilde{V}_2(K, N; \eta) = \frac{1}{2}\sum_{i=d_2+1}^{d} \frac{c}{i^a}e^{\frac{-2KNc\eta}{i^a}}$$

$$= \frac{1}{2}\sum_{i=d_2+1}^{(KN)^{\frac{1}{a}}\left(\frac{K}{\log^b N}\right)^{\frac{1}{2a}}} \frac{c}{i^a}e^{\frac{-2KNc\eta}{i^a}} + \frac{1}{2}\sum_{i=(KN)^{\frac{1}{a}}\left(\frac{K}{\log^b N}\right)^{\frac{1}{2a}}}^{d} \frac{c}{i^a}e^{\frac{-2KNc\eta}{i^a}}$$

$$:= \tilde{\psi}_1 + \tilde{\psi}_2.$$

We have $\psi_2 = \tilde{\psi}_2(1 + o(1))$, and

$$\tilde{\psi}_1 \lesssim \frac{(KN)^{\frac{1}{a}}\left(\frac{K}{\log^b N}\right)^{\frac{1}{2a}}}{KN} = o(\tilde{\psi}_2).$$

So $V_2(K, N; \eta) = \tilde{V}_2(K, N; \eta)(1 + o(1))$.

Finally, we derive a matching upper and lower bound for $\tilde{V}_2(K, N; \eta)$ and conclude the proof:

$$\tilde{V}_2(K, N; \eta) \geq \tilde{J}_2 \gtrsim J_2 \gtrsim \frac{1}{(KN)^{\frac{a-1}{a}}}.$$

$$\tilde{V}_2(K, N; \eta) = \frac{1}{2} \sum_{i=d_2+1}^{(KN)^{\frac{1}{a}}} \frac{c}{i^a} e^{\frac{-2KNc\eta}{i^a}} + \frac{1}{2} \sum_{i=(KN)^{\frac{1}{a}}+1}^{d} \frac{c}{i^a} e^{\frac{-2KNc\eta}{i^a}}$$

$$\leq \frac{1}{2} \sum_{i=1}^{(KN)^{\frac{1}{a}}} \frac{c}{i^a} e^{\frac{-2KNc\eta}{i^a}} + \frac{1}{2} \sum_{i=(KN)^{\frac{1}{a}}+1}^{d} \frac{c}{i^a}$$

$$\lesssim \frac{(KN)^{\frac{1}{a}}}{KN} + \frac{1}{(KN)^{\frac{a-1}{a}}} \lesssim \frac{1}{(KN)^{\frac{a-1}{a}}}.$$

Then we complete the proof. $\qquad\square$

Notice that $\tilde{V}_2(K, N; \eta)$ and $V_2(K, N; \eta)$ are identical to each other, so we can directly apply Lemma I.3 and Lemma I.4 in the remaining proof of Theorem 5.3.

**The Range of Optimal Learning Rate.** First, take $\eta' = 2\log^b(2) - \epsilon$, where $\epsilon := \frac{(a-1)d_2^a}{ac} \frac{\log KN}{KN}$, and we have

$$V_1(K, N; \eta') \leq \frac{d_2 c}{2} \left( 1 - \frac{c\log^b(2)}{d_2^a} + \frac{c\log^b(2)}{d_2^a} \left( 1 - \frac{\epsilon}{\log^b(2)} \right)^{2K} \right)^N$$

$$= \frac{d_2 c}{2} \left( 1 - \frac{2Kc\log^b(2)}{d_2^a} \times \frac{\epsilon}{\log^b(2)}(1 + o(1)) \right)^N$$

$$= \frac{d_2 c}{2} \left( 1 - \frac{2(a-1)}{a} \frac{\log KN}{N}(1 + o(1)) \right)^N$$

$$= \frac{d_2 c}{2} e^{N \log\left(1 - \frac{2(a-1)}{a} \frac{\log KN}{N}(1+o(1))\right)}$$

$$\approx \frac{1}{(KN)^{\frac{2(a-1)}{a}}} = o(V_2(K, N; \eta')),$$

where the last inequality comes from Lemma I.6. Then we have

$$\bar{\mathcal{R}}(K, N; \eta') = V_1(K, N; \eta') + V_2(K, N; \eta')$$

$$= \tilde{V}_2(K, N; \eta')(1 + o(1))$$

$$= \tilde{V}_2(K, N; 2)(1 + o(1))$$

$$= \left( \frac{1}{2} \sum_{i=d_1+1}^{d} \frac{c}{i^a} e^{\frac{-4\log^b(2)KNc}{i^a}} \right) (1 + o(1)).$$

Then we prove that $\eta^* \in [2\log^b(2) - o(1), 2\log^b(2)]$. We prove by contradiction, and assume that there exist a constant $\epsilon > 0$ and a sequence $(N_i)_{i=1}^{\infty} \to \infty$ such that $\eta^*(N_i) \leq 2\log^b(2) - \epsilon$ for all $i \geq 1$. As we only analyze with respect to the sequence $(N_i)_{i=1}^{\infty}$, without loss of generality, we take $(N_i)_{i=1}^{\infty} = \mathbb{N}$. By Lemma I.2, we have

$$\bar{\mathcal{R}}^*(K, N) \geq V_2(K, N; \eta^*) = \tilde{V}_2(K, N; \eta^*)(1 + o(1))$$

$$\geq \left[ \tilde{V}_2(K, N; 2) + \epsilon \frac{\partial}{\partial \eta} \tilde{V}_2 \left( K, N; 2\log^b(2) \right) \right] (1 + o(1)) > \bar{\mathcal{R}}(K, N; \eta')$$

when $N$ is sufficiently large, which is a contradiction. So

$$\bar{\mathcal{R}}^*(K, N) = V_1(K, N; \eta^*) + V_2(K, N; \eta^*)$$

$$= V_1(K, N; \eta^*) + \tilde{V}_2(K, N; \eta^*)(1 + o(1))$$

$$= V_1(K, N; \eta^*) + \tilde{V}_2 \left( K, N; 2\log^b(2) \right) (1 + o(1)) \leq \bar{\mathcal{R}}(K, N; \eta').$$

So $V_1(K, N; \eta^*) = o\left(\tilde{V}_2\left(K, N; 2\log^b(2)\right)\right)$, and $\bar{\mathcal{R}}^*(K, N) = \tilde{V}_2(K, N; 2\log^b(2))(1 + o(1)) \approx \frac{1}{(KN)^{\frac{a-1}{a}}}$.

### I.4.2 Proof of Theorem D.1, Large-$K$ case

**The Expected Excess Risk Approximation.**

**Lemma I.7.** *Suppose the assumptions Theorem 5.3 hold. When $K = \omega(\log^b N)$, we have* $V_2(K, N; \eta) \approx \frac{1}{\left(N \log^b N\right)^{\frac{a-1}{a}}}$.

*Proof of Lemma I.7.* By $K = \omega(\log^b N)$, there exists a constant $N_6 > 0$ such that $K > \log^b N$ when $N \geq N_6$. We notice that when $i = \Theta\left(\left(N\log^b N\right)^{\frac{1}{a}}\right)$, $\log(i+1) = \Theta(\log N)$. Then, when $N \geq N_6$ and $d \geq 3(KN)^{\frac{1}{a}} \geq 3\left(N\log^b N\right)^{\frac{1}{a}}$, we have

$$V_2(K, N; \eta) \geq \frac{1}{2} \sum_{i=\left(N\log^b N\right)^{\frac{1}{a}}}^{3\left(N\log^b N\right)^{\frac{1}{a}}} \frac{c}{i^a}\left(1 - \frac{c\log^b(i+1)}{i^a}\right)^N$$

$$\geq \frac{1}{2}\frac{2\left(N\log^b N\right)^{\frac{1}{a}}}{3^a N\log^b N}(1 - \frac{c_{11}}{N})^N$$

$$\gtrsim \frac{1}{\left(N\log^b N\right)^{\frac{a-1}{a}}}.$$

For the upper bound, we have

$$\bar{\mathcal{R}}(K, N; \eta) \leq \frac{1}{2}\sum_{i=1}^{\infty}\frac{c}{i^a}\left(1 - \frac{c\log^b(i+1)}{i^a} + \frac{c\log^b(i+1)}{i^a}\left(1 - \frac{\eta}{\log^b(i+1)}\right)^{2K}\right)^N$$

$$\leq \frac{1}{2}\sum_{i=1}^{\left(N\log^b N\right)^{\frac{1}{a}}}\frac{c}{i^a}\left(1 - \frac{c\log^b(i+1)}{i^a} + \frac{c\log^b(i+1)}{i^a}\left(1 - \frac{\eta}{\log^b(i+1)}\right)^{2K}\right)^N$$

$$+ \frac{1}{2}\sum_{i=\left(N\log^b N\right)^{\frac{1}{a}}+1}^{\infty}\frac{c}{i^a}.$$

When $K = \omega(\log^b N)$ and $i \leq \left(N\log^b N\right)^{\frac{1}{a}}$,

$$\left(1 - \frac{\eta}{\log^b(i+1)}\right)^K \leq \left(1 - \frac{c_{12}}{\log^b N}\right)^K = e^{K\log\left(1-\frac{c_{12}}{\log^b N}\right)}$$

$$\leq e^{-K\frac{c_{12}}{\log^b N}} = o(1).$$

So there exists $N_7$ such that when $N \geq N_7$, $\left(1 - \frac{\eta}{\log^b(i+1)}\right)^K \leq \frac{1}{2}$, and when $N \geq \max(N_6, N_7)$,

$$\bar{\mathcal{R}}(K, N; \eta) \leq \frac{1}{2}\sum_{i=1}^{\left(N\log^b N\right)^{\frac{1}{a}}}\frac{c}{i^a}\left(1 - \frac{c\log^b(i+1)}{2i^a}\right)^N + \frac{1}{2}\sum_{i=\left(N\log^b N\right)^{\frac{1}{a}}+1}^{\infty}\frac{c}{i^a}.$$

One can derive that $\max_x \frac{c}{x^a}\left(1 - \frac{c\log^b(x+1)}{2x^a}\right)^N = \Theta\left(\frac{1}{N\log^b N}\right)$.

So finally, we have

$$V_2(K, N; \eta) \le \bar{\mathcal{R}}(K, N; \eta) \lesssim \frac{1}{\left(N \log^b N\right)^{\frac{a-1}{a}}} + \frac{1}{\left(N \log^b N\right)^{\frac{a-1}{a}}}$$
$$\lesssim \frac{1}{\left(N \log^b N\right)^{\frac{a-1}{a}}},$$

and we get the result. $\qquad\qquad\qquad\qquad\qquad\qquad\qquad\qquad\qquad\qquad\qquad\square$

**The Range of Optimal Learning Rate.** First, take $\eta' = 1.5 \log^b(2)$, and we have

$$V_1(K, N; \eta') \le \frac{d_2 c}{2} \left(1 - \frac{c \log^b(2)}{d_2^a} + \frac{c \log^b(2)}{d_2^a} \max\left(0.5, 1 - \frac{1.5 \log^b(2)}{\log^b(d_2 + 1)}\right)^{2K}\right)^N$$
$$= \frac{d_1 c}{2} \left(1 - \Theta(1)\right)^N$$
$$= o(V_2(K, N; \eta')),$$

where the last inequality comes from Lemma I.5. Then we have

$$\bar{\mathcal{R}}(K, N; \eta') = V_1(K, N; \eta') + V_2(K, N; \eta')$$
$$= \tilde{V}_2(K, N; \eta')(1 + o(1))$$

It is obvious that $\eta^* \in \left[\log^b(2), 2\log^b(2)\right]$. We know that

$$\bar{\mathcal{R}}^*(K, N) = V_1(K, N; \eta^*) + V_2(K, N; \eta^*) \le \bar{\mathcal{R}}(K, N; \eta') = V_2(K, N, \eta')(1 + o(1))$$
$$\overline{\sim} \frac{1}{\left(N \log^b N\right)^{\frac{a-1}{a}}}.$$

## I.5 $E(K, N)$ FOR LOGARITHMIC POWER-LAW SPECTRUM: PROOF OF THEOREM 5.3

### I.5.1 PROOF OF THEOREM 5.3, SMALL-$K$ CASE

The proof here is almost a reproduction of the proof in Appendix I.2.1.

### I.5.2 PROOF OF THEOREM 5.3, LARGE-$K$ CASE

Consider the multi-epoch training setting with $d = \Omega\left((KN)^{\frac{1}{a+b}}\right)$. By Lemmas I.7 and J.7, there exist constants $C_7, C_8 > 0$ such that

$$\frac{C_8}{N(\log N)^b} \le \bar{\mathcal{R}}^*(K, N) \le \frac{C_7}{N(\log N)^b}. \tag{31}$$

Let $T'$ be defined by matching the expected risks:

$$\bar{\mathcal{R}}^*(K, N) = \bar{\mathcal{R}}^*(1, T'). \tag{32}$$

In the one-pass case, we use the constants $C_3, C_4 > 0$ (as defined in the proof of Theorem 5.2) to control $\bar{\mathcal{R}}^*(1, T')$.

We claim that

$$\left(\frac{C_4}{C_7}\right)^{\frac{a}{a-1}} N(\log N)^b \le T' \le \left(\frac{C_3}{C_8}\right)^{\frac{a}{a-1}} N(\log N)^b. \tag{33}$$

*Proof of the claim.* We argue by contradiction.

1. **Upper bound violation.** If $T' > \left(\frac{C_3}{C_8}\right)^{\frac{a}{a-1}} N(\log N)^b$, then the one-pass upper bound together with Equation (31) (multi-epoch lower bound) imply

$$\bar{\mathcal{R}}^*(K, N) < \bar{\mathcal{R}}^*(1, T'),$$

   which contradicts the defining equality Equation (32).

2. **Lower bound violation.** If $T' < \left(\frac{C_4}{C_7}\right)^{\frac{a}{a-1}} N(\log N)^b$, then $d = \Omega\left((KN)^{\frac{1}{a+b}}\right)$ yields

$$d = \Omega\big((N(\log N)^b)^{1/a}\big) = \Omega\big((T')^{1/a}\big),$$

   so the one-pass lower bound together with Equation (31) (multi-epoch upper bound) give

$$\bar{\mathcal{R}}^*(K, N) > \bar{\mathcal{R}}^*(1, T'),$$

   again contradicting Equation (32).

Both violations are impossible; hence Equation (33) holds. □

Thus, in the large-$K$ multi-epoch regime, the matched one-epoch training time satisfies $T' = \Theta\big(N(\log N)^b\big)$ up to fixed constants. Therefore, the desired characterization of $E(K, N)$ follows directly.

## J   ADDITIONAL TECHNICAL LEMMAS

**Lemma J.1.** *For any PSD matrix $\boldsymbol{A}$, it holds that*

$$\langle \boldsymbol{H}, \boldsymbol{A} \rangle \leq tr(\boldsymbol{H})\|\boldsymbol{A}\|.$$

*Proof.* We denote the PSD decomposition of $\boldsymbol{H}$ by

$$\boldsymbol{H} = \sum_{i=1}^{d} \lambda_i q_i q_i^\top$$

where $\lambda_i$ and $q_i$ are the eigenvalues and corresponding eigenvectors of $\boldsymbol{H}$. So we get

$$
\begin{aligned}
\langle \boldsymbol{H}, \boldsymbol{A} \rangle &= \left\langle \sum_{i=1}^{d} \lambda_i q_i q_i^\top, \boldsymbol{A} \right\rangle \\
&= \sum_{i=1}^{d} \lambda_i q_i^\top \boldsymbol{A} q_i \\
&\leq \sum_{i=1}^{d} \lambda_i \|\boldsymbol{A}\| \\
&= \mathrm{tr}(\boldsymbol{H})\|\boldsymbol{A}\|,
\end{aligned}
$$

which completes the proof. □

**Lemma J.2.** *When $l \geq 1$, we have*

$$(1+x)^l \leq 1 + 2lx, \ x \in [0, \frac{\log 2}{l}]$$

*Proof.* We define $f(x) := (1+x)^l - (1 + 2lx)$. Calculating the derivative and notice the fact that $2^x - 1 \geq (\log 2)x$, we obtain

$$
\begin{aligned}
f^{'}(x) &= l(1+x)^{l-1} - 2l \\
&\leq l(1 + 2^{\frac{1}{l}} - 1)^{l-1} - 2l \\
&\leq l \times 2^{\frac{l-1}{l}} - 2l \leq 0.
\end{aligned}
$$

The above equation completes the proof. □

**Lemma J.3.** *When $l \geq 1$, we have*

$$(1-x)^{2l} \leq 1 - lx, \ x \in [0, \frac{1}{6l}]$$

.

*Proof.* We define $g(x) := (1-x)^{2l} - (1-lx)$. Calculating the derivative, we obtain

$$g'(x) = -2l(1-x)^{2l-1} + l \leq 0 \quad \text{when} \quad x \in [0, 1 - 2^{-\frac{1}{2l-1}}].$$

Notice that $h(x) = 2^x$ is convex, so for $x \in [0, 1]$, we have

$$h(-x + 0 \times (1-x)) \leq xh(-1) + (1-x)h(0),$$

that is

$$2^{-x} \leq 1 - \frac{x}{2} \quad \text{when} \quad x \in [0, 1].$$

So

$$1 - 2^{-\frac{1}{2l-1}} \geq 1 - \left(1 - \frac{1}{2(2l-1)}\right)$$

$$= \frac{1}{2(2l-1)} \geq \frac{1}{6l} \quad \text{when} \quad l \geq 1,$$

which concludes the proof. $\qquad\square$

**Lemma J.4.** *Given $N$ data points such that $\boldsymbol{x}_0, \cdots \boldsymbol{x}_{n-1} \overset{i.i.d}{\sim} \mathcal{N}(0, \boldsymbol{H})$, and define $\boldsymbol{A} = (\boldsymbol{I} - \eta \boldsymbol{x}_{N-1} \boldsymbol{x}_{N-1}^\top) \cdots (\boldsymbol{I} - \eta \boldsymbol{x}_0 \boldsymbol{x}_0^\top)$. Then we have*

$$\mathbb{E}\|\boldsymbol{A} - \mathbb{E}\boldsymbol{A}\|^l \leq \left(\sqrt{\delta_A \eta^2 N l}\right)^l,$$

*where $\delta_A := \tilde{C} 8eD^4 \log d$ for some absolute constant $\tilde{C} > 0$.*

*Proof.* We define $\boldsymbol{Q} := \boldsymbol{A} - \mathbb{E}\boldsymbol{A}$ for convenience. We can obtain a concentration inequality for $\|\boldsymbol{Q}\|$ due to the boundedness of $\boldsymbol{x}$ according to Theorem 7.1 in Huang et al. (2022).

We define

$$\boldsymbol{Y}_i := \boldsymbol{I} - \eta \boldsymbol{x}_i \boldsymbol{x}_i^\top$$

For any $1 \leq i \leq N$, we can choose $m_i = 1$, and we have

$$\|\boldsymbol{Y}_i - \mathbb{E}\boldsymbol{Y}_i\| = \|\eta(\boldsymbol{H} - \boldsymbol{x}_i \boldsymbol{x}_i^\top)\| \leq 2D^2 \eta := \sigma_i$$

So we know that $M_{\boldsymbol{A}} = 1, v_{\boldsymbol{A}} = 4D^4 \eta^2 N$, and

$$\mathbb{P}\{\|\boldsymbol{Q}\| \geq t\} \leq de^{-\frac{t^2}{2ev_{\boldsymbol{A}}}} = de^{-\frac{t^2}{8eD^4 \eta^2 N}} \text{ when } t^2 \geq 8eD^4 \eta^2 N.$$

Furthermore, we have

$$\mathbb{P}\{\|\boldsymbol{Q}\| \geq t\} \leq e^{-\frac{t^2}{16eD^4 \eta^2 N}} \text{ when } t^2 \geq 16eD^4 \eta^2 N \log d.$$

So there exists a non-negative sub-Gaussian random variable $Z$, s.t

$$\mathbb{P}\{\|\boldsymbol{Q}\| \geq t\} \leq \mathbb{P}\{Z \geq t\} \leq e^{-\frac{t^2}{16eD^4 \eta^2 N}} \text{ when } t^2 \geq 16eD^4 \eta^2 N \log d.$$

Then for all $l \geq 1$, we can get

$$
\begin{aligned}
\mathbb{E}\|\boldsymbol{Q}\|^l &= \mathbb{E}\|\boldsymbol{Q}\|^l (\mathbb{1}_{\{\|\boldsymbol{Q}\| \leq \sqrt{16eD^4\eta^2 N \log d}\}} + \mathbb{1}_{\{\|\boldsymbol{Q}\| > \sqrt{16eD^4\eta^2 N \log d}\}}) \\
&\leq \left( \sqrt{16eD^4\eta^2 N \log d} \right)^l + \mathbb{E}\|\boldsymbol{Q}\|^l \mathbb{1}_{\{\|\boldsymbol{Q}\| > \sqrt{16eD^4\eta^2 N \log d}\}} \\
&\leq \left( \sqrt{16eD^4\eta^2 N \log d} \right)^l + \int_{\sqrt{16eD^4\eta^2 N \log d}}^{+\infty} \mathbb{P}\{\|\boldsymbol{Q}\| \geq t\} l t^{l-1} \, dt \\
&\leq \left( \sqrt{16eD^4\eta^2 N \log d} \right)^l + \int_0^{+\infty} \mathbb{P}\{Z \geq t\} l t^{l-1} \, dt \\
&\leq \left( \sqrt{16eD^4\eta^2 N \log d} \right)^l + \mathbb{E}Z^l \\
&\leq \left( \sqrt{16eD^4\eta^2 N \log d} \right)^l + (\sqrt{C16eD^4\eta^2 Nl \log d})^l \\
&\leq \left( \sqrt{\tilde{C}8eD^4\eta^2 Nl \log d} \right)^l.
\end{aligned}
$$

where $C$ and $\tilde{C}$ are absolute constants, the fifth inequality is due to Proposition 2.5.2 in (Vershynin, 2018). $\qquad\square$

**Lemma J.5.** *For any $l \leq K$, we have*

$$
\mathbb{E} \left\| \prod_{k=1}^l \boldsymbol{A}^{(k)} - (\mathbb{E}\boldsymbol{A})^l \right\| \leq \left( \sqrt{\delta_{\mathrm{A}}\eta^2 Nl} + \|\mathbb{E}\boldsymbol{A}\| \right)^l - \|\mathbb{E}\boldsymbol{A}\|^l,
$$

*where $\delta_{\mathrm{A}}$ is the same positive constant appearing in Lemma J.4.*

*Proof.* Let $a = \|\mathbb{E}\boldsymbol{A}\|$ and $c_s = \sqrt{\tilde{C}8eD^4\eta^2 Ns \log d}$ for some $s$. Define the perturbation $\boldsymbol{Q}^{(k)} = \boldsymbol{A}^{(k)} - \mathbb{E}\boldsymbol{A}$. Expanding the product as

$$
\prod_{k=1}^l \boldsymbol{A}^{(k)} = \prod_{k=1}^l \left( \boldsymbol{Q}^{(k)} + \mathbb{E}\boldsymbol{A} \right) = \sum_{m=0}^l \sum_{\mathcal{S} \in \binom{[l]}{m}} \boldsymbol{P}_{\mathcal{S}},
$$

where $\boldsymbol{P}_{\mathcal{S}}$ is the matrix product with $\boldsymbol{Q}^{(k)}$ at positions $k \in \mathcal{S}$ and $\mathbb{E}\boldsymbol{A}$ elsewhere, preserving order. The difference is

$$
\prod_{k=1}^l \boldsymbol{A}^{(k)} - (\mathbb{E}\boldsymbol{A})^l = \sum_{m=1}^l \sum_{\mathcal{S} \in \binom{[l]}{m}} \boldsymbol{P}_{\mathcal{S}}.
$$

By the triangle inequality and linearity of expectation:

$$
\mathbb{E} \left\| \prod_{k=1}^l \boldsymbol{A}^{(k)} - (\mathbb{E}\boldsymbol{A})^l \right\| \leq \sum_{m=1}^l \sum_{\mathcal{S} \in \binom{[l]}{m}} \mathbb{E}\|\boldsymbol{P}_{\mathcal{S}}\|.
$$

For each $\mathcal{S}$, decompose into $t$ maximal consecutive blocks $\mathcal{B}_1, \ldots, \mathcal{B}_t$ with sizes $s_1, \ldots, s_t$ ($\sum s_i = m$). By Hölder inequality and Lemma J.4:

$$
\mathbb{E}\|\boldsymbol{P}_{\mathcal{S}}\| \leq a^{l-m} \mathbb{E} \prod_{i=1}^t \prod_{j \in \mathcal{B}_i} \left\| \boldsymbol{Q}^{(j)} \right\| \leq a^{l-m} \prod_{i=1}^t \prod_{j \in \mathcal{B}_i} \left( \mathbb{E} \left\| \boldsymbol{Q}^{(j)} \right\|^{s_i} \right)^{\frac{1}{s_i}} \leq a^{l-m} \prod_{i=1}^t c_{s_i}.
$$

Since $c_s = \sqrt{\tilde{C}8eD^4\eta^2 Ns \log d}$ is increasing in $s$ and $s_i \leq l$:

$$
c_{s_i} \leq c_l \quad \Rightarrow \quad \mathbb{E}\|\boldsymbol{P}_{\mathcal{S}}\| \leq a^{l-m} c_l^m.
$$

Summing over all $\mathcal{S}$ with $|\mathcal{S}| = m$:

$$\sum_{\mathcal{S} \in \binom{[l]}{m}} \mathbb{E}\|\boldsymbol{P}_{\mathcal{S}}\| \leq \binom{l}{m} a^{l-m} c_l^m.$$

Thus the total bound is:

$$\sum_{m=1}^{l} \binom{l}{m} a^{l-m} c_l^m = (a + c_l)^l - a^l,$$

completing the proof. $\qquad \square$

**Lemma J.6.** *For any $l \leq K$, it holds that*

$$\mathbb{E}\left\|\prod_{k=1}^{l} \boldsymbol{A}^{(k)} - (\mathbb{E}\boldsymbol{A})^l\right\|^2 \leq \left[\left(\sqrt{2\delta_{\mathrm{A}}\eta^2 Nl} + \|\mathbb{E}\boldsymbol{A}\|\right)^l - \|\mathbb{E}\boldsymbol{A}\|^l\right]^2,$$

*where $\delta_{\mathrm{A}}$ is the same positive constant appearing in Lemma J.4.*

*Proof.* Set $a = \|\mathbb{E}\boldsymbol{A}\|_2$ and $c_l = \sqrt{\widetilde{C}16eD^4\eta^2 Nl \log d}$. Define the perturbation $\boldsymbol{Q}^{(k)} = \boldsymbol{A}^{(k)} - \mathbb{E}\boldsymbol{A}$. Expand the matrix product as:

$$\prod_{k=1}^{l} \boldsymbol{A}^{(k)} = \prod_{k=1}^{l} \left(\boldsymbol{Q}^{(k)} + \mathbb{E}\boldsymbol{A}\right) = \sum_{m=0}^{l} \sum_{\mathcal{S} \in \binom{[l]}{m}} \boldsymbol{P}_{\mathcal{S}},$$

where $\boldsymbol{P}_{\mathcal{S}}$ denotes the ordered matrix product with $\boldsymbol{Q}^{(k)}$ at positions $k \in \mathcal{S}$ and $\mathbb{E}\boldsymbol{A}$ elsewhere. The target difference is:

$$\prod_{k=1}^{l} \boldsymbol{A}^{(k)} - (\mathbb{E}\boldsymbol{A})^l = \sum_{m=1}^{l} \sum_{\mathcal{S} \in \binom{[l]}{m}} \boldsymbol{P}_{\mathcal{S}}.$$

For the squared spectral norm, we have:

$$\mathbb{E}\left\|\sum_{m=1}^{l} \sum_{\mathcal{S}} \boldsymbol{P}_{\mathcal{S}}\right\|^2 \leq \mathbb{E}\left(\sum_{m=1}^{l} \sum_{\mathcal{S}} \|\boldsymbol{P}_{\mathcal{S}}\|\right)^2$$

$$= \sum_{m=1}^{l} \sum_{n=1}^{l} \sum_{\mathcal{S}_m} \sum_{\mathcal{S}_n} \mathbb{E}\left[\|\boldsymbol{P}_{\mathcal{S}_m}\| \|\boldsymbol{P}_{\mathcal{S}_n}\|\right],$$

where $\mathcal{S}_m$ and $\mathcal{S}_n$ range over all subsets of $[l]$ with sizes $m$ and $n$, respectively. For each pair $(\mathcal{S}_m, \mathcal{S}_n)$, decompose the union $\mathcal{U} = \mathcal{S}_m \cup \mathcal{S}_n$ into $t$ maximal consecutive blocks $\mathcal{B}_1, \ldots, \mathcal{B}_t$ with sizes $s_i = |\mathcal{B}_i|$ ($\sum_{i=1}^{t} s_i = |\mathcal{U}| = m + n$). By Hölder inequality and Lemma J.4:

$$\mathbb{E}\left[\|\boldsymbol{P}_{\mathcal{S}_m}\| \|\boldsymbol{P}_{\mathcal{S}_n}\|\right] \leq a^{2l-m-n} \mathbb{E}\prod_{i=1}^{t} \prod_{j \in \mathcal{B}_i} \|\boldsymbol{Q}_j\|$$

$$\leq a^{2l-m-n} \prod_{i=1}^{t} \prod_{j \in \mathcal{B}_i} \mathbb{E}\left(\|\boldsymbol{Q}_j\|^{m+n}\right)^{\frac{1}{m+n}}$$

$$\leq a^{2l-m-n} \left(\sqrt{\widetilde{C}8eD^4\eta^2 N(m+n)\log d}\right)^{m+n}$$

$$\leq a^{2l-m-n} c_l^{m+n}.$$

The combinatorial count satisfies:

$$\sum_{\mathcal{S}_m} \sum_{\mathcal{S}_n} 1 = \binom{l}{m}\binom{l}{n}.$$

Combining all terms:

$$\mathbb{E}\left\|\prod_{k=1}^{l}\boldsymbol{A}^{(k)}-(\mathbb{E}\boldsymbol{A})^l\right\|^2 \le \sum_{m=1}^{l}\sum_{n=1}^{l}\binom{l}{m}\binom{l}{n}a^{2l-m-n}c_l^{m+n} = \left[(a+c_l)^l - a^l\right]^2,$$

where the last equality follows from the binomial theorem applied to $(a+c_l)^{2l}$. □

**Lemma J.7.** *Consider a function of training time $T$ given by*

$$\mathcal{L}(T) = \frac{1}{2}\sum_{i=d_1+1}^{d}\frac{c}{i^l}e^{-\frac{2Tc\eta}{i^a}},$$

*where $c, l$ are some absolute constants, $d_1 = \Theta(1)$, and $l > 1$. Then we have:*

1. $\mathcal{L}(T) \lesssim \frac{1}{T^{\frac{l-1}{a}}}$;

2. *Given $d = \Theta\left((KN)^{\frac{1}{a}}\right)$, $\mathcal{L}(T) \gtrsim \frac{1}{T^{\frac{l-1}{a}}}$.*

*Proof.* Computing the derivative of $f(x) = \frac{c}{x^l}e^{-\frac{2Tc\eta}{x^a}}$, we have

$$\arg\max_x f(x) = \Theta\left((KN)^{\frac{1}{a}}\right),$$

$$\max_x f(x) = \Theta\left(\frac{1}{(KN)^{\frac{l}{a}}}\right).$$

Then

1. For the upper bound, we have

$$\mathcal{L}(T) \le \frac{1}{2}\sum_{i=d_1+1}^{\infty}\frac{c}{i^l}e^{-\frac{2Tc\eta}{i^a}} \le \frac{1}{2}\sum_{i=d_1+1}^{(KN)^{\frac{1}{a}}}\frac{c}{i^l}e^{-\frac{2Tc\eta}{i^a}} + \frac{1}{2}\sum_{i=(KN)^{\frac{1}{a}}+1}^{\infty}\frac{c}{i^l}$$

$$\lesssim (KN)^{\frac{1}{a}} \times \frac{1}{(KN)^{\frac{l}{a}}} + \frac{1}{(KN)^{\frac{l-1}{a}}} \lesssim \frac{1}{(KN)^{\frac{l-1}{a}}}.$$

2. For the lower bound, when $d \ge 3T^{\frac{1}{a}}$, we have

$$\mathcal{L}(T) \ge \frac{1}{2}\sum_{i=(KN)^{\frac{1}{a}}}^{3(KN)^{\frac{1}{a}}}\frac{c}{i^l}e^{-\frac{2Tc\eta}{i^a}} \ge \frac{1}{2}\frac{c}{3^l(KN)^{\frac{l}{a}}}e^{-2c\eta} \times 2(KN)^{\frac{1}{a}} \gtrsim \frac{1}{(KN)^{\frac{l-1}{a}}}.$$

The above equation comletes the proof. □

**Lemma J.8.** *Given an estimator of the excess risk for ME and OP cases*

$$\tilde{S}_2(K, N; \eta) = \frac{1}{2}\sum_{i=d_1+1}^{d}\frac{c}{i^a}e^{\frac{-2KNc\eta}{i^a}},$$

*and*

$$\tilde{S}_2(1, T'; \eta) = \frac{1}{2}\sum_{i=d_1+1}^{d}\frac{c}{i^a}e^{\frac{-2T'c\eta}{i^a}}$$

*for some $d_1 = \Theta(1)$. If the ME excess risk and OP excess risk satisfy that*

$$\bar{\mathcal{R}}(K, N; \eta) = \tilde{S}_2(K, N; \eta)(1 + o(1))$$

$$\bar{\mathcal{R}}(1, T'; \eta) = \tilde{S}_2(1, T'; \eta)(1 + o(1)),$$

*then give $d = \Omega(T^{\frac{1}{a}})$ and when $T' \asymp T$, it holds that*

$$E(K, N) \in \left[K(1 - o(1)), K(1 + o(1))\right].$$

*Proof.* We define $H(T) = \tilde{S}_2(K, N; \eta)$ and $\alpha = \frac{T'}{T}$. By definition of $E(K, N)$, we have $T' = E(K, N)N$. Our goal is to prove that $\alpha \in [1 - o(1), 1 + o(1)]$.

Solving $\bar{\mathcal{R}}(K, N; \eta) = \bar{\mathcal{R}}(1, T'; \eta)$, we can get $H(T)(1 + o_N(1)) = H(T')(1 + o_{T'}(1))$. We define $\delta(K, N) = \frac{\bar{\mathcal{R}}(K, N; \eta) - \tilde{S}_2(K, N; \eta)}{\tilde{S}_2(K, N; \eta)} = o(1)$, and $\delta(1, T')) = \frac{\bar{\mathcal{R}}(1, T'; \eta) - \tilde{S}_2(1, T'; \eta)}{\tilde{S}_2(1, T'; \eta)} = o(1)$. Then we can derive that

$$H(T')(1 - \delta(1, T')) \leq H(T)(1 + \delta(K, N))$$
$$H(T')(1 + \delta(1, T')) \geq H(T)(1 - \delta(K, N))$$

which indicates that

$$-\delta(1, T')H(T') - \delta(K, N)H(T) \leq H(T') - H(T) \leq \delta(1, T')H(T') + \delta(K, N)H(T).$$

Notice that $H(T)$ is strongly convex, and we have $H(T) \asymp \frac{1}{(KN)^{\frac{a-1}{a}}}$ and $H'(T) = \frac{1}{2} \sum_{i=1}^{d} \frac{c}{i^{2a}} e^{\frac{-2KNc\eta}{i^a}} \asymp \frac{1}{(KN)^{\frac{2a-1}{a}}}$ by Lemma J.7. We are now ready to prove that $\alpha \in [1 - o(1), 1 + o(1)]$.

$$-\frac{1}{T^{(2-\frac{1}{a})}}(T' - T) \lesssim H'(T)(T' - T) \leq H(T') - H(T) \leq H'(T')(T' - T) \lesssim -\frac{1}{T'^{(2-\frac{1}{a})}}(T' - T)$$
$$\delta(1, T')H(T') + \delta(K, N)H(T) \lesssim \frac{\delta(1, T')}{T'^{(1-\frac{1}{a})}} + \frac{\delta(K, N)}{T^{(1-\frac{1}{a})}} \lesssim \frac{o(1)}{T^{(1-\frac{1}{a})}}.$$

So

$$\frac{T - T'}{T^{(1-\frac{1}{a})}} \lesssim \frac{o(1)}{T^{(1-\frac{1}{a})}}$$
$$-\frac{o(1)}{T^{(1-\frac{1}{a})}} \lesssim -\frac{1}{T'^{(1-\frac{1}{a})}}(T' - T).$$

Direct calculation yields the result. $\qquad\square$

**Lemma J.9** (Hyper-Contractivity). *Given $d$-dimension random vector $\boldsymbol{x} \sim \mathcal{D}$ satisfying that $\|\boldsymbol{x}\| \leq D$ for some constant $D$, and the covariance matrix $\boldsymbol{H} := \mathbb{E}_{\boldsymbol{x} \sim \mathcal{D}}[\boldsymbol{x}\boldsymbol{x}^\top] = \text{diag}(\lambda_1, \lambda_2, \ldots, \lambda_d)$, where $\lambda_1 \geq \lambda_2 \geq \cdots \geq \lambda_d \geq c$ for some constant $c > 0$, then the following holds:*

$$\mathbb{E}[\boldsymbol{x}\boldsymbol{x}^\top \boldsymbol{P}\boldsymbol{x}\boldsymbol{x}^\top] \leq \alpha \, tr(\boldsymbol{H}\boldsymbol{P})\boldsymbol{H}$$

*for some constant $\alpha > 0$ independent of $\boldsymbol{P}$.*

*Proof.* By Dieuleveut et al. (2017), the above lemma holds for data distributions with a bounded kurtosis along every direction, i.e., there exists a constant $\kappa > 0$ such that

$$\text{for every } \boldsymbol{v} \in \mathbb{R}^d, \ \mathbb{E}\left[\langle \boldsymbol{v}, \boldsymbol{x} \rangle^4\right] \leq \kappa \langle \boldsymbol{v}, \boldsymbol{H}\boldsymbol{v} \rangle^2.$$

So that it suffices to verify the above inequality. Since $\lambda_d \geq c$, we have

$$\langle \boldsymbol{v}, \boldsymbol{H}\boldsymbol{v} \rangle^2 \geq c^2 \|\boldsymbol{v}\|^4.$$

For the left side, by the triangle inequality and that $\|\boldsymbol{x}\|$ is bounded

$$\langle \boldsymbol{v}, \boldsymbol{x} \rangle^4 \leq \|\boldsymbol{v}\|^4 \|\boldsymbol{x}\|^4 \leq D^4 \|\boldsymbol{v}\|^4.$$

Combining the above two inequalities gives

$$\mathbb{E}\left[\langle \boldsymbol{v}, \boldsymbol{x} \rangle^4\right] \leq \frac{D^4}{c^2} \langle \boldsymbol{v}, \boldsymbol{H}\boldsymbol{v} \rangle^2.$$

Now setting $\kappa = \frac{D^4}{c^2}$ completes the proof. $\qquad\square$

## K    THEORETICAL ANALYSIS UNDER LEARNING RATE DECAY

While the main body of this paper focuses on SGD with a constant learning rate, practical training almost always utilizes learning rate schedules. In this section, we study multi-epoch versus one-pass training under a class of polynomially decaying learning rate schedules. We show that one-pass training with a polynomial learning rate schedule is suboptimal relative to the statistical minimax rate $\frac{\sigma^2 d}{N}$ implied by the Cramér–Rao lower bound, which is $\frac{\sigma^2 d}{2N}$ (Jain et al., 2017). Meanwhile, multi-epoch training can almost attain this rate.

### K.1    PROBLEM SETUP

We consider the same linear regression model, i.i.d. training data and the initial parameter value $\boldsymbol{w}_0 = 0$ as in Section 3. In addition, we assume the label noise is bounded almost surely as $|\xi| \leq D'$ for technical convenience. We run SGD algorithm with a learning rate schedule $E = \{\eta_t\}_{t \geq 0}$ (with $\eta_t \leq D^{-2}$), where the sampled index $j_t$ is defined as in Section 3. The update is

$$\boldsymbol{w}_{t+1} = \boldsymbol{w}_t - \eta_t \nabla_{\boldsymbol{w}} \ell(\boldsymbol{w}_t; \boldsymbol{x}_{j_t}, y_{j_t}) = \boldsymbol{w}_t - \eta_t \boldsymbol{x}_{j_t} \boldsymbol{x}_{j_t}^\top (\boldsymbol{w}_t - \boldsymbol{w}^*) + \eta_t \xi_{j_t} \boldsymbol{x}_{j_t}$$

Next, given a $K$-epoch SGD over $N$ data points, with learning rate schedule $E$, we define $\mathcal{W}_{K,N,E}$ to be the distribution of $\boldsymbol{w}_{KN}$. Based on this, we define the expected excess risk of a given $K$-epoch SGD over $N$ data points with learning rate schedule $E$ as $\bar{\mathcal{R}}(K, N; E) := \mathbb{E}_{\boldsymbol{w} \sim \mathcal{W}_{K,N,E}}[\mathcal{R}(\boldsymbol{w})]$. We further assume that Assumption 4.1 holds, and $\boldsymbol{x}$ has a continuous probability density function $p(\boldsymbol{x})$ with upper bound $M$ on a $d$-dimension ball with radius $D$. We also define $\lambda_{\min} := \lambda_d$, and $\kappa_{\boldsymbol{H}} = \frac{\lambda_1}{\lambda_d}$ to be the condition number of $\boldsymbol{H}$.

We then introduce a polynomial learning rate schedule class that we consider.

**Definition K.1** (Polynomial Learning-Rate Schedule Class). *Given the $K$-epoch SGD trained with $N$ fresh data samples, we define a learning rate schedule class $\Gamma :=$ $\{E(\eta_0, b, \alpha) : \eta_0 > 0, b > 0, 0.5 \leq \alpha \leq 1\}$, where $E(\eta_0, b, \alpha) = \{\eta_t(\eta_0, b, \alpha)\}_{t=0}^{KN-1}$ that satisfies $\eta_t = \frac{\eta_0}{1 + bt^\alpha}$.*

### K.2    SUB-OPTIMALITY OF ONE-PASS TRAINING

**Theorem K.1** (Sub-Optimality of One-Pass Training). *For one-pass SGD with dataset size of $N$, given any learning rate schedule $E \in \Gamma$, for some constant $C$, when $N$ is sufficiently large, we have*

$$\bar{\mathcal{R}}(1, N; E) \geq \frac{C\sigma^2 d}{N} \cdot \kappa_{\boldsymbol{H}}.$$

*Proof.* This can be seen directly from the proof of Theorem 1, strongly convex case in Ge et al. (2019). □

Theorem K.1 gives a lower bound for one pass training that scales proportionally to the condition number $\kappa_{\boldsymbol{H}}$, to be specific, $\bar{\mathcal{R}}(1, N; E) \gtrsim \frac{\sigma^2 d}{N} \cdot \kappa_{\boldsymbol{H}}$. Since the condition number can be arbitrarily large, the lower bound indicates that one-pass training may be arbitrarily far from the minimax rate, showing its suboptimality.

### K.3    OPTIMALITY OF MULTI-EPOCH TRAINING

Theorem K.2 shows that multi-epoch training can remove the dependency on the condition number and reach the statistical minimax rate.

**Theorem K.2** (Optimality of Multi-Epoch Training). *For multi-epoch SGD with dataset size $N$, there exists a learning rate schedule $E = E(\frac{1}{D^2}, 1, 1) \in \Gamma$ that satisfies $\bar{\mathcal{R}}(\infty, N; E) = \frac{\sigma^2 d}{2N}(1 + o(1))$.*

Before we begin our main part of the proof, we first give some additional notations, and introduce some technical lemmas.

**Additional Notations.** We denote $\boldsymbol{w}_{i_t}^{k_t} := \boldsymbol{w}_t$ and $\eta_{i_t}^{k_t} := \eta_t$, where $i_t$ and $k_t$ are defined in Section 3. We view the SGD training dynamics as an empirical risk minimization (ERM) problem. To be specific, we define $f_i(\boldsymbol{w}) = \frac{1}{2}(y_i - \langle \boldsymbol{w}, \boldsymbol{x}_i \rangle)^2$, and the ERM problem can be written as

$$\min_{\boldsymbol{w}} \mathcal{F}(\boldsymbol{w}) = \frac{1}{N} \sum_{i=0}^{N-1} f_i(\boldsymbol{w}).$$

We write its solution as $\hat{\boldsymbol{w}}$. Define the empirical covariance and its minimal eigenvalue $\widehat{\boldsymbol{H}} := \frac{1}{N} \sum_{i=1}^{N} \boldsymbol{x}_i \boldsymbol{x}_i^{\top}, \mu := \lambda_{\min}(\widehat{\boldsymbol{H}})$.Note that $\mathcal{F}(\boldsymbol{w})$ is $\mu$-strongly convex. As $\mathbb{P}(\mu = 0) = 0$, we assume that $\mu > 0$. Also, $f_i(\boldsymbol{w})$ is quadratic and $L$-smooth with $L \leq D^2$, since $f_i$ is quadratic with its quadratic coefficient $\boldsymbol{x}_i \boldsymbol{x}_i^T \lesssim D^2 \boldsymbol{I}$. We write $\kappa = \frac{D^2}{\mu} = O\left(\mu^{-1}\right)$. Furthermore, we adopt the convention that $\boldsymbol{w}_N^k = \boldsymbol{w}_0^{k+1}$.

**Technical Lemmas.**

**Lemma K.1.** *Define two events* $\mathcal{G} := \left\{ \mu \geq \frac{\lambda_{\min}}{2} \right\}, \mathcal{B} := \mathcal{G}^c$, *and define* $\delta_N := 2d \cdot \exp\left(-\frac{N\lambda_{\min}^2}{32D^4}\right)$. *Then we have* $\Pr(\mathcal{B}) \leq \delta_N$.

*Proof.* Let

$$\widehat{\boldsymbol{H}} - \boldsymbol{H} = \frac{1}{N} \sum_{i=1}^{N} \boldsymbol{Z}_i, \qquad \boldsymbol{Z}_i := \boldsymbol{x}_i \boldsymbol{x}_i^{\top} - \boldsymbol{H}.$$

Then $\mathbb{E}[\boldsymbol{Z}_i] = 0$, and since $\boldsymbol{x}_i \boldsymbol{x}_i^{\top} \succeq 0$ and $\|\boldsymbol{x}_i\|^2 \leq D^2$, we have $\boldsymbol{x}_i \boldsymbol{x}_i^{\top} \preceq D^2 \boldsymbol{I}$, and also $\boldsymbol{H} \preceq D^2 \boldsymbol{I}$ (because $\lambda_{\max}(\boldsymbol{H}) \leq \mathbb{E}\|\boldsymbol{x}\|^2 \leq D^2$). Hence

$$-D^2 \boldsymbol{I} \preceq \boldsymbol{Z}_i \preceq D^2 \boldsymbol{I}, \qquad \Rightarrow \qquad \|\boldsymbol{Z}_i\| \leq D^2 \text{ a.s.}$$

By Theorem 1.3 in (Tropp, 2012),

$$\Pr\left( \left\| \frac{1}{N} \sum_{i=1}^{N} \boldsymbol{Z}_i \right\| \geq t \right) \leq 2d \cdot \exp\left(-\frac{Nt^2}{8D^4}\right). \tag{34}$$

Substituding $t = \frac{\lambda_{\min}}{2}$,

$$\Pr\left( \left\| \frac{1}{N} \sum_{i=1}^{N} \boldsymbol{Z}_i \right\| \geq \frac{\lambda_{\min}}{2} \right) \leq 2d \cdot \exp\left(-\frac{N\lambda_{\min}^2}{32D^4}\right) = \delta_N.$$

Finally, since

$$\mu = \lambda_{\min}(\widehat{\boldsymbol{H}}) \geq \lambda_{\min}(\boldsymbol{H}) - \|\widehat{\boldsymbol{H}} - \boldsymbol{H}\| = \lambda_{\min} - \|\widehat{\boldsymbol{H}} - \boldsymbol{H}\|,$$

the event $\{\mu < \lambda_{\min}/2\}$ implies $\|\widehat{\boldsymbol{H}} - \boldsymbol{H}\| > \lambda_{\min}/2$. Therefore $\Pr(\mathcal{B}) \leq \delta_N$. $\square$

**Lemma K.2** (Finite negative moments of $\mu$ for large $N$). *Fix any* $p > 0$. *Then there exist constants* $N_0 \in \mathbb{N}$ *and* $C < \infty$, *depending only on* $(p, d, c, D, M)$ *(in particular, independent of* $N$*), such that for all* $N \geq N_0$,
$$\mathbb{E}[\mu^{-p}] \leq C.$$

*Proof.* Let $V_{d-1} := \mathrm{Vol}_{d-1}(B_{d-1}(0,1))$ be the volume of $(d-1)$-dimension unit ball, and define

$$A := 2M V_{d-1} D^{d-1}.$$

Fix any $\boldsymbol{u} \in \mathbb{S}^{d-1}$ and $r \in (0, D)$. Since $p(\boldsymbol{x}) \leq M$ on $\{\|\boldsymbol{x}\| \leq D\}$,

$$\Pr\left(|\boldsymbol{u}^{\top}\boldsymbol{x}| \leq r\right) = \int_{\{\|\boldsymbol{x}\| \leq D, \, |\boldsymbol{u}^{\top}\boldsymbol{x}| \leq r\}} p(\boldsymbol{x}) \, d\boldsymbol{x} \leq M \, \mathrm{Vol}_d\left(\{\|\boldsymbol{x}\| \leq D, \, |\boldsymbol{u}^{\top}\boldsymbol{x}| \leq r\}\right).$$

The set $\{\|\boldsymbol{x}\| \leq D, \, |\boldsymbol{u}^{\top}\boldsymbol{x}| \leq r\}$ is a slab of thickness $2r$ intersected with the radius-$D$ ball, hence its volume is at most $2r$ times the maximal $(d-1)$-dimensional cross-section $V_{d-1}D^{d-1}$, so

$$\Pr\left(|\boldsymbol{u}^{\top}\boldsymbol{x}| \leq r\right) \leq A r \qquad (\boldsymbol{u} \in \mathbb{S}^{d-1}, \, r \in (0, D]). \tag{35}$$

In particular, taking $r = D$ and using $|\boldsymbol{u}^\top \boldsymbol{x}| \leq \|\boldsymbol{x}\| \leq D$ a.s. yields $1 = \Pr(|\boldsymbol{u}^\top \boldsymbol{x}| \leq D) \leq AD$, hence $A \geq 1/D$.

Fix $t \in \left(0, \frac{1}{16A^2N}\right]$ and set

$$r := 2\sqrt{Nt}, \qquad \varepsilon := \frac{\sqrt{Nt}}{D}.$$

Since $A \geq 1/D$, we have $\frac{1}{16A^2N} \leq \frac{D^2}{16N}$, hence $r \leq D/2$ and $\varepsilon \leq 1/4$.

By the variational characterization,

$$\mu = \min_{\|\boldsymbol{u}\|=1} \frac{1}{N} \sum_{i=1}^N (\boldsymbol{u}^\top \boldsymbol{x}_i)^2.$$

If $\mu \leq t$, let $\boldsymbol{u}_\star \in \mathbb{S}^{d-1}$ attain the minimum. Then for every $i$,

$$(\boldsymbol{u}_\star^\top \boldsymbol{x}_i)^2 \leq \sum_{j=1}^N (\boldsymbol{u}_\star^\top \boldsymbol{x}_j)^2 \leq Nt, \quad \text{so} \quad |\boldsymbol{u}_\star^\top \boldsymbol{x}_i| \leq \sqrt{Nt} = \frac{r}{2}.$$

Let $\mathcal{N}_\varepsilon$ be an $\varepsilon$-net of $\mathbb{S}^{d-1}$ in Euclidean norm with $|\mathcal{N}_\varepsilon| \leq (3/\varepsilon)^d$. Pick $\boldsymbol{u} \in \mathcal{N}_\varepsilon$ with $\|\boldsymbol{u} - \boldsymbol{u}_\star\| \leq \varepsilon$. Using $\|\boldsymbol{x}_i\| \leq D$,

$$|\boldsymbol{u}^\top \boldsymbol{x}_i| \leq |\boldsymbol{u}_\star^\top \boldsymbol{x}_i| + \|\boldsymbol{u} - \boldsymbol{u}_\star\| \|\boldsymbol{x}_i\| \leq \sqrt{Nt} + \varepsilon D = 2\sqrt{Nt} = r \qquad \forall i.$$

Therefore,

$$\{\mu \leq t\} \subseteq \bigcup_{\boldsymbol{u} \in \mathcal{N}_\varepsilon} \bigcap_{i=1}^N \{|\boldsymbol{u}^\top \boldsymbol{x}_i| \leq r\}.$$

Using independence and equation 35 (note $r \leq D$),

$$\Pr(\mu \leq t) \leq |\mathcal{N}_\varepsilon| \cdot \sup_{\|\boldsymbol{u}\|=1} \Pr(|\boldsymbol{u}^\top \boldsymbol{x}| \leq r)^N \leq \left(\frac{3}{\varepsilon}\right)^d (Ar)^N$$

$$= \left(\frac{3D}{\sqrt{Nt}}\right)^d \left(2A\sqrt{Nt}\right)^N = (3D)^d (2A)^N N^{\frac{N-d}{2}} t^{\frac{N-d}{2}}. \tag{36}$$

By Lemma K.1, there exists a constant $\gamma$ such that $\Pr(\mu \leq \lambda_d/2) \leq d \exp(-\gamma N)$. For $p > 0$ and any nonnegative $Y$, we have

$$\mathbb{E}[Y^{-p}] = p \int_0^\infty t^{-p-1} \Pr(Y \leq t) \, dt.$$

Apply this to $Y = \mu$. Let

$$t_0 := \frac{1}{16A^2N}.$$

Choose $N_0 := \lceil \max\{d + 2p + 2, (8A^2c)^{-1}\} \rceil$. Then when $N \geq N_0$, we have $N > d + 2p$ and $t_0 \leq \frac{\lambda_d}{2}$. We split the integral at $t_0$ and $\frac{\lambda_d}{2}$:

$$\mathbb{E}[\mu^{-p}] = p \int_0^{t_0} t^{-p-1} \Pr(\mu \leq t) \, dt + p \int_{t_0}^{\frac{\lambda_d}{2}} t^{-p-1} \Pr(\mu \leq t) \, dt + p \int_{\frac{\lambda_d}{2}}^{D^2} t^{-p-1} \Pr(\mu \leq t) \, dt$$

$$=: I_3 + I_4 + I_5.$$

*Bound on $I_5$.* Since $\Pr(\mu \leq t) \leq 1$,

$$I_5 \leq p \int_{\frac{\lambda_d}{2}}^{D^2} t^{-p-1} \, dt = \left(\frac{\lambda_d}{2}\right)^{-p}.$$

*Bound on $I_4$.* For $t \leq \frac{\lambda_d}{2}$, $\Pr(\mu \leq t) \leq \Pr(\mu \leq \frac{\lambda_d}{2})$, hence by Lemma K.1,

$$I_4 \leq p \Pr(\mu \leq \frac{\lambda_d}{2}) \int_{t_0}^{\frac{\lambda_d}{2}} t^{-p-1} \, dt \leq \Pr(\mu \leq \frac{\lambda_d}{2}) t_0^{-p} \leq d \, e^{-\gamma N} (16A^2N)^p.$$

**Bound on $I_3$.** Using equation 36 and $N > d + 2p$,

$$I_3 \leq p(3D)^d (2A)^N N^{\frac{N-d}{2}} \int_0^{t_0} t^{\frac{N-d}{2}-p-1} \, dt = p(3D)^d (2A)^N N^{\frac{N-d}{2}} \cdot \frac{t_0^{\frac{N-d}{2}-p}}{\frac{N-d}{2}-p}.$$

Since $N \geq d + 2p + 2$ implies $\frac{N-d}{2} - p \geq 1$, and $t_0 = \frac{1}{16A^2 N}$, a direct simplification gives

$$I_3 \leq p(3D)^d \, 2^{2d+4p} \, A^{d+2p} \, N^p \, 2^{-N}.$$

Combining the bounds for $I_3$, $I_4$ and $I_5$, we obtain the result. $\square$

**Lemma K.3** (Chung-type lemma for $\alpha < \beta$). *Let $\{\xi_k\}_{k \geq 0}$ be a sequence of positive real numbers. Suppose there exist $k_0 > 0$, $A > 0$ and $\alpha > 0, \beta > 0$ with $\alpha \leq \beta$ such that for all $k \geq 0$,*

$$\xi_{k+1} \leq \exp\left(-\frac{\alpha}{k_0 + k + 1}\right) \xi_k + \frac{A}{(k_0 + k + 1)^{\beta+1}}. \tag{37}$$

*Then for any $K \geq 1$,*

$$\xi_K \leq \exp\left(-\alpha \sum_{i=1}^K \frac{1}{k_0 + i}\right) \xi_0 + \frac{e^{\alpha/(k_0+1)} A}{(k_0 + K)^\alpha} \sum_{k=1}^K (k_0 + k)^{\alpha-\beta-1}. \tag{38}$$

*Moreover, when $\alpha < \beta$, Equation (38) can be further simplified to:*

$$\xi_K \leq \left(\frac{k_0 + 1}{k_0 + K}\right)^\alpha \xi_0 + \frac{e^{\alpha/(k_0+1)} A}{\beta - \alpha} \cdot \frac{\beta - \alpha + 1}{(k_0 + K)^\alpha}. \tag{39}$$

*Proof.* Define for $m \geq 1$

$$a_m := \exp\left(-\frac{\alpha}{k_0 + m}\right), \qquad c_m := \frac{A}{(k_0 + m)^{\beta+1}}.$$

Then we can reweite equation 37 as $\xi_{k+1} \leq a_{k+1}\xi_k + c_{k+1}$. Unrolling the recursion gives

$$\xi_K \leq \xi_0 \prod_{j=1}^K a_j + \left(\prod_{j=1}^K a_j\right) \sum_{k=1}^K \left(\prod_{j=1}^k a_j\right)^{-1} c_k.$$

Note $\prod_{j=1}^m a_j = \exp\left(-\alpha \sum_{i=1}^m \frac{1}{k_0+i}\right)$. Using the standard bound

$$\log \frac{k_0 + m}{k_0 + 1} \leq \sum_{i=1}^m \frac{1}{k_0 + i} \leq \log \frac{k_0 + m}{k_0 + 1} + \frac{1}{k_0 + 1},$$

we obtain for all $m \geq 1$:

$$e^{-\alpha/(k_0+1)} \left(\frac{k_0 + 1}{k_0 + m}\right)^\alpha \leq \prod_{j=1}^m a_j \leq \left(\frac{k_0 + 1}{k_0 + m}\right)^\alpha.$$

Hence $\left(\prod_{j=1}^k a_j\right)^{-1} \leq e^{\alpha/(k_0+1)} \left(\frac{k_0+k}{k_0+1}\right)^\alpha$ and therefore

$$\sum_{k=1}^K \left(\prod_{j=1}^k a_j\right)^{-1} c_k \leq e^{\alpha/(k_0+1)} \frac{A}{(k_0+1)^\alpha} \sum_{k=1}^K (k_0 + k)^{\alpha-\beta-1}.$$

Multiplying by $\prod_{j=1}^K a_j \leq \left(\frac{k_0+1}{k_0+K}\right)^\alpha$ yields equation 38.

When $\alpha < \beta$, since $x \mapsto x^{\alpha-\beta-1}$ is decreasing and convex,

$$\sum_{k=1}^K (k_0 + k)^{\alpha-\beta-1} \leq (k_0 + 1)^{\alpha-\beta-1} + \int_1^K (k_0 + x)^{\alpha-\beta-1} \, dx$$

$$= (k_0 + 1)^{\alpha-\beta-1} + \frac{(k_0 + 1)^{\alpha-\beta} - (k_0 + K)^{\alpha-\beta}}{\beta - \alpha}$$

$$< (k_0 + 1)^{\alpha-\beta-1} + \frac{(k_0 + 1)^{\alpha-\beta}}{\beta - \alpha}] \leq \frac{\beta - \alpha + 1}{\beta - \alpha}.$$

Plugging this into equation 38 and using $\prod_{j=1}^K a_j \leq \left(\frac{k_0+1}{k_0+K}\right)^\alpha$ gives equation 39. $\square$

**Lemma K.4.** *Define*

$$G := \sup_{\mathcal{F}(\boldsymbol{w}) \leq \mathcal{F}(\boldsymbol{w}_0)} \max_{i \in [n]} \|\nabla f_i(\boldsymbol{w})\|.$$

*Then we have*

$$G \leq D \sup_{i \in [n]} |\xi_i| + D^2 \left( \|\hat{\boldsymbol{w}} - \boldsymbol{w}^*\| + \sqrt{\frac{D^2}{2\mu}} \|\hat{\boldsymbol{w}}\| \right). \tag{40}$$

*Proof.* By Section A.1 in (Ahn et al., 2020), we have

$$\{\mathcal{F}(\boldsymbol{w}) \leq \mathcal{F}(\boldsymbol{w}_0)\} \subseteq \{\|\boldsymbol{w} - \hat{\boldsymbol{w}}\|^2 \leq \frac{\mathcal{F}(\boldsymbol{w}_0) - \mathcal{F}(\hat{\boldsymbol{w}})}{2\mu}\}.$$

Note that

$$\mathcal{F}(\boldsymbol{w}_0) - \mathcal{F}(\hat{\boldsymbol{w}}) = \frac{1}{2} (\boldsymbol{w}_0 - \hat{\boldsymbol{w}})^\top \hat{\boldsymbol{H}} (\boldsymbol{w}_0 - \hat{\boldsymbol{w}})$$
$$\leq \frac{D^2 \|\hat{\boldsymbol{w}}\|^2}{2}$$

and

$$\|\nabla f_i(\boldsymbol{w})\| = \| (y_i - \langle \boldsymbol{w}, \boldsymbol{x}_i \rangle) \boldsymbol{x}_i\| = \|\langle \boldsymbol{w}^* - \boldsymbol{w}, \boldsymbol{x}_i \rangle \boldsymbol{x}_i + \xi_i \boldsymbol{x}_i\|$$
$$\leq D|\xi_i| + D^2 \|\boldsymbol{w}^* - \boldsymbol{w}\| \leq D|\xi_i| + D^2 (\|\boldsymbol{w}^* - \hat{\boldsymbol{w}}\| + \|\hat{\boldsymbol{w}} - \boldsymbol{w}\|).$$

Therefore,

$$G \leq \sup_{\|\boldsymbol{w} - \hat{\boldsymbol{w}}\| \leq \sqrt{\frac{D^2}{2\mu}} \|\hat{\boldsymbol{w}}\|} \max_{i \in [n]} D|\xi_i| + D^2 (\|\boldsymbol{w}^* - \hat{\boldsymbol{w}}\| + \|\hat{\boldsymbol{w}} - \boldsymbol{w}\|)$$

$$\leq D \sup_{i \in [n]} |\xi_i| + D^2 \left( \|\hat{\boldsymbol{w}} - \boldsymbol{w}^*\| + \sqrt{\frac{D^2}{2\mu}} \|\hat{\boldsymbol{w}}\| \right).$$

$\square$

Combining Lemma K.3, Equation (40) and Proposition 16 in (Ahn et al., 2020), we derive the following corollary.

**Corollary K.1.** *For any $K \geq 1$,*

$$\mathbb{E}_{\pi_1, \cdots, \pi_K} \left\| \boldsymbol{w}_0^{K+1} - \hat{\boldsymbol{w}} \right\|^2 \leq \frac{\|\hat{\boldsymbol{w}}\|^2 + \frac{12\sqrt{e}}{D^2}(3 + 4\kappa) \left( \left( \sup_{i \in [n]} |\xi_i| \right)^2 + D^2 \|\hat{\boldsymbol{w}} - \boldsymbol{w}^*\|^2 + \frac{D^4}{2\mu} \|\hat{\boldsymbol{w}}\|^2 \right)}{(NK)^a}. \tag{41}$$

*Proof.* Let $a := \frac{\mu}{2D^2} \in \left(0, \frac{1}{2}\right), B := G^2 \left(\frac{3}{D^4} + \frac{4\kappa L}{D^6}\right)$, and $\zeta_t := \mathbb{E}\|\boldsymbol{w}_t - \boldsymbol{w}^\star\|^2$ for the global iterates $\{x_t\}_{t=0}^T$ $(T = NK)$. By Proposition 16 in (Ahn et al., 2020), for each update with stepsize $\eta_t$,

$$\zeta_{t+1} \leq \left(1 - \frac{\mu}{2}\eta_t\right)\zeta_t + 3\eta_t^2 G^2 + 4\eta_t^3 \kappa L G^2.$$

Using $1 - x \leq e^{-x}$ and $\eta_t = \frac{1/D^2}{1+t}$, we obtain

$$\zeta_{t+1} \leq \exp\left(-\frac{\mu}{2} \cdot \frac{1/D^2}{1+t}\right)\zeta_t + \frac{3G^2}{D^4(1+t)^2} + \frac{4\kappa L G^2}{D^6(1+t)^3}.$$

With $a = \frac{\mu}{2D^2}$ and $\frac{1}{(1+t)^3} \leq \frac{1}{(1+t)^2}$, this simplifies to

$$\zeta_{t+1} \leq \exp\left(-\frac{a}{1+t}\right)\zeta_t + \frac{B}{(1+t)^2}, \qquad B := G^2 \left(\frac{3}{D^4} + \frac{4\kappa L}{D^6}\right).$$

Now apply Lemma K.3 with
$$k_0 = 0, \qquad \alpha = a, \qquad \beta = 1, \qquad A = B,$$
at $K = T = NK$. This yields that
$$\mathbb{E}_{\pi_1, \cdots, \pi_K} \left\| \boldsymbol{w}_0^{K+1} - \hat{\boldsymbol{w}} \right\|^2 \leq \frac{\|\boldsymbol{w}_0 - \hat{\boldsymbol{w}}\|^2}{(NK)^a} + \frac{2e^a}{1-a} \cdot \frac{B}{(NK)^a}$$
$$\leq \frac{\|\boldsymbol{w}_0 - \hat{\boldsymbol{w}}\|^2 + \frac{4\sqrt{e}G^2}{D^4}(3 + 4\kappa)}{(NK)^a}.$$

Recall that $\boldsymbol{w}_0 = 0$. Combining this inequality with Equation (40), and note that $(a + b + c)^2 \leq 3(a^2 + b^2 + c^2)$, we have the result. $\qquad\square$

**Lemma K.5.** *Define $a_0 := \frac{\lambda_{\min}}{4D^2}$. Then there exists a constant $N_0$ such that when $N > N_0$, we have*
$$\mathbb{E}[\|\boldsymbol{w}_T - \hat{\boldsymbol{w}}\|^2] \leq \frac{C_9}{(KN)^{a_0}} + C_{10}\delta_N^{\frac{1}{2}}. \tag{42}$$

*Proof.* Define $\boldsymbol{X} = (\boldsymbol{x}_1, \cdots, \boldsymbol{x}_N)^\top, \boldsymbol{y} = (y_1, \cdots, y_N)^\top, \boldsymbol{\xi} = (\xi_1, \cdots, \xi_N)^\top$. We have
$$\hat{\boldsymbol{w}} = \left(\boldsymbol{X}^\top \boldsymbol{X}\right)^{-1} \boldsymbol{X}^\top \boldsymbol{y}$$
$$= \left(\boldsymbol{X}^\top \boldsymbol{X}\right)^{-1} \boldsymbol{X}^\top \left(\boldsymbol{X}\boldsymbol{w}^* + \boldsymbol{\xi}\right)$$
$$= \boldsymbol{w}^* + \left(\boldsymbol{X}^\top \boldsymbol{X}\right)^{-1} \boldsymbol{X}^\top \boldsymbol{\xi}. \tag{43}$$

Thus
$$\mathbb{E}_{\boldsymbol{\xi}}\|\hat{\boldsymbol{w}} - \boldsymbol{w}^*\|^2 = \sigma^2 \text{tr}\left(\boldsymbol{X}^\top \boldsymbol{X}\right)^{-1} \leq \frac{\sigma^2 d}{\mu N},$$

and
$$\mathbb{E}_{\boldsymbol{\xi}}\|\hat{\boldsymbol{w}}\|^2 \leq 2\left(\|\hat{\boldsymbol{w}} - \boldsymbol{w}^*\|^2 + \|\boldsymbol{w}^*\|^2\right) \leq \frac{2\sigma^2 d}{\mu N} + 2\|\boldsymbol{w}^*\|^2.$$

Consequently, we have
$$\mathbb{E}_{\boldsymbol{\xi}}\mathbb{E}_{\text{shuffle}} \left\| \boldsymbol{w}_0^{K+1} - \hat{\boldsymbol{w}} \right\|^2 \leq \frac{C_{11}\mu^{-2} + C_{12}\mu^{-1} + C_{13}}{(NK)^a}.$$

Therefore,
$$\mathbb{E}[\|\hat{\boldsymbol{w}} - \boldsymbol{w}^*\|^2] = \mathbb{E}[\|\hat{\boldsymbol{w}} - \boldsymbol{w}^*\|^2 \mathbf{1}_{\mathcal{G}}] + \mathbb{E}[\|\hat{\boldsymbol{w}} - \boldsymbol{w}^*\|^2 \mathbf{1}_{\mathcal{B}}]$$
$$\leq \frac{C_{11}\left(\frac{2}{\lambda_d}\right)^2 + C_{12}\left(\frac{2}{\lambda_d}\right) + C_{13}}{(KN)^{a_0}} + \mathbb{E}[\left(C_{11}\mu^{-2} + C_{12}\mu^{-1} + C_{13}\right)\mathbf{1}_{\mathcal{B}}].$$

By Lemma K.2, we have $\mathbb{E}[\mu^{-4}] < \infty$ when $N > d + 8$. Thus by Hölder and $\Pr(\mathcal{B}) \leq \delta_N$,
$$\mathbb{E}[\kappa \mathbf{1}_{\mathcal{B}}] \leq (\mathbb{E}[\kappa^2])^{1/2} \Pr(\mathcal{B})^{1-1/2}$$
Plugging these bounds into the previous display proves equation 42. $\qquad\square$

Now we begin the main part of the proof of Theorem K.2.

*Proof of Theorem K.2.* Note that through a direct calculation, we have $\mathcal{L}(\boldsymbol{w}) = \frac{\lambda}{2}(\boldsymbol{w}_T - \boldsymbol{w}^*)^2$. Then we can upper bound $\bar{\mathcal{R}}(K, N; E)$:
$$\bar{\mathcal{R}}(K, N; E) = \mathbb{E}\frac{(\boldsymbol{w}_T - \boldsymbol{w}^*)^\top \boldsymbol{H}(\boldsymbol{w}_T - \boldsymbol{w}^*)}{2}$$
$$= \mathbb{E}\frac{(\boldsymbol{w}_T - \hat{\boldsymbol{w}} + \hat{\boldsymbol{w}} - \boldsymbol{w}^*)^\top \boldsymbol{H}(\boldsymbol{w}_T - \hat{\boldsymbol{w}} + \hat{\boldsymbol{w}} - \boldsymbol{w}^*)}{2}$$
$$= \mathbb{E}\frac{1}{2}\langle \boldsymbol{H}, (\hat{\boldsymbol{w}} - \boldsymbol{w}^*)(\hat{\boldsymbol{w}} - \boldsymbol{w}^*)^\top\rangle + \mathbb{E}\frac{1}{2}\langle \boldsymbol{H}, (\boldsymbol{w}_T - \hat{\boldsymbol{w}})(\boldsymbol{w}_T - \hat{\boldsymbol{w}})^\top\rangle$$
$$+ \mathbb{E}(\boldsymbol{w}_T - \hat{\boldsymbol{w}})^\top \boldsymbol{H}(\hat{\boldsymbol{w}} - \boldsymbol{w}^*)$$
$$= I_6 + I_7 + I_8.$$

Next, we characterize $I_6, I_7$ and $I_8$ separately.

By Equation (43), we have

$$I_6 = \mathbb{E} \frac{\sigma^2 d}{2N} \langle \boldsymbol{H}, \hat{\boldsymbol{H}}^{-1} \rangle = \frac{\sigma^2 d}{2N} \left( 1 + \mathbb{E} \langle \boldsymbol{H}, \hat{\boldsymbol{H}}^{-1} - \boldsymbol{H}^{-1} \rangle \right).$$

We view $\frac{\sigma^2 d}{2N}$ as the main term, and view $\frac{\sigma^2 d}{2N} \mathbb{E} \langle \boldsymbol{H}, \hat{\boldsymbol{H}}^{-1} - \boldsymbol{H}^{-1} \rangle$ as the error term. A naive bound of the error term can be given as follows:

$$\mathbb{E} \langle \boldsymbol{H}, \hat{\boldsymbol{H}}^{-1} - \boldsymbol{H}^{-1} \rangle \leq \mathbb{E} D^2 \|\hat{\boldsymbol{H}}^{-1} - \boldsymbol{H}^{-1}\| = \mathbb{E} D^2 \left\| \boldsymbol{H}^{-1} \left( \boldsymbol{H} - \hat{\boldsymbol{H}} \right) \hat{\boldsymbol{H}}^{-1} \right\|$$

$$\leq \mathbb{E} \frac{D^2}{\lambda_d \mu} \|\boldsymbol{H} - \hat{\boldsymbol{H}}\| \leq \frac{D^2}{\lambda_d} \sqrt{\mathbb{E} \mu^{-2}} \sqrt{\mathbb{E} \|\boldsymbol{H} - \hat{\boldsymbol{H}}\|^2}.$$

By Lemma K.2, $\mathbb{E} \mu^{-2} \leq C$ for some constant $C$ independent of $N$; by Equation (34) and Proposition 2.5.2 in (Vershynin, 2018), we have $\mathbb{E} \|\boldsymbol{H} - \hat{\boldsymbol{H}}\|^2 \leq \tilde{C} N^{-0.5}$ for some constant $\tilde{C}$. Therefore, we have $I_6 = \frac{\sigma^2 d}{2N} (1 + o(1))$.

By Lemma K.5, we can derive an upper bound for $I_7$:

$$I_7 \leq \frac{D^2}{2} \mathbb{E} \|\boldsymbol{w}_T - \hat{\boldsymbol{w}}\|^2 \leq \frac{D^2}{2} \left( \frac{C_9}{(KN)^{a_0}} + C_{10} \sqrt{\delta_N} \right),$$

where $\delta_N = o\left(N^{-1}\right)$.

$I_8$ can be bounded by $I_6$ and $I_7$ using the Cauchy-Schwarz inequality:

$$I_8 = \langle \boldsymbol{H}^{0.5}(\boldsymbol{w}_T - \hat{\boldsymbol{w}}), \boldsymbol{H}^{0.5}(\hat{\boldsymbol{w}} - \boldsymbol{w}^*) \rangle \leq \sqrt{2I_7} \sqrt{2I_6} = 2\sqrt{I_6 I_7}.$$

By letting $K \to \infty$ and combining the characterizations of $I_6, I_7$ and $I_8$, we obtain the result. $\qquad \square$

Theorem K.2 together with Theorem K.1 gives a characterization for $E(\infty, N)$. Specifically, we solve

$$\frac{\sigma^2 d}{2N} (1 + o(1)) = \bar{\mathcal{R}}(\infty, N, E) = \bar{\mathcal{R}}(1, N', E) \geq \frac{C \sigma^2 d \kappa_{\boldsymbol{H}}}{N'}.$$

Thus we have

$$E(\infty, N) \geq 2 C \kappa_{\boldsymbol{H}}.$$

$E(\infty, N)$ grows linearly with the condition number $\kappa_{\boldsymbol{H}}$, which can be arbitrarily large. This proves that under practical training setups using polynomial learning rate schedules, multi-epoch training shows a significant advantage over one-pass training.

