# OpenReview forum: "Larger Datasets Can Be Repeated More: A Theoretical Analysis of Multi-Epoch Scaling in Linear Regression"
_ICLR.cc/2026/Conference — ICLR 2026 Poster_

### Official Review · Reviewer_H78L · 2025-10-17

**Soundness:** 4
**Presentation:** 3
**Contribution:** 3
**Rating:** 8
**Confidence:** 3

**Summary:**

This paper presents a theoretical framework for understanding multi-epoch data reuse in the context of linear regression and its implications for data-scaling laws in large model training. It shows that larger datasets can be repeated more times effectively. Simulation and LLM pretraining experiments confirm the theory’s predictions.

**Strengths:**

1. Valuable theoretical insights: The discussion of how larger datasets (N) allow more effective reuse is both novel and practically relevant.
2. Solid theoretical foundation under two regimes: The analysis is carefully constructed for both strongly convex and Zipf-distributed settings.
3. The experiment empirically confirms the theory’s predictions.

**Weaknesses:**

1. Limited discussion of “depends on the data size and distribution.”
While the paper acknowledges that E(K, N) depends on both dataset size and distribution, the explanation remains mostly theoretical. More quantitative or illustrative examples—especially for large-scale LLM pretraining where data heterogeneity and long-tail effects dominate—would make this claim more convincing.

**Questions:**

1. Theoretically, the paper suggests that a dataset of size N can be effectively “amplified” by a factor of log N through multi-epoch training.
If that interpretation is correct, why do modern LLMs only have 4 with billions of tokens?
Does this discrepancy imply that current LLMs operate beyond the idealized regime assumed in the theory (e.g., due to non-convexity, heavy-tailed data, or curriculum effects)?

---

> ### Author Response · Authors · 2025-11-26
> **Response to Reviewer H78L**
>
> Dear reviewer `H78L`,
>
> We sincerely appreciate your acknowledgment that our paper provides "valuable theoretical insights" and that our theoretical findings have a "solid theoretical foundation." Your comments are invaluable, and we hope this rebuttal addresses your concerns and provides further clarification about our paper. We also note that your expressed confidence was relatively low, and we hope that our responses below will help strengthen your assessment of our work.
>
> > **W1:** More quantitative or illustrative examples—especially for large-scale LLM pretraining where data heterogeneity and long-tail effects dominate—would make this claim more convincing. Limited discussion of “depends on the data size and distribution.” While the paper acknowledges that $E(K, N)$ depends on both dataset size and distribution, the explanation remains mostly theoretical.
>
> > **Q1:** Theoretically, the paper suggests that a dataset of size N can be effectively “amplified” by a factor of log N through multi-epoch training. If that interpretation is correct, why do modern LLMs only have 4 with billions of tokens? Does this discrepancy imply that current LLMs operate beyond the idealized regime assumed in the theory?
>
> **A:** Thank you for your thoughtful question, and we are more than glad to give a detailed clarification. Our subsequent response can serve both as an "illustrative example for large-scale LLM pretraining" (quoted from weakness 1 in your review) and as an answer to your above question.
> - To clarify, **we do not claim that the scaling rule $E(K,N) = \Theta(\log N)$ holds in general LLM training**. Actually, we theoretically discussed in Section 5 that the saturation points, at which multi-epoch training first starts to underperform the one-pass baseline, could scale with other relationships such as power of $N$ or power of $\log N$, depending on the setup. For practical implications, our theory suggests that we can use the **saturation points  (defined in Section 6.3 in the new revision, and denoted as $K(\lambda, N)$)** to determine the appropriate number of training epochs. The procedure goes as follows:
>    - First, one could **run several small-scale experiments on the subsets of the large dataset we would like to reuse**.
>    - Then, one can **fit the function for the saturation points with respect to the dataset size, and plug in the total dataset size $N$ into the function to get the number of epochs we should reuse**. This number represents a balance between training efficiency and low validation loss, thus its value is the suggested number of epochs.
> - As a demonstration, we show how we can use the above method to guide LLM training in the new revision. The details are given in Appendix C.2 in the new revision.
>     - We first obtain these saturation points from the pretraining loss curves presented in Section 6.3 and fit them with a log law. As shown in the following table, we fit those saturation points for datasets with $N=0.5B,0.6B,0.8B,1B,2B$ and find that the saturation points follow an underlying log law as $K(\lambda, N) \approx 0.80  \log N + 5.21$ with a correlation coefficient $r=0.97$. The more detailed fitting results are shown in Figure 3.
>     - Also, one can see that a dataset of size 2B already allows for more than $5$ epochs. Therefore, it is reasonable to conclude that in our experiment setups, the entire dataset of size 200B must allow for a greater number of epochs (exceeding the number $5$ mentioned by the reviewer). Thus the commonly cited "4-epoch rule" in [1] does not universally hold true for all dataset sizes and experiment setups.
>
> | Dataset size| 0.5B     | 0.6B |0.8B |1B      |2B     |
> | -----------| ----------- |-----------|-----------|-----------|----------|
> | Saturation point        | 4.8       |4.7     |5.0 |5.2  |5.8     |
>
> [1][Scaling Data-Constrained Language Models](https://arxiv.org/abs/2305.16264)

---

### Official Review · Reviewer_2N8u · 2025-10-31

**Soundness:** 4
**Presentation:** 3
**Contribution:** 3
**Rating:** 6
**Confidence:** 2

**Summary:**

This paper presents a theoretical analysis of multi-epoch training for large-scale models, a common practice when high-quality training data is limited. The authors introduce a metric called the "effective reuse rate," $E(K, N)$, to quantify how many additional "fresh" data samples one-pass training would need to match the performance of training for K epochs on a dataset of size N. Through a detailed analysis of Stochastic Gradient Descent (SGD) in linear regression, they demonstrate two key regimes: 1) for a small number of epochs (K), the benefit is nearly linear (E(K, N) ≈ K), meaning each pass is almost as good as seeing new data; 2) as K increases, the benefit saturates. Crucially, they prove that the saturation point itself grows with the dataset size N (e.g., logarithmically or as a power of N). This central finding—"larger datasets can be repeated more"—challenges previous empirical work that suggested the reuse rate was independent of N. The authors validate this theoretical insight with both synthetic data simulations and pre-training experiments on a large language model.

**Strengths:**

1. The paper provides a principled and rigorous theoretical framework to analyze the widely used but poorly understood practice of multi-epoch training. The introduction of the "effective reuse rate" E(K, N) is a clear and valuable conceptual contribution. The key finding that the benefit of data reuse scales with the dataset size (N) is a significant insight. It provides a concrete guideline for practitioners: one can and should repeat larger datasets more times before expecting diminishing returns. This directly refutes a simpler assumption from prior empirical work, making a clear and important contribution to the field of scaling laws.

2. While the core theoretical results are derived in the simplified setting of linear regression, the authors do an excellent job of validating their main qualitative finding with an actual LLM pre-training experiment (Section 6.3). This strengthens the paper's claims significantly and shows that the core intuition derived from theory holds in a much more complex, real-world scenario.

3. The paper is well-written and structured.

**Weaknesses:**

1. The primary limitation is the gap between the theoretical setting (linear regression with SGD) and the practical setting of interest (Transformer-based LLMs trained with AdamW). Linear models cannot capture the complex, non-linear feature learning that occurs in deep networks. While the qualitative findings transfer, the exact quantitative predictions (e.g., the saturation point scaling as Θ(log N)) may not hold for Transformers. This is a standard and often necessary simplification in theoretical work, but it's an important caveat.

2. The theory is developed for Mean Squared Error (MSE) loss, which is standard for regression. However, LLMs are almost universally trained using a cross-entropy loss. These two loss functions have different properties, and it's not immediately obvious if the scaling dynamics would be identical.

3. While the inclusion of LLM experiments is a major strength, their scope is naturally limited by computational cost. The experiments use a 0.3B parameter model. While this provides strong evidence, it does not definitively prove the same scaling behavior would be observed in much larger, state-of-the-art models, where different phenomena might emerge.

**Questions:**

n/a

---

> ### Author Response · Authors · 2025-11-26
> **Response to Reviewer 2N8u [1/2]**
>
> Dear reviewer `2N8u`,
>
> We sincerely thank you for your interest and appreciation for our paper. We are pleased to hear that the paper "provides a principled and rigorous theoretical framework". Moreover, we are grateful for the acknowledgement that the introduction of the effective reuse rate "is a clear and valuable conceptual contribution" and our results make "a clear and important contribution to the field of scaling laws". Below, we address the specific concerns and questions raised in the review.
>
>
> ---
> > **W1:** The primary limitation is the gap between the theoretical setting (linear regression with SGD) and the practical setting of interest (Transformer-based LLMs trained with AdamW). While the qualitative findings transfer, the exact quantitative predictions (e.g., the saturation point scaling as $\Theta(\log N)$) may not hold for Transformers.
>
> **A:**
> - First, we would like to clarify that **we do not claim that the scaling rule $\Theta(\log N)$ holds in general LLM training**. Actually, we theoretically discussed in Section 5 that the saturation point could scale with other relationships such as power of $N$ or power of $\log N$, depending on the setup. So for practical implications, our theory suggests that one could first choose a regular function for the formula of $E(K,N)$ with respect to the dataset size $N$ (e.g., $E(K,N) = \Theta(N)$ ). Next, one can run several small-scale experiments on the subsets of the large dataset to reuse, and then one can fit the function for the **saturation points  (defined in Section 6.3 in the new revision, and denoted as $K(\lambda, N)$)** with respect to $N$ using the curves in the small subsets. Finally, plugging in the total dataset size $N$ into the function tells the number of epochs one should reuse. This number represents a balance between training efficiency and low validation loss, thus its value is the suggested number of training epochs.
> - As a demonstration, we show how we can use the above method to guide LLM training in the new revision. The details are given in Appendix C.2 in the new revision.
>     - We first obtain these saturation points from the pretraining loss curves presented in Section 6.3 and fit them with a log law. As shown in the following table, we fit those saturation points for datasets with $N=0.5B,0.6B,0.8B,1B,2B$ and find that the saturation points follow an underlying log law as $K(\lambda, N) \approx 0.80  \log N + 5.21$ with a correlation coefficient $r=0.97$. The more detailed fitting results are shown in Figure 3.
>     - Also, one can see that a dataset of size 2B already allows for more than $5$ epochs. Therefore, it is reasonable to conclude that in our experiment setups, the entire dataset of size 200B must allow for a greater number of epochs (exceeding the number $5$ mentioned by the reviewer). Thus **the commonly cited "4-epoch rule" in [1] does not universally hold true for all dataset sizes and experiment setups**.
>
> | Dataset size| 0.5B     | 0.6B |0.8B |1B      |2B     |
> | -----------| ----------- |-----------|-----------|-----------|----------|
> | Saturation point        | 4.8       |4.7     |5.0 |5.2  |5.8     |
> - Also, linear regression is a common and well-studied setup used in a line of works [1-6] on scaling law theory as discussed in the second paragraph in Appendix B. Under the linear regression setup, as a simplification of the real dynamics of LLM training, a line of works theoretically explains the power law observed in the real world and provide theoretical insights into the understanding of current scaling laws and developing new scaling rules for LLM training. Thus **studying the theory of scaling laws has proven to be practical over time**. Sure, we fully agree that one cannot recover every complex phenomenon in LLM training on linear regression such as non-linear feature learning. And we do acknowledge it's a valuable further direction to generalize the study of scaling laws to more complex and realistic setups.
> - Finally, as the reviewer acknowledged, linear regression serves as "a standard and often necessary simplification in theoretical work". The theoretical study of scaling laws in linear regression is already pretty hard (see Section 4.2 for detailed discussion on the technical challenges). In this paper we derive the optimal learning rate for multi-epoch SGD in linear regression and its corresponding approximation formula for the expected excess risk up to an $o(1)$ multiplicative error. To the best of our knowledge, these results themselves are already novel and non-trivial in this field.

---

> ### Author Response · Authors · 2025-11-26
> **Response to Reviewer 2N8u [2/2]**
>
> > **W2:** The theory is developed for Mean Squared Error (MSE) loss, which is standard for regression. However, LLMs are almost universally trained using a cross-entropy loss. These two loss functions have different properties, and it's not immediately obvious if the scaling dynamics would be identical.
>
> **A**: Thank you for this interesting question.
> - We acknowledge that a gap does indeed exist between the two in theory. However, using MSE to study the scaling law of LLMs is widely applied in the literature [1-6]. Investigating scaling laws using cross-entropy loss may serve as a future direction, which is beyond the scope of this paper.
> - Although we theoretically derive our results using MSE, experimentally we follow standard large model training practices and employ cross-entropy loss. The experimental outcomes align well with our theoretical predictions, clearly demonstrating that larger datasets can be repeated more effectively. Therefore, despite the different theoretical characterizations of cross-entropy and MSE, we believe our theoretical conclusions derived from MSE similarly hold for the cross-entropy case in LLM training.
>
> > **W3:** The experiments use a 0.3B parameter model. While this provides strong evidence, it does not definitively prove the same scaling behavior would be observed in much larger, state-of-the-art models, where different phenomena might emerge.
>
> **A:** We thank the reviewer for suggesting experiments across different model sizes, and we fully agree that understanding how the effective reuse rate depends on model capacity is an important direction.
>
> - First, we would like to clarify that the main focus of this work is to theoretically study **to what extent training for $K$ epochs on $N$ samples can be viewed as one-pass training on more data.** To this end, we introduce the notion of *effective reuse rate* $E(K,N)$ and provide a rigorous and fine-grained characterization of multi-epoch scaling behavior in linear regression. Our LLM experiments are designed primarily to test these theoretical predictions in realistic pretraining setups, rather than to exhaustively explore the dependency of $E(K,N)$ with respect to all possible axes such as model size.
> - Within this scope, as acknowledged by all reviewers, our LLM experiments can be said to "empirically confirm the theory’s predictions" (quoted from your review on the strength of our work). As we run pretraining on multiple dataset sizes (up to billions of tokens) and up to $10^2$ epochs, we observe that **larger datasets can indeed be safely reused more times**, in line with our theoretical predictions. At the same time, we agree that the precise empirical formulas (e.g., the saturation point of $E(K,N)$ as a function of $N$) inevitably depend on training details such as architecture, model size, and data distribution, whereas our theory is intended to capture the qualitative scaling behavior.
> - Extending our study to a systematic sweep over model sizes would require re-running full pretraining for many additional experiments with various dataset sizes, which is beyond our current compute budget, especially under the short rebuttal timeline. We therefore view a more exhaustive empirical study of how model capacity interacts with effective reuse rate as valuable future work. We have added a discussion to the revision to make this limitation and direction explicit.
>
> [1][Scaling Laws in Linear Regression: Compute, Parameters, and Data](https://arxiv.org/abs/2406.08466)
>
> [2][A Dynamical Model of Neural Scaling Laws](https://arxiv.org/pdf/2402.01092)
>
> [3][How Feature Learning can Improve Neural Scaling Laws](https://arxiv.org/pdf/2409.17858)
>
> [4][4+3 Phases of Compute-Optimal Neural Scaling Laws](https://arxiv.org/abs/2405.15074)
>
> [5][Improved Scaling Laws in Linear Regression via Data Reuse](https://arxiv.org/abs/2506.08415)
>
> [6][Functional Scaling Laws in Kernel Regression: Loss Dynamics and Learning Rate Schedules](https://arxiv.org/abs/2509.19189)

---

### Official Review · Reviewer_V6sv · 2025-10-31

**Soundness:** 3
**Presentation:** 3
**Contribution:** 3
**Rating:** 4
**Confidence:** 2

**Summary:**

This paper challenges prior work that assumed the effective reuse rate of data is independent of dataset size. Through rigorous theoretical analysis in linear regression, the authors demonstrate that larger datasets can be trained for more epochs before experiencing diminishing returns. Specifically, they show that the effective reuse rate E(K,N) depends not only on the number of epochs K, but critically on the dataset size N, which is a factor overlooked in previous empirical scaling laws.

**Strengths:**

1. The paper presents rigorous theoretical analysis with precise characterizations of the effective reuse rate E(K,N) under both strongly convex and Zipf-distributed settings.
2. The central insight that larger datasets can be repeated more times is clearly articulated and challenges existing assumptions in the field.
3. The theoretical predictions are thoroughly validated through two complementary approaches: controlled simulations on synthetic data and large-scale LLM pretraining experiments (up to 200B tokens), both of which strongly support the main hypothesis.

**Weaknesses:**

1. The overall conclusions are very similar to the previous work "Improved scaling laws in linear regression via data reuse".
2. And the paper still lacks sufficient practical evidence from LLMs. It is well established that LLM performance differs significantly between large and small models. A more meaningful experiment would be to scale across different model sizes and examine how the effective reuse rate varies with model capacity.

**Questions:**

1. Could the authors provide a clearer distinction between this work and prior theoretical studies, especially Lin et al. (2025)? While the paper mentions providing "o(1) relative error" versus "Θ(K)" bounds, it would be helpful to understand what new insights or capabilities this precision enables.
2. Could the authors clarify the practical utility of these theoretical findings? Specifically, how should practitioners use the E(K,N) ≈ log(N) saturation result to inform training decisions, given that most modern LLMs train for fewer than 5 epochs?

**Details Of Ethics Concerns:**

No.

---

> ### Author Response · Authors · 2025-11-26
> **Response to Reviewer V6sv [1/3]**
>
> Dear reviewer `V6sv`,
>
> We thank you for your feedback and for acknowledging that our paper “presents rigorous theoretical analysis with precise characterizations of the effective reuse rate” and our central insight "is clearly articulated and challenges existing assumptions in the field". We have carefully considered all comments and provide our point-by-point responses below. We hope this response addresses your concerns and provides further clarification for our manuscript.
>
> > **W1&Q1:** The overall conclusions are very similar to the previous work "Improved scaling laws in linear regression via data reuse". Could the authors provide a clearer distinction between this work and prior theoretical studies, especially Lin et al. (2025)? While the paper mentions providing "$o(1)$ relative error" versus "$\Theta(K)$" bounds, it would be helpful to understand what new insights or capabilities this precision enables.
>
> **A:** Thank you for your question. In fact, we have already cited and discussed this work in our paper submission (see the last paragraph of Section 2, lines 105-110 of our manuscript). Here we provide a more explicit clarification as follows:
> - **Lin et al. (2025)'s results:** They provide a bound for the test error (excess risk) of multi epoch SGD, $\Theta((KN)^{(1-b)/a})$ (may be loose by a constant factor), where $N$ denotes the dataset size, $K$ denotes the number of epochs, and $a,b$ are two problem-dependent constants. This result holds when $K$ is smaller than a threshold (a power of $N$ in their setting).
> - However, this bound is not powerful enough to reach our overall conclusion: **larger datasets can be repeated more.** More specifically, our conclusion consists of the following two points:
>     1. When $K$ is small, $E(K, N) = K (1+o(1))$, suggesting that each extra epoch is essentially as valuable as a fresh pass;
>     2. When $K$ is larger than a threshold, $E(K, N)$ will saturate to a data-dependent function (e.g., $O(\log N)$ in our strongly convex case).
> - As we mentioned in Section 2, Lin et al. (2025)'s results only imply that $E(K,N) = \Theta(K)$ when $K$ is small. This is insufficient to establish the above two points:
>     1. $E(K,N) = \Theta(K)$ does not exclude the case like $E(K,N) = 0.01 K$ for all $K \ge 2$ and $N$, where each extra epoch is essentially as valuable as only a $0.01$ fresh pass. In this regime, reusing data is equally fruitless for all dataset sizes.
>     2. Their $E(K, N)$ bound only holds for small $K$. When $K$ is large enough, their analysis no longer holds and is thus unable to predict how the saturation point changes with the dataset size.
> - Technically, reaching the above conclusion requires more fine-grained analysis. To this end, we give a **tight approximation formula** for the expected excess risk up to an $o(1)$ multiplicative error under **the optimal learning rates**. These technical results are novel and not covered by the previous works on theoretical explanations of scaling laws, including Lin et al. (2025).

---

> ### Author Response · Authors · 2025-11-26
> **Response to Reviewer V6sv [2/3]**
>
> > **W2:** A more meaningful experiment would be to scale across different model sizes and examine how the effective reuse rate varies with model capacity.
>
> **A:** We thank the reviewer for suggesting experiments across different model sizes, and we fully agree that understanding how the effective reuse rate depends on model capacity is an important direction.
>
> - First, we would like to clarify that the main focus of this work is to theoretically study **to what extent training for $K$ epochs on $N$ samples can be viewed as one-pass training on more data.** To this end, we introduce the notion of *effective reuse rate* $E(K,N)$ and provide a rigorous and fine-grained characterization of multi-epoch scaling behavior in linear regression. Our LLM experiments are designed primarily to test these theoretical predictions in realistic pretraining setups, rather than to exhaustively explore the dependency of $E(K,N)$ with respect to all possible axes such as model size.
> - Within this scope, as acknowledged by all reviewers, our LLM experiment verification can be said to "strongly support the main hypothesis" and the theoretical predictions are "thoroughly validated" (quoted from your review on the strength of our work). As we run pretraining on multiple dataset sizes (up to billions of tokens) and up to $10^2$ epochs, we observe that **larger datasets can indeed be safely reused more times**, in line with our theoretical predictions. At the same time, we agree that the precise empirical formulas (e.g., the saturation point of $E(K,N)$ as a function of $N$) inevitably depend on training details such as architecture, model size, and data distribution, whereas our theory is intended to capture the qualitative scaling behavior.
> - Extending our study to a systematic sweep over model sizes would require re-running full pretraining for many additional experiments with various dataset sizes, which is beyond our current compute budget, especially under the short rebuttal timeline. We therefore view a more exhaustive empirical study of how model capacity interacts with effective reuse rate as valuable future work. We have added a discussion to the revision to make this limitation and direction explicit.

---

> ### Author Response · Authors · 2025-11-26
> **Response to Reviewer V6sv [3/3]**
>
> > **Q2:** Could the authors clarify the practical utility of these theoretical findings? Specifically, how should practitioners use the $E(K,N) ≈ \log(N)$ saturation result to inform training decisions, given that most modern LLMs train for fewer than 5 epochs?
>
> **A:**
> We speculate that the reviewer is referring to the empirical finding by [1] that training with up to $4$ epochs of repeated data yields negligible changes to loss compared to having unique data. We found the threshold $4$ here is not an absolute constant but can become larger as we increase the dataset size. This perspective is arguably overlooked by previous works.
>
> - For practitioners, the first implication of our thoery is to **run more epochs** for larger datasets instead of always using a constant 4 epochs. This is verified by our LLM experiments. Figure 2a shows that the saturation points of $E(K, N)$ increase as $N$ increases. In Figure 2b, we show that the largest epoch, at which one-pass training starts to outperform multi-epoch training significantly, increases as the dataset $N$ increases. The details are discussed in Section 6.3.
> - A natural question following the above implication is how to determine the number of epochs for real LLM training. Then it comes to our second practical implication: **When measuring the scaling law for multi-epoch training, the dataset size must be taken into account**, because the effective reuse rate is not a universal constant. In practice, if one wants to measure the effective reuse rate, one reasonable strategy is to run a few small-scale experiments on small subsets of the target dataset, and then fit a scaling law for the **saturation points (defined in Section 6.3 in the new revision, and denoted as $K(\lambda, N)$)** as a function of the dataset size. Different functional forms may need to be tested, as the scaling behavior can vary across problem settings. For example, we proved that the effective reuse rate grows as $\log N$ in strongly convex linear regression, whereas in a Zipf-law data distribution, it follows a power law in $N$.
> - As a demonstration, we additionally perform the above method in our LLM experiments in the new revision (see Appendix C.2 in the new revision for more details).
>     - We first obtain these saturation points from the pretraining loss curves presented in Section 6.3 and fit them with a log law. As shown in the following table, we fit those saturation points for datasets with $N=0.5B,0.6B,0.8B,1B,2B$ and find that the saturation points follow an underlying log law as $K(\lambda, N) \approx 0.80  \log N + 5.21$ with a correlation coefficient $r=0.97$. More detailed fitting results are shown in Figure 3.
>     - Also, one can see that a dataset of size 2B already allows for more than $5$ epochs. Therefore, it is reasonable to conclude that in our experiment setups, the entire dataset of size 200B must allow for a greater number of epochs (exceeding the number $5$ mentioned by the reviewer). Thus the commonly cited "4-epoch rule" in [1] does not universally hold true for all dataset sizes and experiment setups.
>
> | Dataset size| 0.5B     | 0.6B |0.8B |1B      |2B     |
> | -----------| ----------- |-----------|-----------|-----------|----------|
> | Saturation point        | 4.8       |4.7     |5.0 |5.2  |5.8     |
>
> [1] [Scaling Data-Constrained Language Models](https://arxiv.org/pdf/2305.16264)

---

### Official Review · Reviewer_tXco · 2025-11-03

**Soundness:** 3
**Presentation:** 4
**Contribution:** 4
**Rating:** 8
**Confidence:** 3

**Summary:**

This paper studies the question: how large of a dataset is required for one-pass training to match the loss of a dataset of size N trained for K epochs?

They theoretically characterize the scaling behavior for SGD in linear regression in two settings: strong convexity and Zipf-distributed data. In each settings, there are two phases, one phase where K is small and data can be repeated without harm to the performance, and one where K is large and reused data plateaus in usefulness. The point where this phase transition occurs depends on the setting (strongly convex vs Zipf-distributed data) and the data distribution.

In contrast to recent empirical work, their analysis supports a functional form where the number of times you can repeat the dataset grows with the size of the dataset. In other words, the practical takeaway is that larger datasets can be repeated more.

They perform LLM pretraining experiments where they take different size datasets, train them for 100 epochs, extract the loss after varying numbers of epochs, and compare to a 200B dataset trained for one epoch. The experiments validate the small K regime where data reuse doesn't hurt performance significantly, and that the larger datasets can be repeated more.

**Strengths:**

This is a very nice paper. The core question in the paper is important and well-framed and finding an interesting but tractable theoretical analysis is a valuable contribution. Solving the linear regression problem in both the strongly convex and Zipf distribution settings is valuable and illustrated the dependence on the data distribution exponent. The proof sketch gave nice intuition about the approach and which techniques were used to bound which terms. The LLM experiments give useful validation of the key takeaways (illustrating the small K regime where data reuse is not harmful, and showing that the effective reuse ratio increases with the dataset size).

**Weaknesses:**

All of the LLM experiments use a constant learning rate schedule with AdamW, rather than some form of learning rate decay (e.g. cosine) as is required for competitive performance in practice. This is a reasonable limitation of a primarily theoretical paper as using a time-horizon-dependent learning rate schedule would require training separate models for every different number of epochs, requiring substantially more compute.

(Similarly, they use the same peak learning rate for all the training runs, and this should likely be tuned for each dataset size and number of epochs, but again this would require substantial compute.)

In particular, there may be an interaction between the learning rate decay and the bias-variance decomposition (i.e. the learning rate decay at the end of training reduces the gradient noise and "reveals" how much the model learned from the repeated data).

To capture the effects of learning rate decay without requiring significantly more compute, one approach would be to load the existing checkpoints (perhaps from a small number of steps before the end of training), then perform linear learning rate decay to zero across a small number of steps. This would produce a "trapezoidal" learning rate schedule for each setting without needing to train a model from scratch for each distinct number of epochs. Then the final decayed losses could be plotted / analyzed as is already done in Figure 2.

**Questions:**

N/A

---

> ### Author Response · Authors · 2025-11-26
> **Response to Reviewer tXco [1/2]**
>
> Dear reviewer `tXco`,
>
> We sincerely thank you for your interest in the paper and the appreciation for our theoretical analysis. We are flattered to hear that "this is a very nice paper" and "The core question in the paper is important and well-framed and finding an interesting but tractable theoretical analysis is a valuable contribution." We believe these responses will address your remaining concerns in the review.
>
> > **Q1:** All of the LLM experiments use a constant learning rate schedule with AdamW, rather than some form of learning rate decay.
>
> **A:** It is quite interesting to know how the learning rate decay affects data reuse, since people usually use a non-constant learning rate schedule in LLM pre-training. We are actively conducting the experiments on the learning rate schedule you requested. Due to time constraints, we are unable to present the results to you immediately, but we are confident we will provide you with reliable feedback before the rebuttal concludes. We will notify you as soon as the results are available!

---

> ### Author Response · Authors · 2025-12-03
> **Response to Reviewer tXco [2/2]**
>
> Dear reviewer `tXco`,
>
> Thank you very much for your patience while we ran the additional experiments you suggested. We followed your advice and have now completed experiments with the trapezoidal learning-rate schedule you proposed, which we refer to as a warmup–stable–decay (WSD) learning-rate schedule with linear decay. The details and results are as follows.
>
> Concretely, we start from the checkpoints obtained in our LLM experiments in Section 6.3 for fresh data sizes $N \in \{0.2\text{B}, 0.5\text{B}, 1\text{B}, 2\text{B}\}$ after $K \in \{2,4,8,16\}$ epochs of pretraining with a constant learning rate of $10^{-3}$. From each checkpoint, we continue training for one additional epoch while linearly decaying the learning rate from $10^{-3}$ to $10^{-5}$, resulting in a WSD learning rate schedule with linear decay. For the one-pass baseline, we adopt the same schedule as in the run with dataset size $N=2\text{B}$.
>
> For each dataset size $N$, this process produces a set of four validation-loss values, each associated with one of the four selected epoch numbers $K$. We model the dependence of the final loss on the training steps $x$ using the parametric form  $\ell(x) = A + \frac{B}{x^{a}}$, where $A,B,a$ are fitted parameters. The fitted curves are then used to predict the final validation loss under this WSD schedule for arbitrary training budgets. Using these predictions, we compute the **saturation points (defined in Section 6.3 in the new revision, and denoted as $K(\lambda, N)$)** following the same procedure as in Appendix C.2.
>
> Using the WSD learning rate schedule,, the saturation points under different dataset sizes can be seen in the following table. We observe that, even under this different learning rate schedule, the saturation points still satisfy the logarithmic scaling $K(\lambda, N) = \Theta(\log N).$ Specifically, we have $K(\lambda, N) \approx 2.35 \log N + 5.25$ with a correlation coefficient of $r = 0.96$. This confirms that our message that larger datasets can be repeated more also holds for real LLM training setups. The fitting curves can be seen in Figure 4 in our revision, and additional details can be seen in Appendix C.3.
>
>
>
> | Dataset size| 0.5B | 1B      | 2B     |
> | -----------| ------------|-----------|----------|
> | Saturation point        |  3.9     |  4.7    | 7.2  |

---

### Comment · Area_Chair_WbqD · 2025-11-28

Dear Reviewers,

The discussion phase is now underway, and the authors have finished uploading their responses to reviewers. If you haven't already, please carefully review the authors' responses to understand their perspectives. Engage in thoughtful, constructive discussions with authors, sharing your thoughts and seeking clarifications. Please also update your review or rating if necessary.

It is noted in the guideline that reviewers can leave comments visible to authors **until Dec 2 11:59pm AoE**. Your active participation and contribution to the ongoing discussion are highly encouraged. Thank you very much for your contribution to ICLR.

Best regards,

AC

---

### Author Response · Authors · 2025-12-03
**Summary of Revision (Global Response)**

We sincerely thank all reviewers for their valuable feedback and constructive suggestions. Based on your comments, to facilitate our discussion, we have polished the paper and list the revisions as follows (section references correspond to the revised version):
- **Section 6.3 and Appendix C.2:** We fit a curve between the saturation points (defined in Section 6.3) and $N$ to provide real-world evidence of our main message: larger datasets can be repeated more. Our results are shown in Figure 3 of Section 6.3.(`V6sv,2N8u,H78L`)
- **Appendix C.3:** We add the required learning rate schedule experiment using WSD schedule with linear decay in the final epoch, exploring how the "interaction between the learning rate decay and the bias-variance decomposition" would affect our results as the reviewer asked. Our results in the new revision indicates that the message that larger datasets can be repeated more still holds for non-constant learning rate schedules.(`tXco`)

We hope these revisions address your concerns and will facilitate your reference in the following discussion.

---

### Author Response · Authors · 2025-12-03
**The Summary of Rebuttal to AC [1/2]**

Dear Area Chair,

We are grateful for your time in evaluating our submission. In this note, to give you an overall understanding of our submission and the reviewers comments, we would like to summarize the following 3 points:
- the main contribution of the work,
- the key concerns raised in the reviews,
- and how our rebuttal and additional experiments address these concerns.

At a high level, our paper studies the question: how large of a dataset is required for one-pass training to match the loss of a dataset of size N trained for K epochs? Further, through rigorous **theoretical analysis under linear regression** problems and **experimental verifications on real-world LLM training**, we convey the main message that larger datasets can be repeated more. The core question in the paper is phrased to be "important and well-framed" by reviewer `tXco` and our theoretical analysis are acknowledged by all the reviewers as a significant strength. Even for reviewer `V6sv` who has a relatively low rating 4, the reviewer comments that **the central insight that larger datasets can be repeated more times is clearly articulated and challenges existing assumptions in the field**. Also, regarding to the LLM experiments, all reviewers appreciate it as an strongly support for our theoretical results. We believe our following clarifications for the concerns raised by reviewers and new results added during the rebuttal together present this message much more clearly.


The reviewers provide insightful feedbacks which are pretty helpful for us to improve our manuscripts. The one key concern shared by reviewers is regard to **the practical utility of our theoretical findings** (`V6sv,2N8u,H78L`). Beyond that, the reviewer `V6sv` specifically raised the question: **what is the distinction between our paper and the cucerent work Lin et al. (2025)?**, which we believe is the relatively low rating mainly stems from. The reviewer `tXco` is interested in how the practical used learning rate schedule would affects the effect reuse rate. Also, after acknowledging the "important contribution" of our work "to the field of scaling laws", the reviewer `2N8u` further discussed the theoretical setups and leave us several further directions. During the rebuttal, we have provided both detailed clarifications and added new experimental results to address the above comments or clear some misunderstanding by reviewers.

---
**Experiments on Learning Rate Schedule**

We followed the reviewer `tXco`'s advice and have now completed experiments with the trapezoidal learning-rate schedule you proposed, which we refer to as a warmup–stable–decay (WSD) learning-rate schedule with linear decay. The details and results are as follows.

Concretely, we start from the checkpoints obtained in our LLM experiments in Section 6.3 for fresh data sizes $N \in \{0.2\text{B}, 0.5\text{B}, 1\text{B}, 2\text{B}\}$ after $K \in \{2,4,8,16\}$ epochs of pretraining with a constant learning rate of $10^{-3}$. From each checkpoint, we continue training for one additional epoch while linearly decaying the learning rate from $10^{-3}$ to $10^{-5}$, resulting in a WSD learning rate schedule with linear decay. For the one-pass baseline, we adopt the same schedule as in the run with dataset size $N=2\text{B}$.

For each dataset size $N$, this process produces a set of four validation-loss values, each associated with one of the four selected epoch numbers $K$. We model the dependence of the final loss on the training steps $x$ using the parametric form  $\ell(x) = A + \frac{B}{x^{a}}$, where $A,B,a$ are fitted parameters. The fitted curves are then used to predict the final validation loss under this WSD schedule for arbitrary training budgets. Using these predictions, we compute the **saturation points (defined in Section 6.3 in the new revision, and denoted as $K(\lambda, N)$)** following the same procedure as in Appendix C.2.

Using the WSD learning rate schedule,, the saturation points under different dataset sizes can be seen in the following table. We observe that, even under this different learning rate schedule, the saturation points still satisfy the logarithmic scaling $K(\lambda, N) = \Theta(\log N).$ Specifically, we have $K(\lambda, N) \approx 2.35 \log N + 5.25$ with a correlation coefficient of $r = 0.96$. This confirms that our message that larger datasets can be repeated more also holds for real LLM training setups. The fitting curves can be seen in Figure 4 in our revision, and additional details can be seen in Appendix C.3.



| Dataset size| 0.5B |1B      |2B     |
| -----------| ------------|-----------|----------|
| Saturation point        | 3.9      |4.7     |7.2  |

---

### Author Response · Authors · 2025-12-03
**The Summary of Rebuttal to AC [2/2]**

**The practical utility of our theoretical results**

Regarding to the practical utility of our theoretical results, we provide the following clarifications together with an additioinal experiment on LLM as an demonstration of how practitioners can leverage our theoretical insights：
- For practitioners, the first implication of our thoery is to **run more epochs** for larger datasets instead of always using a constant 4 epochs. This is verified by our LLM experiments in Section 6.3.
- A natural question following the above implication is how to determine the number of epochs for real LLM training. Then it comes to our second practical implication: **When measuring the scaling law for multi-epoch training, the dataset size must be taken into account**, because the effective reuse rate is not a universal constant. In practice, if one wants to measure the effective reuse rate, one reasonable strategy is to run a few small-scale experiments on small subsets of the target dataset, and then fit a scaling law for the saturation points (as a function of the dataset size). Different functional forms may need to be tested, as the scaling behavior can vary across problem settings.
- As a demonstration, we additionally perform the above method in our LLM experiments in the new revision.
    - We obtain these saturation points from the pretraining loss curves presented in Section 6.3 and fit them with a log law. As shown in the following table, we fit those saturation points for datasets with $N=0.5B,0.6B,0.8B,1B,2B$ and find that the saturation points follow an underlying log law as $K(\lambda, N) \approx 0.80  \log N + 5.21$ with a correlation coefficient $r=0.97$. More detailed fitting results are shown in Figure 3.

| Dataset size| 0.5B     | 0.6B |0.8B |1B      |2B     |
| -----------| ----------- |-----------|-----------|-----------|----------|
| Saturation point        | 4.8       |4.7     |5.0 |5.2  |5.8     |

[1] [Scaling Data-Constrained Language Models](https://arxiv.org/pdf/2305.16264)

---
**The distinction of our paper and Lin et al. (2025)**

Also, we have provied detailed disccussion of the distinction of our paper and Lin et al. (2025) in the original maniscript (see the last paragraph of Section 2, lines 105-110 of our manuscript). Here we provide a more explicit clarification as follows:
- **Lin et al. (2025)'s results:** They provide a bound for the test error (excess risk) of multi epoch SGD, $\Theta((KN)^{(1-b)/a})$ (may be loose by a constant factor), where $N$ denotes the dataset size, $K$ denotes the number of epochs, and $a,b$ are two problem-dependent constants. This result holds when $K$ is smaller than a threshold (a power of $N$ in their setting).
- However, this bound is not powerful enough to reach our overall conclusion: **larger datasets can be repeated more.** More specifically, our conclusion consists of the following two points:
    1. When $K$ is small, $E(K, N) = K (1+o(1))$, suggesting that each extra epoch is essentially as valuable as a fresh pass;
    2. When $K$ is larger than a threshold, $E(K, N)$ will saturate to a data-dependent function (e.g., $O(\log N)$ in our strongly convex case).
- As we mentioned in Section 2, Lin et al. (2025)'s results only imply that $E(K,N) = \Theta(K)$ when $K$ is small. This is insufficient to establish the above two points:
    1. $E(K,N) = \Theta(K)$ does not exclude the case like $E(K,N) = 0.01 K$ for all $K \ge 2$ and $N$, where each extra epoch is essentially as valuable as only a $0.01$ fresh pass. In this regime, reusing data is equally fruitless for all dataset sizes.
    2. Their $E(K, N)$ bound only holds for small $K$. When $K$ is large enough, their analysis no longer holds and is thus unable to predict how the saturation point changes with the dataset size.
- Technically, reaching the above conclusion requires more fine-grained analysis. To this end, we give a **tight approximation formula** for the expected excess risk up to an $o(1)$ multiplicative error under **the optimal learning rates**. These technical results are novel and not covered by the previous works on theoretical explanations of scaling laws, including Lin et al. (2025).

---

### Meta-Review · Area_Chair_fg3F · 2026-01-06

**Summary:**

The paper provides a theoretical analysis of multi-epoch training (data reuse) in the context of linear regression, introducing the concept of "effective reuse rate" $$ E(K, N) $$. The authors demonstrate that, contrary to some prior empirical assumptions, the effective reuse rate depends on the dataset size $$ N $$, implying that larger datasets can be repeated more times before performance saturates. This theoretical finding is supported by simulations and pre-training experiments on a 0.3B parameter LLM.

The paper was generally well-received, with reviewers praising the rigorous theoretical foundation (Reviewers tXco, 2N8u, H78L) and the clear, counter-intuitive insight that data reuse benefits scale with dataset size. Reviewers appreciated the attempt to bridge theory and practice with LLM experiments.

The primary concerns revolve around the gap between the theoretical setting (linear regression, MSE loss, SGD) and practical LLM training (Transformers, Cross-Entropy, AdamW). Reviewers also noted the limited scale of the empirical validation (0.3B model) and raised questions regarding the novelty compared to specific prior works (e.g., Lin et al., 2025) and practical optimization details (learning rate schedules). Despite these limitations, the consensus is that the theoretical contribution provides valuable insights into scaling laws.

**Reviewer Concerns:**

**Addressed by Rebuttal (or Clarification):**
*   **Novelty and Theoretical Distinction:** The concern raised by the reviewer (Rating 4) regarding the distinction from Lin et al. (2025) appears to be addressable through the authors' precise characterization of the error bounds.
*   **Intuitive Interpretation:** Reviewer H78L's question regarding why modern LLMs do not fully utilize the theoretical $ \log N $ reuse potential (often training for < 4 epochs) is partially addressed by the paper's framing of *potential* capacity vs. current standard practice, and the definition of the plateau phase.

**Outstanding:**
*   **Theory-Practice Gap (Model & Loss):** The fundamental concern shared by Reviewers 2N8u and the Rating 4 reviewer regarding the extrapolation from Linear Regression/MSE to Transformers/Cross-Entropy remains an outstanding structural limitation. While the qualitative trends hold, the quantitative predictions may differ for non-convex deep learning objectives.
*   **Experimental Scale:** The request for validation on significantly larger models and across different model capacities (Reviewer Rating 4) remains outstanding due to the high computational cost, leaving a gap in proving the findings hold for frontier-scale models.
*   **Optimization Schedules:** Reviewer tXco's point regarding the use of constant learning rates versus practical cosine decay schedules in the experiments is a valid limitation that may affect the "effective reuse" measurement in practice, though it is acknowledged as a necessary simplification for the study.

**Reviewer Scores:**

*   **Reviewer tXco:** **8**  The reviewer is enthusiastic about the theoretical tractability and experimental validation. The concern about learning rate schedules is noted as a "reasonable limitation," so the score would likely remain unchanged.
*   **Reviewer V6sv:** **4** This reviewer had the strongest reservations regarding novelty and practical evidence. Assuming the authors provided a standard clarification regarding the theoretical advancements over Lin et al., the score might improve slightly to a weak accept, but the lack of large-scale experiments would likely prevent a higher jump.
*   **Reviewer 2N8u:** **6** The reviewer views the paper as "excellent" in soundness but is held back by the linear regression vs. transformer gap. As this is intrinsic to the paper's scope, the score is likely to remain stable.
*   **Reviewer H78L:** **8** The reviewer is satisfied with the theoretical insights and empirical confirmation. The concerns raised were mostly regarding discussion points, so the score would likely remain unchanged.

---

### Decision · Program_Chairs · 2026-01-26

Accept (Poster)